# Toward the cellular-scale simulation of motor-driven cytoskeletal assemblies

Wen Yan[1]\*, Saad Ansari[2], Adam Lamson[1,2], Matthew A Glaser[2], Robert Blackwell[1], Meredith D Betterton[1,2,3], Michael Shelley[1,4]\*

[1]Center for Computational Biology, Flatiron Institute, New York, United States; [2]Department of Physics, University of Colorado Boulder, Boulder, United States; [3]Department of Molecular, Cellular, and Developmental Biology, University of Colorado Boulder, Boulder, United States; [4]Courant Institute, New York University, New York, United States

**Abstract** The cytoskeleton – a collection of polymeric filaments, molecular motors, and cross-linkers – is a foundational example of active matter, and in the cell assembles into organelles that guide basic biological functions. Simulation of cytoskeletal assemblies is an important tool for modeling cellular processes and understanding their surprising material properties. Here, we present *aLENS* (a Living Ensemble Simulator), a novel computational framework designed to surmount the limits of conventional simulation methods. We model molecular motors with crosslinking kinetics that adhere to a thermodynamic energy landscape, and integrate the system dynamics while efficiently and stably enforcing hard-body repulsion between filaments. Molecular potentials are entirely avoided in imposing steric constraints. Utilizing parallel computing, we simulate tens to hundreds of thousands of cytoskeletal filaments and crosslinking motors, recapitulating emergent phenomena such as bundle formation and buckling. This simulation framework can help elucidate how motor type, thermal fluctuations, internal stresses, and confinement determine the evolution of cytoskeletal active matter.

\*For correspondence:
wyan@flatironinstitute.org (WY);
mshelley@flatironinstitute.org (MS)

**Competing interest:** The authors declare that no competing interests exist.

## Editor's evaluation

This article presents a new method for simulating cytoskeletal dynamics inside cells. This is an important problem in the life sciences, and the numerical methods and derived results described in the paper seem very promising to facilitate computational modelling of cell dynamics. Although the user- friendliness of the software can still be improved, the method will be of interest to a broad community of biologists and biophysicists.

## Introduction

Living systems are built hierarchically, where smaller structures assemble themselves into larger functional ones. Such organization is fundamental to life, where it is seen across scales from molecules to organelles to cells to tissues to organisms. An example is the cellular cytoskeleton, made up of polymer filaments (and other accessory proteins) crosslinked by motor proteins that exert forces by walking processively along filaments (*Howard, 2001*). Cytoskeletal assemblies such as the cortex, mitotic spindle, and cilia and flagella, underlie cell polarity, division, and movement (*Bornens, 2008*; *Barnhart et al., 2015*; *McIntosh, 2016*; *Pollard and O'Shaughnessy, 2019*). Cytoskeletal components have been reconstituted outside of cells to study self-organization (*Nédélec et al., 1997*; *Foster et al., 2015*) and to create new active materials (*DeCamp et al., 2015*). Understanding how cytoskeletal structures assemble from their molecular components remains challenging, in part because of the

**Figure 1.** The computational model and demonstration of aLENS. (**A**) *aLENS* simulates dynamics of rigid filaments crosslinked and driven by motors, thermal fluctuations, and steric interactions. Motors bind to, unbind from, and walk along filaments. (**B**) To achieve high efficiency, *aLENS* computes motor forces implicitly, and steric interactions through a novel geometric constraint method that avoids filament overlaps. (**C1-C3**) Example simulation of microtubules organized into asters by minus-end-directed motors. The 300 s Brownian simulation contains 3200 microtubules, each 1 µm long, inside a sphere of radius 3 µm. The initial position of each microtubule is random and the half of each filament on the minus-end is colored pink. Three end-pausing dynein motors are fixed at the minus-end of each microtubule and walk toward the minus-end of any microtubule they crosslink. After initial contraction into a single large aster, strong steric interactions in the aster center break up the system into several smaller asters and a bottle-brush structure. (**C4**) Motors are highly concentrated at the centers of asters.

The online version of this article includes the following video for figure 1:

**Figure 1—video 1.** Contraction and break-up of simulated microtubule asters.
https://elifesciences.org/articles/74160/figures#fig1video1

variety of motors and crosslinkers with different behavior. Improved understanding of the cytoskeleton would allow us to predict how molecular perturbations change cell behavior and to design new complex and adaptive materials (*Li and Gundersen, 2008*; *Fletcher and Mullins, 2010*; *Needleman and Dogic, 2017*).

Computational modeling of the cytoskeleton has elucidated principles of self-organization, suggested hypotheses for experimental test, and helped interpret results of experiments (*Gao et al., 2015b*; *Rincon et al., 2017*; *Bun et al., 2018*; *Saintillan et al., 2018*; *Varghese et al., 2020*). Several software packages for cytoskeletal modeling are currently available, including Cytosim (*Nedelec and Foethke, 2007*), MEDYAN (*Popov et al., 2016*), AFINES (*Freedman et al., 2017*), and CyLaKS (*Fiorenza et al., 2021*). A challenge for molecular simulation is the large size of cytoskeletal systems, typically $10^4$–$10^7$ or more filaments (*Petry, 2016*). While current simulations may reach $O(10^4 - 10^5)$ filaments (*Belmonte et al., 2017*; *Strübing et al., 2020*), molecular modeling has required significant compromises in treating steric interactions and motor-proteins.

Here, we describe *aLENS*, a framework of computational methods and software designed to more efficiently and accurately simulate large cytoskeletal systems (*Figure 1*). Since motor proteins must bind, crosslink, and unbind from filaments to evolve such systems, *aLENS* simulates motors as traversing a (well-defined) free energy landscape *Lamson et al., 2021*. This prevents artificial energy flux during crosslinking and maintains detailed balance in the passive limit. As motors crosslink filaments, the spacing between filaments is on the order of the length of motor proteins (10–100 nm) (*Figure 1A*), comparable to the filament diameter. Therefore, steric interactions between filaments occur frequently and must be treated carefully to avoid unphysical filament overlap, stress, and deformation (*Figure 1B*). Most other cytoskeletal simulation methods implement a repulsive pairwise potential between filaments, but this requires a small timestep for hard potentials because of the instability of timestepping methods (*Heyes and Melrose, 1993*). Therefore, potential-based models limit simulations to short timescales. To circumvent this limitation, here we utilize our recently developed constraint method to enforce hard-core repulsion between particles (*Anitescu et al., 1996*; *Yan et al., 2019*). We further develop constraint-based modeling by introducing a related method to treat stiff spring forces due to crosslinking motors. Both steric interactions and crosslinking forces are incorporated in a unified implicit solver. This approach ensures numerical stability of the method and allows for timesteps two or more orders of magnitude larger than currently available. Additionally, *aLENS* is parallelized with OpenMP and MPI to reach length and timescales comparable to those of experiments (Figure 5 and 7).

As an illustration of *aLENS*, *Figure 1C* (and movie *Figure 1—video 1*) shows a simulation of 3200 microtubules within a spherical volume driven by 9600 motors that, when bound, walk to the microtubule minus-end (modeling the activity of dynein). Although the microtubules are initially unorganized (C1), the combination of motor crosslinking and walking causes the microtubule minus-ends to contract into the center of a large aster (C2). The motor-driven steric interactions between filaments, however, eventually fragment this into smaller asters and bottle-brush-like structures (C3,C4). This simulation displays the complex interplay between steric and crosslinking forces in determining the dynamics and steady state configurations of cytoskeletal materials.

## Methodology

In this work, we model filaments as rigid spherocylinders. (While not presented here, flexible filaments can be modeled within our framework as segmented, jointed filaments; See Appendix H.) Crosslinking motors are modeled as Hookean spring tethers connecting two binding domains referred to as heads, with steric interactions between motors neglected.

As outlined below, our algorithm performs three tasks sequentially at every timestep: motor diffusion and stepping, motor binding and unbinding, and filament movement. The major computational challenges arise in task 2, computing binding and unbinding while maintaining realistic macroscopic statistics, and in task 3, updating filament position while overcoming stiffness constraints and maintaining steric exclusion. The timestep is determined by the shortest characteristic timescale in the simulated system (filament collision, motor binding/unbinding kinetics, and filament motion). All other degrees of freedom (e.g. internal conformational changes of motor binding heads) are assumed to occur on shorter timescales.

### 1. Crosslinking motor diffusion and stepping

Each unbound motor executes Brownian motion independently. Each bound motor updates information on the filament to which it is attached, following filament movement in the previous timestep. During the motor movement step, singly bound motors move $v_m \Delta t$ and doubly bound motors move $v_F \Delta t$ along the filaments. Here, $v_F$ is the motor stepping velocity that depends on force on the motor head (*Gao et al., 2015a*):

$$v_F(F_{\text{proj}}) = v_m \max\left(0, \min(1, 1 + F_{\text{proj}}/F_{\text{stall}})\right), \tag{1}$$

where $F_{\text{proj}}$ is the projection of tether force along filament in the stepping direction. As typically found experimentally, this stepping model means that if $F_{\text{proj}}$ is assisting stepping, the velocity saturates at $v_m$; while for $F_{\text{proj}}$ hindering stepping, stepping is halted when $F_{\text{proj}} = -F_{\text{stall}}$.

## 2. Crosslinker binding and unbinding

In filament networks, the spatial variation of unbound and bound motors is integral to network self-organization. For example, crosslinking proteins concentrate in volumes with high filament densities, producing ripening effects as passive crosslinkers are depleted from the bulk (*Weirich et al., 2017*) (e.g. see *Figure 1C*). Furthermore, if motors or crosslinkers bind, unbind, or diffuse at rates not set by free energy barriers, the system's energy and/or entropy can be artificially elevated or lowered, changing the system dynamics and steady-state configuration. Entropic forces bundle and increase overlaps among crosslinked filaments (*Lansky et al., 2015*; *Gaska et al., 2020*), and free-energy-dependent binding kinetics contribute to organization of cortical microtubules (*Allard et al., 2010*) and induce actin bundling (*Yang et al., 2006*).

Ad-hoc models, like those that attach crosslinking motors to filaments at a fixed length or randomly sample a uniform distribution to set the binding length, are unlikely to recover the force or final configuration of bundled filaments. For example, if passive crosslinkers only bind in a non-stretched configuration, they will not generate entropic forces that drive bundle overlap, as seen experimentally (*Lansky et al., 2015*). Further, if crosslinkers are modeled as binding with a uniform length distribution and zero tether rest length, the contractile stress of networks will be overestimated, condensing filament networks with greater rapidity.

The assemblies of filaments/motors are assumed to explore an underlying free energy landscape, where all 'fast' degrees of freedom can be subsumed into an effective free energy that depends only on filament and crosslinking motor degrees of freedom. We require that our model correctly recapitulates the distribution and chemical kinetics of crosslinking proteins in the passive limit, that is, when $v_m = 0$ for the bound velocity of motor heads. We achieve this with a kinetic Monte Carlo procedure in which motor protein binding and unbinding events are modeled as stochastic processes. Transition rates recover the correct limiting (equilibrium) distribution by imposing detailed balance (Appendix

**Table 1.** The transition rates between all possible states of a crosslinker $U \rightleftharpoons (S_A, S_B) \rightleftharpoons D$. $(S_A, S_B)$ means either head $A$ or $B$ is bound but the other is unbound. All binding rates account for the linear binding density $\epsilon$ is the length of filament with center-of-mass position $x_i$ and orientation $p_i$ inside the capture sphere with cutoff radius $r_{c,S}$ relative to position of motor/crosslinker $x$. The sum is over all possible candidate filaments. The unbound-singly bound transition $U \rightleftharpoons (S_A, S_B)$ is determined by the association constant $K_a$ and the force-independent off rate $k_{o,S}$. Similarly, the singly bound-doubly bound transition $(S_A, S_B) \rightleftharpoons D$ is determined by the association constant $K_e$ and force-independent off rate $k_{o,D}$ is the Boltzmann factor. $E(\ell)$ in the in the $(S_A, S_B) \rightleftharpoons D$ transition rates refers to the tether energy of a motor $E(\ell) = \frac{1}{2}\kappa_{xl}\left(\ell_f - \ell_0\right)^2$. $\ell_0$ is the free length of a motor, while $\ell_f$ is the length for computing the force when attached to filaments and $j$ at locations $s_i$ and $s_j$: $\ell_f(s_i, s_j, x_i, p_i, x_j, p_j)$. The dimensionless factor $\lambda$ determines the energy dependence in the unbinding rate. Both binding and unbinding rates must depend on $\lambda$ and $k_{o,d}$ such that the equilibrium constant recovers the Boltzmann factor $\exp[-\beta E(\ell_f)]$ For force-dependent binding models, the $E(\ell)$ can be simply replaced by the tether force $F(\ell)$. This is not used for results shown in this work, but implemented in the code.

| Process | Rate | Value |
|---|---|---|
| $U \rightarrow (S_A, S_B)$ | $R_{\text{on},S}(\boldsymbol{x})$ | $k_{o,S} \dfrac{3\epsilon K_a}{4\pi r_{c,S}^3} \sum_i L_{\text{in},i}(\boldsymbol{x})$ |
| $(S_A, S_B) \rightarrow U$ | $R_{\text{off},S}$ | $k_{o,S}$ |
| $(S_A, S_B) \rightarrow D$ | $R_{\text{on},D}(s_i)$ | $k_{o,D}\epsilon K_e \sum_j \int_{L_j} ds_j \exp\left[-(1-\lambda)\beta E(\ell_f(s))\right]$ |
| $D \rightarrow (S_A, S_B)$ | $R_{\text{off},D}(s_i, s_j)$ | $k_{o,D} \exp\left[\lambda E(\ell_f)\right]$ |

C). That is, we model binding and unbinding as passive processes, but it is in principle possible that certain such processes consume chemical energy.

To enforce the macroscopic thermodynamic statistics, including correct equilibrium bound-unbound concentrations and distributions (Appendix C) (*Gao et al., 2015a*; *Lamson et al., 2019*; *Allard et al., 2010*), we explicitly model each crosslinker as a Hookean spring connecting two binding heads labeled as $A$ or $B$. Each crosslinker has four possible states: both heads unbound ($U$), either $A$ or $B$ singly bound ($S_A$ or $S_B$), or both heads (doubly) bound ($D$). For each timestep $\Delta t$, we first calculate the rates $R(t)$ at which each head ($A$ and $B$) transitions from their current state to a new binding state (i.e. for the transitions $U \rightleftharpoons (S_A, S_B) \rightleftharpoons D$). The transition probabilities are modeled as inhomogeneous Poisson processes with the cumulative probability function

$$P(\Delta t) = 1 - \exp\left(-\int_0^{\Delta t} R(t)dt\right) = 1 - \exp\left(-R(0)\Delta t + O(\Delta t^2)\right). \tag{2}$$

The transitions $U \rightleftharpoons (S_A, S_B)$ do not stretch or compress the tether and so do not depend on tether deformation energy. However, the transitions $(S_A, S_B) \rightleftharpoons D$ do account for tether deformation energy (*Table 1*).

## 3. Filament dynamics

We sought to develop a stable, large-timestep method for updating the position of filaments, subject to spring forces from crosslinking motors, steric interactions, and Brownian motion. This requires addressing two stability restrictions on the timestep $\Delta t$. The first arises in models that use a stiff repulsive pairwise potential to prevent filament overlaps. For example, the Lennard-Jones potential $V \sim (\sigma/r)^{12} - (\sigma/r)^6$, where $r$ is the separation between filaments, is so steeply varying that it requires small $\Delta t$ for stability. As a result, soft alternatives such as a harmonic potential are often used (*Nedelec and Foethke, 2007*). These soft potentials allow partial filament overlaps, and may therefore lead to unphysical system dynamics and stresses (*Heyes and Melrose, 1993*).

The second stability restriction arises from the fast relaxation times of crosslinking motors. When crosslinkers connect two parallel filaments, the spring tether length $\ell_f$ relaxes according to $\dot{\ell}_f = -\lambda(\ell_f - \ell_0)$, where $\ell_0$ is the preferred length and $\lambda = N\kappa_{xl}/(4\pi\eta L/\log(2L/D_{fil}))$ (*Howard, 2001*). Explicit timestepping schemes require $\Delta t < C/\lambda$, for some constant $C$. For $N = 10$ motors, tether stiffness $\kappa_{xl} \approx 100\,\text{pN}\,\mu m^{-1}$, and slender body drag coefficient $4\pi\eta L/\log(2L/D_{fil}) \approx 0.003\,\text{pN s}\,\mu m^{-1}$ for $1\,\mu m$-long microtubules in aqueous solvent, we have $1/\lambda \approx 3 \times 10^{-6}\,\text{s}$.

We overcome these difficulties with a novel, linearized implicit Euler timestepping scheme, which extends on our previous work on enforcing non-overlap conditions (*Yan et al., 2019*). This technique is inspired by constraint-based methods for granular flow (*Tasora et al., 2013*). When collisions occur between filaments, the minimal distance between them attains $\Phi_{col} = 0$ with collision force $\gamma_{col} > 0$. If not colliding, $\Phi_{col} > 0$ and $\gamma_{col} = 0$. This mutually exclusive condition is called a complementarity constraint, written as $0 \leq \Phi_{col} \perp \gamma_{col} \geq 0$. If one crosslinking motor connects these two filaments, its length $\ell_f$ and force magnitude $\gamma_{xl}$ satisfy the Hookean spring model $\gamma_{xl} = -\kappa_{xl}(\ell_f - \ell_0)$, which is an equality constraint.

We integrate the equation of motion such that these two types of constraints for all possible collisions and all crosslinking motors are satisfied. We briefly derive the method here, and all details can be found in Appendix C. Because the method is specific to rigid particles with arbitrary shape, we shall use 'particle' and 'filament' interchangeably.

Each particle is tracked by its center location $x \in \mathbb{R}^3$ in the lab frame and its orientation $\theta = [s, p] \in \mathbb{R}^4$ as a quaternion (*Delong et al., 2015*). $[s, p]$ are the scalar and vector parts of the quaternion, respectively. Using a quaternion to track the rotational kinematics of a rigid body is a standard computational approach due to its compact memory footprint (4 floating point numbers) and its singularity-free nature. The geometric configuration at time $t$ for all $N$ filaments can be written as a column vector with $7N$ entries:

$$\mathcal{C}(t) = \left[x_1, \theta_1, \ldots, x_N, \theta_N\right]^T \in \mathbb{R}^{7N}. \tag{3}$$

Similarly, we use the vectors $\mathcal{U}, \mathcal{F} \in \mathbb{R}^{6N}$ to represent the translational & angular velocities, and forces & torques of all particles, respectively. We relate $\mathcal{U}$ to $\mathcal{F}$ via a mobility matrix $\mathcal{M} \in \mathbb{R}^{6N \times 6N}$, dependent only upon the geometry $\mathcal{C}$, and relate $\mathcal{U}$ to $\dot{\mathcal{C}}(t)$ via a geometric matrix $\mathcal{G}$:

$$\dot{\mathcal{C}}(t) = \mathcal{G}\mathcal{U}, \quad \mathcal{U} = \mathcal{M}\mathcal{F}, \tag{4}$$

Because the biological filaments we consider mostly have lengths on the nm to $\mu m$ scales and inertial effects can be ignored. In the following, the subscript $c$ refers to constraints, which includes both unilateral (with subscript $u$) and bilateral (with subscript $b$) constraints. For our problem, unilateral constraints refer to collision constraints while bilateral constraints refer to crosslinking motor constraints. The subscript $nc$ refers to non-constraint.

For unilateral constraints, we define the grand distance vector $\boldsymbol{\Phi}_u = \left[\Phi_{u,1}, \Phi_{u,2}, \cdots, \Phi_{u,N_u}\right]^T \in \mathbb{R}^{N_u}$, where each $\Phi_{u,j}$ is the minimum distance between a pair of filaments. Similarly, for bilateral constraints we define the grand distance vector $\boldsymbol{\Phi}_b = \left[\ell_{f,1}, \ell_{f,2}, \cdots, \ell_{f,N_b}\right]^T \in \mathbb{R}^{N_b}$, containing the length $\ell_{f,j}$ of the doubly bound motor $j$. There are in total $N_u$ possibly colliding pairs of filaments and $N_b$ cross-linking motors. The force magnitude corresponding to these constraints are also written as vectors, $\boldsymbol{\gamma}_u = \left[\gamma_{u,1}, \gamma_{u,2}, \cdots, \gamma_{u,N_u}\right]^T \in \mathbb{R}^{N_u}$ and $\boldsymbol{\gamma}_b = \left[\gamma_{b,1}, \gamma_{b,2}, \cdots, \gamma_{b,N_b}\right]^T \in \mathbb{R}^{N_b}$. The two types of constraints can be summarized as:

$$0 \leq \boldsymbol{\Phi}_u(\mathcal{C}) \perp \boldsymbol{\gamma}_u \geq 0,$$
$$\mathcal{K}\left[\boldsymbol{\Phi}_b(\mathcal{C}) - \boldsymbol{\Phi}_b^0\right] = -\boldsymbol{\gamma}_b. \tag{5}$$

Here, $\boldsymbol{\Phi}_u$ and $\boldsymbol{\gamma}_u$ satisfy the complementarity (collision) constraints, while $\boldsymbol{\Phi}_b$ and $\boldsymbol{\gamma}_b$ satisfy the Hookean spring law. Here, $\mathcal{K} \in \mathbb{R}^{N_b \times N_b}$ is a diagonal matrix consisting of all the stiffness constants, while $\boldsymbol{\Phi}_b^0$ represents the rest length of every crosslinking motor.

*Equations 4 and 5* define a differential-variational-inequality (DVI). This is solvable when closed by a geometric relation mapping the force magnitude $\boldsymbol{\gamma}_u$ and $\boldsymbol{\gamma}_b$ to the force vectors $\mathcal{F}_u$ and $\mathcal{F}_b$:

$$\mathcal{F}_u = \mathcal{D}_u \boldsymbol{\gamma}_u, \quad \mathcal{F}_b = \mathcal{D}_b \boldsymbol{\gamma}_b, \tag{6}$$

where $\mathcal{D}_u$ and $\mathcal{D}_b$ are sparse matrices containing the orientation norm vectors of all constraint forces (*Anitescu et al., 1996*; *Yan et al., 2020* and Appendix D). Next, we discretize this DVI using the linearized implicit Euler timestepping scheme with $\Delta t = h$ at timestep $k$:

$$\frac{1}{h}(\mathcal{C}^{k+1} - \mathcal{C}^k) = \mathcal{G}^k \mathcal{U}^k, \quad \mathcal{U}^k = \mathcal{M}^k\left(\mathcal{F}_u^k + \mathcal{F}_b^k + \mathcal{F}_{nc}^k\right), \tag{7a}$$

$$\mathcal{F}_u^k = \mathcal{D}_u^k \boldsymbol{\gamma}_u^k, \quad \mathcal{F}_b^k = \mathcal{D}_b^k \boldsymbol{\gamma}_b^k, \tag{7b}$$

$$0 \leq \boldsymbol{\Phi}_u^{k+1} \perp \boldsymbol{\gamma}_u^k \geq 0, \tag{7c}$$

$$\mathcal{K}^k\left[\boldsymbol{\Phi}_b^{k+1} - \boldsymbol{\Phi}_b^0\right] = -\boldsymbol{\gamma}_b^k. \tag{7d}$$

The unknowns to be solved for at every timestep are the constraint (collision and motor tether) force magnitude $\boldsymbol{\gamma}_u^k, \boldsymbol{\gamma}_b^k$. This is a nonlinear DVI because $\boldsymbol{\Phi}_u^{k+1}, \boldsymbol{\Phi}_b^{k+1}$ are nonlinear functions of geometry $\mathcal{C}^{k+1}$, although $\mathcal{C}^{k+1}$ is linearly dependent on $\boldsymbol{\gamma}_u^k$ and $\boldsymbol{\gamma}_b^k$. For a small timestep ($h \to 0$), this nonlinearity can be linearized by Taylor expansion, for example, $\boldsymbol{\Phi}_u^{k+1} = \boldsymbol{\Phi}_u^k + h \nabla_{\mathcal{C}} \boldsymbol{\Phi}_u \mathcal{G}^k \mathcal{U}^k$. Then, this nonlinear DVI can be converted to a convex quadratic programming problem (*Nocedal and Wright, 2006*) (details in Appendix D):

$$\min_{\boldsymbol{\gamma}} f(\boldsymbol{\gamma}^k) = \frac{1}{2}\boldsymbol{\gamma}^{k,T} \boldsymbol{M}^k \boldsymbol{\gamma}^k + \boldsymbol{q}^{k,T} \boldsymbol{\gamma}, \tag{8a}$$

$$\text{subject to } \left[\boldsymbol{I}^{N_u \times N_u} \quad \boldsymbol{0}\right] \boldsymbol{\gamma}^k \geq 0. \tag{8b}$$

Here, $\boldsymbol{\gamma}^k = [\boldsymbol{\gamma}_u^k, \boldsymbol{\gamma}_b^k] \in \mathbb{R}^{N_u + N_b}$ is a column vector, and

$$\boldsymbol{M}^k = \begin{bmatrix} \mathcal{D}_u^{k,T} \\ \mathcal{D}_b^{k,T} \end{bmatrix} \mathcal{M}^k \begin{bmatrix} \mathcal{D}_u^k & \mathcal{D}_b^k \end{bmatrix} + \begin{bmatrix} 0 & 0 \\ 0 & \frac{1}{h}\mathcal{K}^{k,-1} \end{bmatrix}, \quad \boldsymbol{q} = \begin{bmatrix} \frac{1}{h}\boldsymbol{\Phi}_u^k + \mathcal{D}_u^{k,T}\mathcal{M}^k\mathcal{F}_{nc}^k \\ \frac{1}{h}\left(\boldsymbol{\Phi}_b^k - \boldsymbol{\Phi}_b^0\right) + \mathcal{D}_b^{k,T}\mathcal{M}^k\mathcal{F}_{nc}^k \end{bmatrix}. \tag{9}$$

One way to understand the constraint optimization method is that the implicit temporal integration 'jumps' on a timescale that bypasses the relaxation timescales of unilateral and bilateral constraints (collisions and crosslinking motor springs). In the limit of motor tethers being infinitely stiff ($\mathcal{K}^{-1} \rightarrow \mathbf{0}$), the quadratic term coefficient matrix $\mathbf{M}$ is still symmetric-positive-semi-definite (SPSD) and the *Equation 8* is still convex and can be efficiently solved. Physically speaking, in this case the bilateral constraints degenerate from deformable springs to non-compliant joints.

## Instantiation in a massively parallel computing environment

Our methods naturally lend themselves to high-performance parallel computing architectures. We utilize both MPI and OpenMP and use standard spatial domain decomposition to balance the number of motors and filaments across MPI processors. The motor update step samples the vicinity of every motor, where we use a parallel near-neighbor detection algorithm and update all motors in parallel. The most expensive part of the method is finding the solution to *Equation 8*, because of its very large dimension, equal to the total number of close pairs of filaments plus the number of crosslinking proteins. We use a fully parallel Barzilai-Borwein Projected Gradient Descent (BBPGD) solver (*Yan et al., 2019*) because the gradient $\nabla f = \mathbf{M}\gamma + \mathbf{q}$ is efficiently computed by one parallel sparse matrix-vector multiplication operation.

*aLENS* is written in a modular design using standard object-oriented C ++ and is available on GitHub as discussed at the end of the Discussion section.

## Verification and benchmarks

To validate and benchmark *aLENS*, we first note that its collision handling approach has already been benchmarked for the pure-filament phase, and shown to accurately reproduce the equation of state and the isotropic-nematic liquid crystal phase transition of densely packed rigid Brownian rods (*Yan et al., 2019*). This capacity to accurately compute the dense packing phase of fibers makes *aLENS* valuable to simulate many dense biological filament assemblies. The accurate treatment of steric interactions extends beyond other simulation methods and software, where steric interactions are often approximated by soft repulsive potentials or neglected.

We now further benchmark of *aLENS* by simulating mixtures of filaments and motors and directly comparing simulation results with experimental data. Although there are many parameters in our motor model, these comparisons don't involve fitting of model parameters to experimental data. Instead, we chose motor parameters as measured from experimental data (*Scharrel et al., 2014*; *Fürthauer et al., 2019*) or estimate them based on similar motor proteins (*Cross and McAinsh, 2014*).

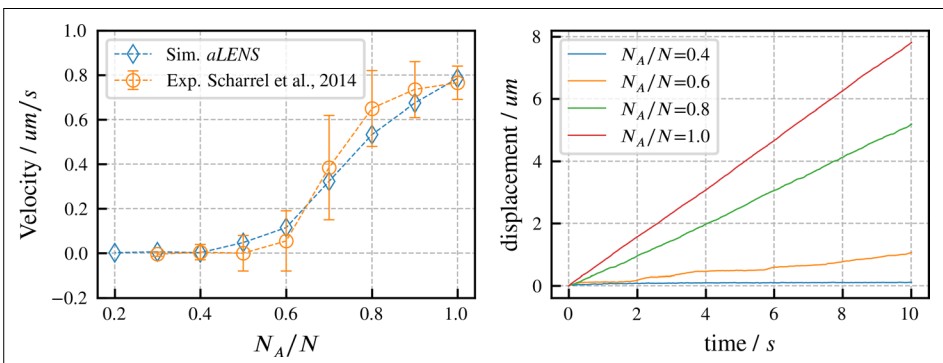

**Figure 2.** Directed transport velocity and displacement of microtubules driven by mixed active and inactive Kinesin-1 motors. The total number of active and inactive motors is fixed at $N = 100$ for all simulations. $N_A$ is the number of active motors. Left panel: comparison of microtubule velocities as a function of $N_A/N$ from *aLENS* simulations (blue diamonds) with from the reference experiment (orange circles) (*Scharrel et al., 2014*). Right panel: displacement vs. time of the transported microtubule obtained from simulation for several values of $N_A/N$. The free walking velocity of active motors was set to $1.0\,\mu m\,s^{-1}$ to match the experimental sliding velocity at $N_A/N = 1$. There are no other fitting parameters. All motor parameters are estimates based on experiments on Kinesin-1 (*Scharrel et al., 2014*) or similar motor proteins (*Cross and McAinsh, 2014*).

## Directed transport of microtubules by mixed active and inactive motors

We begin by verifying our motor model by reproducing results from experiments on directed microtubule transport (*Scharrel et al., 2014*). As in the experimental system, the simulation begins with a fixed number of motors with one head attached to a fixed surface while the other head interacts with one microtubule. Some motor heads are active and can drive gliding of the microtubule, while other heads are inactive and behave as passive crosslinkers that hinder microtubule motion. Here, $N_A$ is the number of active motors and $N$ is the total number of motors (active and inactive). The microtubule velocity increases as $N_A/N$ increases from 0 to 1 in experiments (*Scharrel et al., 2014*) and in our simulations. As shown in *Figure 2*, our simulations quantitatively reproduce the experimental data. To achieve this agreement, we set the active motor velocity to $1.0\,\mu m\,s^{-1}$, so the sliding velocity at $N_A/N = 1$ matches experiment. Apart from this one experimentally constrained velocity, there are no fitting parameters in our simulation (further motor parameters are in Appendix B). In initial trial simulations, we found that changing the total motor number $N$ didn't noticeably affect the microtubule transport velocity. Therefore, for the results shown here we fixed $N = 100$, similar to the experimental system. Since the transport trajectory is stable without stochastic noise, as shown in *Figure 2*, there is no need to perform ensemble average to determine the transport velocity. Therefore, we ran 1 simulation for 10 s for each ratio $N_A/N$.

## Self-straining state of actively crosslinked microtubule networks

As an additional verification, we compare *aLENS* with results of recent experiments of *Fürthauer et al., 2019* in which many-microtubule assemblies are densely packed into a nematic bundle and crosslinked by a large number of motors. In this heavily crosslinked nematic regime, microtubules are found to be transported by motors along the nematic director direction at a constant velocity in a direction determined by individual microtubule polarity. Experimentally, microtubule velocity was found to be independent of the local average polarity of the ensemble, as has been observed in extract spindles (*Needleman et al., 2010*), and (over the range of experimental conditions) independent of motor density. This phenomenon of oppositely oriented, constant velocity microtubule fluxes was referred to as 'self-straining motion', with the system interpreted as being composed to two polar microtubule gels whose inter-connecting motors pulled them past one another.

We simulate this experiment using 3,000 model microtubules with $L = 0.5\,\mu m$. Initially the filaments are confined in a tube of diameter $D = 1\,\mu m$, randomly initialized with their orientations along the $+x$ (pink) and $-x$ (white) directions, and packed at about 30% volume fraction. The simulated system is periodic along the $x$ direction, with periodic tube length $3\,\mu m$. There are approximately 25 motors per microtubule according to the experimental estimates, and in our simulations we vary the motor-to-microtubule number $N_m$ from 10 to 30. There is no accurate measurement for

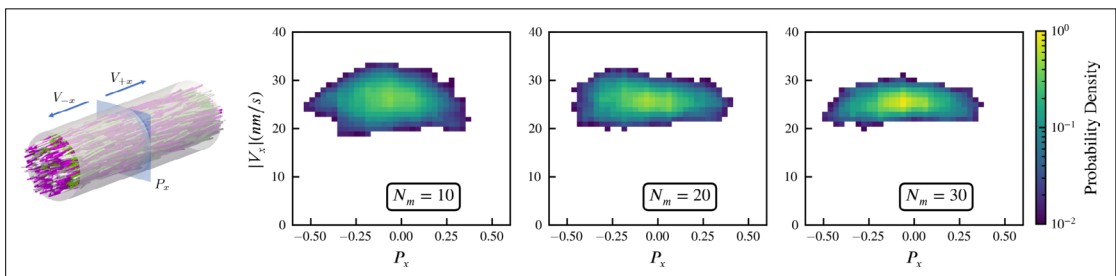

**Figure 3.** Sampled microtubule straining motion velocity vs local polarity in actively crosslinked microtubule network. The left panel shows the simulation geometry and the sampling procedure. Microtubules are randomly initialized with orientations along the $+x$ (pink) or $-x$ (white) directions. XCTK2 motors are colored green. $N_m$ is the number of XCTK2 motors per microtubule. We sample the local average polarity and straining velocity by inserting planes orthogonal to the $x$-axis into the collected data, matching the photobleaching technique used in experimental measurement (*Fürthauer et al., 2019*). For every sampling plane (e.g. the blue pane in the snapshot), we choose five sample points symmetrically on this plane and draw a square sampling window with edge length $0.2\,\mu m$ around each sample point. For each sampling window, we compute the average polarity $P_x$ along the $x$-axis for all microtubules intersecting this sampling window at a given time. We then compute the velocities, averaged over $10\,s$ (a duration chosen to match the experimental timescale), of microtubules intersecting each sampling window and moving along the $+x$ and $-x$ directions. $V_{+x}$ and $V_{-x}$ are computed from those two groups for each sampling window. The straining velocity is computed as $V_x = V_{+x} - V_{-x}$. Therefore, for every sampling window at each sampling timestep we have a pair of data values $P_x, V_x$. The right three panels show the joint probability distribution of $(P_x, V_x)$ computed from 900,000 sampling planes for each simulation, for $N_m = 10, 20, 30$, respectively.

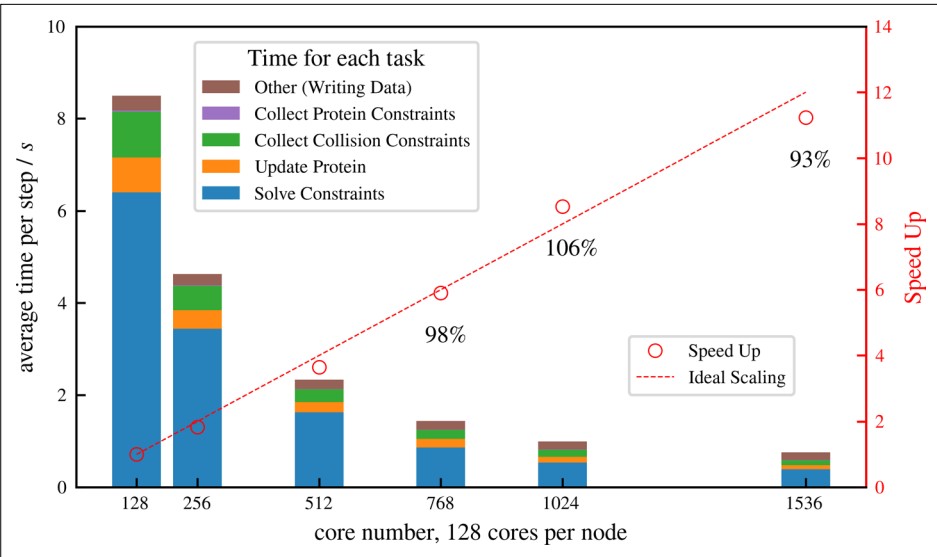

**Figure 4.** Strong scaling (fixed system size while increasing number of cores) efficiency of a system similar to but more than 10 times larger than that shown in **Figure 6**, comprising 1 million microtubules and 3 million motors. There are in total approximately 8 million constraints per time step, which is changing from step to step because collision pairs are changing and crosslinkers are stochastically binding and unbinding. The simulation is run for 100 computing steps with 1 data-saving step and the average per-step wall-clock time is shown in the figure.

the XCTK2 motor in these experimental conditions. Therefore, we used experimental estimates of $46\,\mathrm{nm\,s^{-1}}$ for the walking speed of NCD motors (**Furuta and Toyoshima, 2008**). To approximate the experimental measurement of velocity that used line photobleaching (**Fürthauer et al., 2019**), we sample the local polarity and straining velocity using virtual sampling planes, as shown in the left panel of **Figure 3**. As in **Fürthauer et al., 2019**, **Figure 3** shows that the straining velocity $V_x$ is largely independent of the number of motors $N_m$ and the local average polarity $P_x$ over the range simulated.

Intuitively, the straining velocity $V_x$ is predominantly determined by the free walking velocity of the motors in limit of many cross-linkers. From our simulations, we find a straining velocity of approximately $26\,\mathrm{nm\,s^{-1}}$, close to the experimental measurement of $18.6 \pm 0.9$.

## Large-scale parallelization efficiency

Simulation of cellular-scale cytoskeletal assemblies requires methods that can reach large system sizes and timescales. Therefore, we developed *aLENS* to efficiently utilize modern high-performance computing resources. Millions of objects and constraints can be simulated with *aLENS*. **Figure 4** shows detailed parallel efficiency measurements for one large-scale test case, similar to that in Figure 6, but more than 10 times larger. Here we track 1 million microtubules and 3 million motors for 100 timesteps. The performance is benchmarked on a cluster interconnected with infiniband and each node has two AMD EPYC 7742 CPUs, each having 64 cores at 2.5 GHz. We launched hybrid MPI + OpenMP jobs such that each MPI rank has 16 OpenMP threads. On average at each timestep the constraint optimization solver handles approximately 8 million collision and doubly bound motor constraints. The number of constraints changes at every timestep due to a variable number of collision pairs and to stochastic binding and unbinding of motors.

We achieve nearly ideal linear speed up as the number of cores increases (**Figure 4**). At 1536 cores, the efficiency remains at 93% and each timestep takes less than 1 s, making it possible to track such large systems on experimental timescales (a few seconds) within days or weeks of computing time. More importantly, the constraint optimization allows a $\Delta t$ that is one or two orders of magnitude larger than conventional pairwise potential methods. For the system simulated in **Figure 4**, *aLENS* can reach $1\,\mathrm{s}$ physical time per day, using a timestep size of 1.0.

## Results

Here, we illustrate the ability to use *aLENS* to study the interplay between microscopic dynamics and macroscopic order in active cytoskeletal assemblies. The specific examples shown here are the formation and extension of a band of microtubule bundles, polarity sorting of short microtubules on a spherical shell, the development of asters with and without thermal fluctuations, and the effect of confinement on assembling microtubule-motor mixtures. For the results presented here, all simulations were conducted in solvent with viscosity $\eta = 0.01 \, \mathrm{pN \, s} \, \mu m^{-2}$ at room temperature, using a fixed timestep $\Delta t = 10^{-4} \, \mathrm{s}$ unless otherwise stated.

### Bundle formation and buckling in a filament band

Microtubules driven by crosslinking motors can bundle; sliding of microtubules within the bundles causes them to fracture dynamically (*Sanchez et al., 2012*; *Foster et al., 2015*; *Roostalu et al., 2018*). We study such phenomena through a large-scale simulation of 100,000 filaments modeling microtubules and 500,000 minus-end-directed motor proteins modeled after dynein (*Figure 5*). Motor crosslinking drives contraction of initially disordered, bundled filaments (*Figure 5A and B*). Aligning steric and crosslinking forces drive the system into a series of well-aligned bundles spanning several filament lengths (*Figure 5C*, see *Figure 5—video 1* and *Figure 5—video 2*). The motors slide filaments parallel to each other, generating macroscopic extensile motion. Later, the extended network buckles and fractures (*Figure 5C*).

The macroscopic stresses and dynamics depend on the spatial organization of filaments and motor-driven sliding. To characterize this, we measure the joint probability distribution of the local nematic order parameter $S_{\mathrm{local}}$ and the number $N_d$ of neighboring filaments crosslinked to a filament (*Figure 5D*). While the network contracts, the distribution of $N_d$ does not change significantly because the number of motors per filament and the maximum number of neighboring filaments within a densely packed structure remain roughly constant. As filaments align, they become near-perfectly nematic ($S_{\mathrm{local}} \approx 1$), although less-ordered regions occur between aligned bundles of different orientations (*Figure 5C1 and D2*).

Inside the bundles, filament sliding by motors leads to transport along the local nematic director. Projecting filament trajectories onto the lab-frame $x$-axis, we observe left- and right-moving filaments that speed up early in the simulation, and then maintain constant average velocities at later time ($t > 4 \, \mathrm{s}$ in *Figure 5E*), as filaments align due to steric and motor forces (*Figure 5F*). Note that velocity and stresses plateau only when the nematic order saturates.

The filament motions created by motors cause the densely-packed filaments to collide often, creating a net extensile stress along the bundles' axes (*Figure 5F*). However, the fixed simulation box size hinders the networks' elongation, causing the bundles aligned with $x$-axis to buckle due to the net extensile stress (*Figure 5F*, see *Figure 5—video 1* and *Figure 5—video 2*). In contrast, bundles not aligned with the $x$-axis are not constrained and so evolve into straight spikes. This misalignment of bundles is seen as a small net stress in the $y, z$-directions for $t \geq 4 \, \mathrm{s}$ (*Figure 5F*).

### Polarity sorting in a spherical shell

Crosslinking motors on antiparallel filaments drive polarity sorting, which transports filaments to regions of like polarity. This has been well-studied on a planar periodic geometry, e.g. (*Gao et al., 2015b*). Here, we use *aLENS* to examine the effect of confinement geometry on polarity sorting (*Figure 6*). The geometry is designed to explore the polarity sorting phenomena where initial filament alignment occurs in a spherical geometry and significantly affects the dynamics and steady state of the system. In this simulation, 100,000 filaments with aspect ratio $L/D_{\mathrm{fil}} = 10$ are confined between two closely spaced concentric spherical shells at 40% volume fraction. The shell gap is $\Delta R = 0.102 \, \mu m$, shorter than the filament length, with $\Delta R/D_{\mathrm{fil}} \approx 4$ so filaments can move over each other in a restricted way. The filaments are initialized such that the nematic directors are along the meridians everywhere. 200,000 motors, modeled after kinesin-5 tetramers, drive relative filament sliding (*Figure 6A*). Brownian motion is modeled at room temperature $300 \, \mathrm{K}$ and timestep $\Delta t$ is set to 1. Motors move toward minus ends of bound filaments at $v_m = 1.0 \, \mu m \, s^{-1}$. Once they reach the minus ends, they immediately detach.

Motors walk along the filaments, driving sliding of antiparallel filaments (*Figure 6B*). This leads to polarity-sorted regions at the north and south 'poles' of the sphere, meaning that the filament

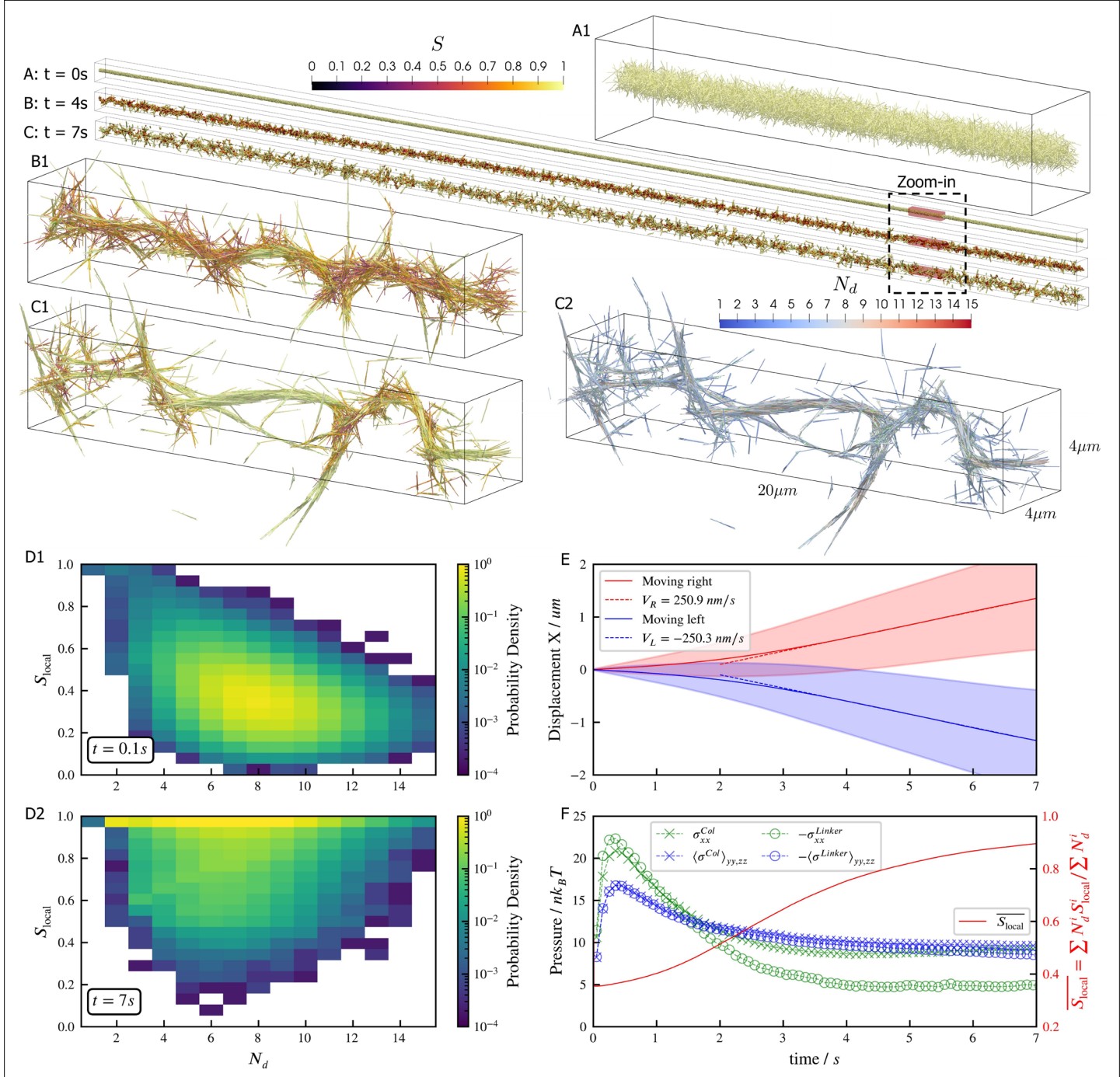

**Figure 5.** Results for the bundling-buckling simulation of 100,000 microtubules and 500,000 dynein motors in the periodic simulation box of $600 \times 10 \times 10 \, \mu m$. Brownian motion of microtubules is turned off. Each dynein has one non-motile head permanently attached to a microtubule and the other motile head walks processively with maximum velocity $1 \, \mu m \, s^{-1}$. If bound, the motile head moves toward the microtubule minus-end, and detaches upon reaching it. Detailed parameters for this motor are tabulated in the Appendix 1. Every microtubule has 5 dynein motors permanently attached to randomly chosen, fixed locations along the length. The initial configuration of microtubules is randomly generated, with their orientations sampled from an isotropic distribution and centers uniformly distributed within a cylinder of length $600 \, \mu m$ and diameter $0.3 \, \mu m$. The motile heads of all dynein motors are unbound initially. (**A, B, C**) The bundle at $t = 0 \, s$, $4 \, s$, and $7 \, s$. Microtubules are colored by their local nematic order parameter $S_{local} = \sqrt{\frac{3}{2} Q_{ij} Q_{ij}}$, with $Q_{ij} = \langle p_i p_j \rangle - \frac{1}{3} \delta_{ij}$, $\boldsymbol{p}$ being the unit orientation vector of each microtubule pointing from the minus to the plus end, and $\boldsymbol{\delta}$ the Kronecker delta tensor. The average $\langle . \rangle$ is taken over each microtubule plus all microtubules that are directly crosslinked to it by dynein motors. (**A1, B1, and C1**) Zoom-in views of the small region marked by red box in A, B, and C. (**C2**) The same region in C1 but colored by $N_d$, the number of microtubules averaged over when computing $S_{local}$. (**D1 and D2**) The joint probability distributions $S_{local}$ and $N_d$ for each microtubule for the entire

*Figure 5 continued on next page*

*Figure 5 continued*

systems at $t = 0.1$ s, when the dyneins crosslink microtubules but microtubules barely move from initial configuration, and at $t = 7$ s, when the bundle is nematic. (**E**) The average trajectories (solid lines) and their standard deviation (shaded area) of left-moving and right-moving microtubules. Dashed lines show linear fits to the average trajectory after $t = 4$ s, with results $V_R \approx V_L \approx 250 \, \text{nm s}^{-1}$. (**F**) The normal stresses and the weighted average $\overline{S_{\text{local}}}$ over time. Due to the symmetry in the $y, z$ directions, only their average is shown $\langle \sigma \rangle_{yy,zz} = \frac{1}{2} \left( \sigma_{yy} + \sigma_{zz} \right)$. Collision stress is positive (extensile) and crosslinker stress is negative (contractile). The weighted average $\overline{S_{\text{local}}} = \sum N_d^i S_{\text{local}}^i / \sum N_d^i$.

The online version of this article includes the following video for figure 5:

**Figure 5—video 1.** Contraction and buckling of a long microtubule-motor bundle.

https://elifesciences.org/articles/74160/figures#fig5video1

**Figure 5—video 2.** Motor motion and stretching during the contraction and buckling of a long microtubule-motor bundle.

https://elifesciences.org/articles/74160/figures#fig5video2

orientation $p$ on average points toward the poles. Filaments with reversed initial polarity are transported to the equatorial region (*Figure 6C1*). In contrast to the planar geometry (*Gao et al., 2015b*), we did not observe the formation of polar lanes with boundaries between polarity-sorted regions approximately parallel to the polarity direction. Instead, on the sphere the boundaries between polarity-sorted regions are approximately orthogonal to the polarity directions, as more clearly illustrated by plotting the polarity divergence (*Figure 6C1*).

Motors also accumulate in some regions according to the filament polarity (*Figure 6C1*). These motor accumulation regions are actually regions where the divergence of filament polarity field is positive, meaning areas of overlap of filament minus-ends (*Figure 6C2, C5 and G*). This accumulation is illustrated by the positive correlation between motor density $n$ and $\nabla \cdot p$ at $t = 4$ s in *Figure 6D*. Furthermore, motor accumulation regions appear to show slightly lower filament volume fraction (*Figure 6C2 and C4*), as shown in *Figure 6F*. These correlations can be understood through the behavior of crosslinking motors near filament ends (*Figure 6G*). Once polarity sorted regions of filaments form, as the blue arrows represent, $\nabla \cdot p > 0$ in regions where minus-ends meet minus-ends and vice versa in regions where plus-ends meet plus-ends. Minus-end directed motors accumulate in regions with $\nabla \cdot p > 0$, while plus-end motors accumulate in regions with $\nabla \cdot p < 0$. Once motors accumulate, they may attach to both minus ends and push them away such that the distance between minus ends is the length of motors. As a result, the volume fraction of filaments in that region is below average.

In contrast, if the motors stop walking but do not detach when they reach the minus ends (end-pausing, EP), the filament network contracts (Fig. 6E1-5) with volume fraction increases from 40% to 60% and eventually freezes at $t = 0.27$ s. We observe neither substantial polarity sorting nor motor accumulation. This indicates that the ability of motors to continuously walk, without end-pausing, is crucial to effective polarity sorting.

## Aster formation in bulk

Aster formation is driven by motor pausing at ends of rigid filaments (end-pausing). Previous work has focused on how motor biophysics affects aster formation (*Belmonte et al., 2017*; *Roostalu et al., 2018*). An additional contributor to aster formation may be thermal fluctuations, which are difficult to tune experimentally but can be easily modulated in simulations (*Figure 7*). To examine this, we simulated 40,000 filaments and 80,000 processive, minus-end-directed, end-pausing motors starting from the same spatially uniform and orientationally isotropic random configuration (*Figure 7A*). In one version of the model, we included thermal fluctuations that drive filament motion (*Figure 7D* and movie *Figure 7—video 1*), while in the other thermal fluctuations of filaments were neglected (*Figure 7E* and movie *Figure 7—video 2*). The resulting structure of the system is significantly different in the absence of filament thermal motion, showing that thermal fluctuations influence the asters' shape, structure, and ultimate spatial organization. With filament thermal motion, a number of dispersed, spherically symmetric, dense asters form. By contrast, in the absence of thermal motion the number of asters is larger and more regularly spaced, but their shape is more irregular and they contain fewer filaments (*Figure 7D* vs E).

These differences are clear in the radial distribution function of filament minus ends, which are clustered by motors paused at filament ends (*Figure 7B*). On large length scales, the radial distribution reflects larger and denser asters for the simulation with thermal fluctuations that drive filament

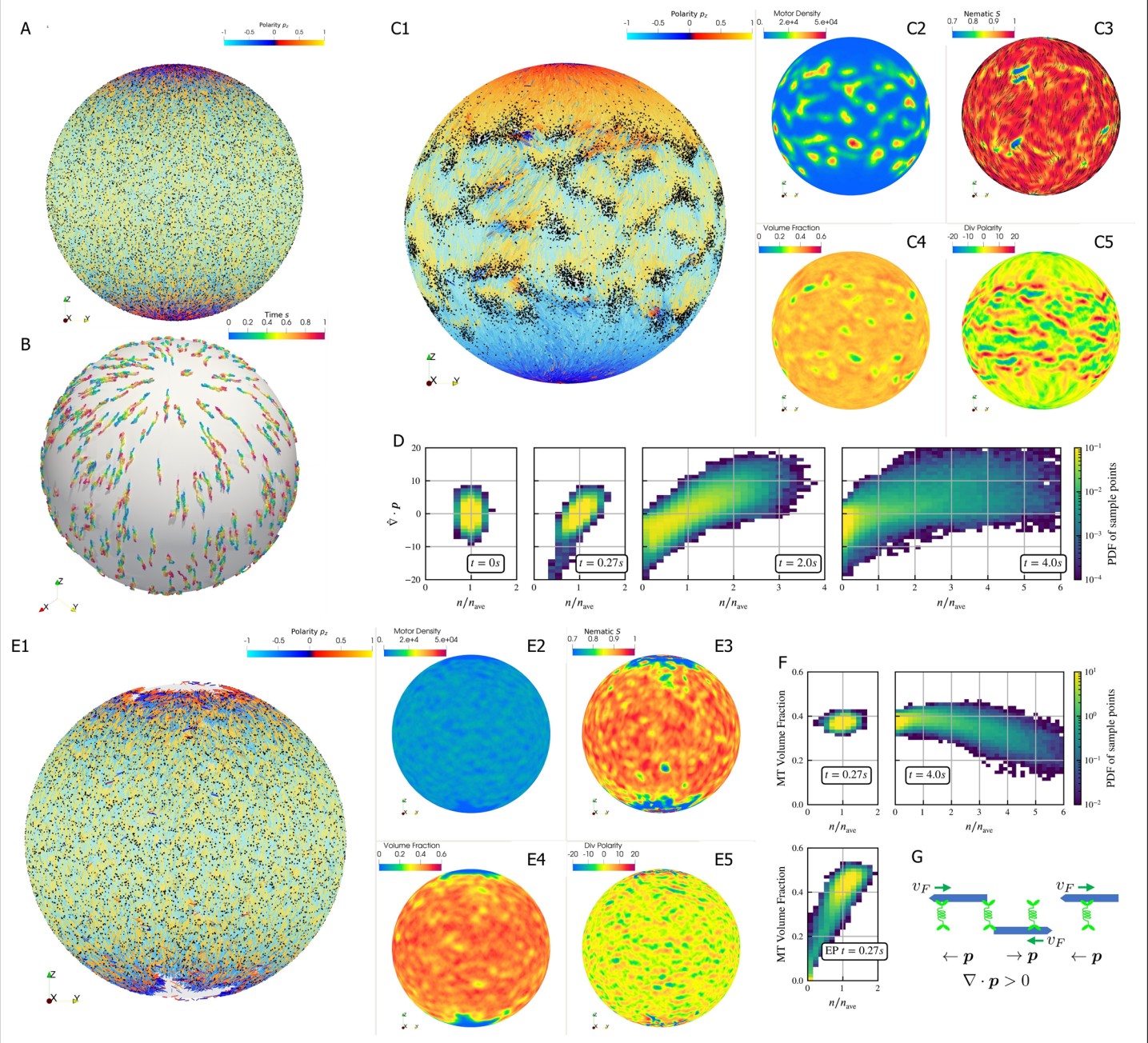

**Figure 6.** Results for the polarity sorting simulation in a spherical shell. Initially, 100,000 0.25 μm-long filaments modeling microtubules and 200,000 motors modeling crosslinking kinesin-like proteins are placed between two concentric spherical shells with radii $r_{\text{in}} = 5\,\mu m$ and $r_{\text{out}} = 5.102\,\mu m$, to maintain the volume fraction of filaments between these two shells at 40%. Initially, all filaments are evenly distributed on the spherical shell, with their orientation randomly chosen to be either $\pm e_\theta$ at each point, where $e_\theta$ is the polar basis norm vector of spherical coordinate system. The pure filament system is relaxed for $1\,s$ to resolve the overlaps in the initial configuration. Afterwards at $t = 0$, 200,000 motors are added to the system homogeneously distributed between the two shells. Sample points are evenly placed to measure the statistics by averaging the volume within $0.25\,\mu m$ from each sample point. (**A**) The configuration at $t = 0$. Filaments are colored by their polarity, while motors are colored as black dots. Only randomly selected 10% of all motors (same after) are shown in the image to illustrate the distribution. (**B**) Randomly selected trajectories of filaments from $t = 0\,s$ to $1\,s$. Trajectories are colored by time. It is clear that filaments move along the meridians. (**C1-C5**) Configuration and statistics at $t = 4\,s$. (**C1**) The filaments and motors. Motors clearly concentrate in some areas. (**C2**) The motor number density, that is, number of motors per $1\,\mu m^3$. (**C3**) The nematic director field (shown as black bars) and the nematic order parameter $S$. (**C4**) The filament volume fraction. (**C5**) The divergence of polarity field $\nabla \cdot p$ non-dimensionalized by filament length, that is, change of mean polarity per filament length. (**D**) The development of the correlation between motor number density $n/n_{\text{ave}}$ and the polarity divergence field, at different times of the simulation. Clearly high $n/n_{\text{ave}}$ are correlated with positive polarity divergence. (**E1-E5**) Configuration and statistics at $t = 0.27\,s$ for a comparative simulation where motors have end-pausing, arranges in the same style as C1-C5.

*Figure 6 continued on next page*

*Figure 6 continued*

This case shows significant contraction instead of polarity sorting as filaments are pulled away from the north and south poles and the overall volume fraction significantly increases to approximately 60%. The structure becomes densely packed and does not significantly evolve further. (**F**) The correlation between motor number density $n/n_{ave}$ and the local filament volume fraction. For the polarity sorting case at $t = 4\,s$ the motor number density correlates with low filament volume fraction. This is not seen in the end pausing (EP) case. (**G**) A schematic for the correlations shown in D and F.

movement. In simulations of both cases, two prominent peaks appear in the radial distribution funcation at small length scales $r = 25\,nm = D_{fil}$ and $r = 78\,nm = \ell_0 + D_{fil}$ which correspond to scale on which filaments bind to or are crosslinked by motors, respectively (*Figure 7B, D2 and E2*). The relatively small peak between these two maxima correspond to filaments that are geometrically confined between two crosslinked filaments.

These differences arise from the fact that athermal filaments do not move unless driven by motors, which requires that two filaments are close enough to become crosslinked. This suggests that, at steady state, athermal aster centers are separated by twice the filament length. In contrast, with thermal motion filaments may diffuse $\sim 1\,\mu m$ in $1\,s$. This allows filaments to diffuse until they are captured in regions of high motor density, such as aster centers. Furthermore, with thermal fluctuations the asters themselves diffuse, which leads to aster coalescence (*Figure 7D1*). These observations and estimated lengthscale are quantitatively confirmed by analyzing the static structure factor of aster centers (details in Appendix F), which shows that the athermal simulation has approximately three times more asters than the thermal case (*Figure 7D* vs E).

The differences in the dynamics of aster formation are also reflected in stress measurements (*Figure 7C*), where the more crowded filament configurations of the thermal case produces a larger stress throughout the simulation. In both cases, the motor-induced stress $\Pi^{Linker}$ initially increases quickly, reaching a peak at roughly $t = 4\,s \sim 5\,s$, similar to the behavior during bundle contraction shown above (*Figure 5F*), before declining. The average time required for motors to walk to filament ends, $\tau_{walk} = L/v_m \approx 5\,s$, determines the initial contraction timescale. After reaching minus ends, motors pause and relax toward their equilibrium lengths. As a result, both the motor and collision stress grow in magnitude as more motors accumulate at minus ends.

## Confined filament-motor protein assemblies

Confinement of cytoskeletal structures plays an important role in cells, where the cytoskeleton is spatially constrained by membranes, organelles, and other cellular structures. Although in the previous examples we studied open periodic geometry, here we show results of cylindrical confinement. The microtubule motor system is constrained inside a cylinder with periodic boundary conditions at the cylinder ends. The impermeable boundary of the cylinder surface to motors and filaments was implemented by our complementarity constraints.

Similar to the previous bulk cases, motors move filaments to create high-density crosslinked filament aggregates that coexist with a relatively low density vapor of non-crosslinked filaments. In bulk systems as shown above and in previous work, end-pausing motors drive aster formation because crosslinking motors pull filament ends together. A confining cylindrical boundary strongly modifies the conformation of these aggregated structures (*Figure 8*). These simulations used $0.25\,\mu m$ long filaments at a fixed packing fraction ($\phi = 0.16$), confined in two cylinders with diameters $D_{cyl} = 0.25\,\mu m$ and $0.75\,\mu m$.

For a small-diameter cylinder where one filament length can fit across the cylinder ($D_{cyl}/L = 1$, $D_{cyl} = 0.25\,\mu m$), the cylinder is too narrow for asters to form. Instead, motor sliding and end-pausing drive the filaments into polarity-sorted bilayers (PSBs, *Figure 8A* and movie *Figure 8—video 1*). A single polarity-sorted bilayer contains a central interface of highly crosslinked filament minus-ends between two antiparallel polar layers of filaments (*Figure 8*). At steady state, the system consists of individual PSBs separated by low-density vapor regions containing few motors. As expected, the local nematic order parameter $S_{local}^x(x)$ nearly reaches 1 within PSBs. Even the the vapor phase is close to nematic $S_{local}^x \approx 0.6$ (*Figure 8A4*), due to the strong confinement effect.

Next we increased the diameter of the cylinder to $D_{cyl}/L = 3$ ($D_{cyl} = 0.75\,\mu m$) to weaken the confinement (*Figure 8B* and movie *Figure 8—video 2*). Here, the polarity-sorted bilayers are not present, because the larger cylinder diameter allows filaments to reorient and organize into bottle-brush-like aggregates (BBs). In the bottle brushes, filament plus ends are oriented radially outward from the

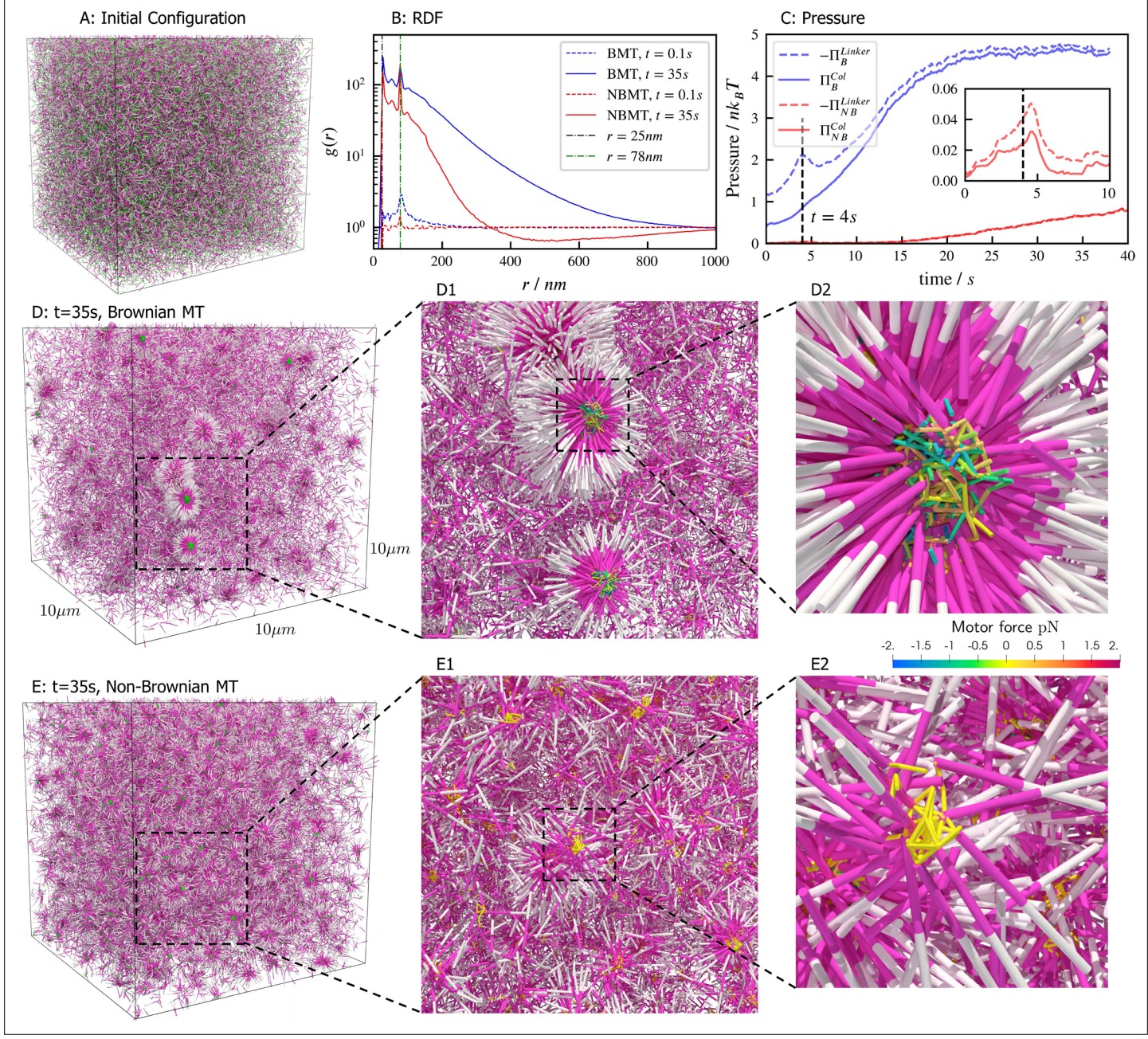

**Figure 7.** Results for the aster formation simulations with Brownian motion of simulated microtubules turned on (BMT) and off (NBMT). Initially, 40,000 0.5 $\mu m$-long filaments modeling microtubules and 80,000 motors modeling crosslinking kinesin-like proteins are placed in a periodic cubic box of $10 \times 10 \times 10 \, \mu m$ with uniform distribution. Filament orientations are isotropic and motors are all in the unbound state. Motors are assumed to have two minus-end-directed walking heads with symmetric properties. They are assumed to pause when they reach the minus end of filaments until detaching. Detailed parameters are tabulated in Appendix B. (**A, D, and E**) Simulation snapshots. Each filament is shown as a cylinder colored in half pink (minus end) and half white (plus end). (**A**) The initial configuration for both NMT and BNMT cases. Each motor is colored as a green dot. (**D and E**) The snapshot for both cases at $t = 35$ s. D1-2 and E1-2 Expanded views of a aster core for D and E. Only doubly bound motors are shown in D and E (in green color), and in D1-2 and E1-2 (colored by the spring force). Negative values mean the crosslink forces are contractile (attractive). (**B**) The radial distribution function (RDF) $g(r)$ for the minus ends of all filaments at $t = 0.1$ s (dashed lines) and $t = 35$ s (solid lines). The first peak of $g(r)$ at $r = 25$ nm corresponds to close contacts between filaments. The second peak of $g(r)$ at $r = 78$ nm$=25$ nm$+53$ nm corresponds to the minus ends of filaments crosslinked by motors whose rest length is 53 nm. Blue and red lines are results for the BMT and NBMT cases, respectively. (**C**) The collision (solid) and crosslinker (dashed) pressure for BMT (blue) and NBMT (red) cases. Pressure is defined as the trace of the stress tensor: $\Pi = \frac{1}{3}\mathrm{Tr}\boldsymbol{\sigma}$. The collision pressure $\Pi^{Col}$ is positive (extensile), and the motor pressure $\Pi^{Linker}$ is negative (contractile). The inset plot shows the pressure for the NBMT case in the initial stage of the simulation. The black dashed lines mark the time $t = 4$ s.

*Figure 7 continued on next page*

*Figure 7 continued*

The online version of this article includes the following video for figure 7:

**Figure 7—video 1.** Aster formation in bulk of Brownian microtubules.

https://elifesciences.org/articles/74160/figures#fig7video1

**Figure 7—video 2.** Aster formation in bulk of Non-Brownian microtubules.

https://elifesciences.org/articles/74160/figures#fig7video2

cylinder axis, forming a hedgehog line defect capped by half asters (*Figure 8B2*). Motors become highly concentrated along the line defects at the center of the cylinder (*Figure 8B3*). The radial hedgehog structure of BBs is evidenced by a negative local nematic order parameter (*Figure 8B4*, blue line). The splayed nature of the BBs produces a lower relative packing fraction of $\sim 2.5$ times the vapor when compared to the PSBs (*Figure 8B4*, red line).

## Discussion

We designed *aLENS* to (i) model crosslinking motor kinetics conforming to an underlying free energy landscape, (ii) circumvent the timescale limitation imposed by conventional explicit timestepping methods, and (iii) efficiently utilize modern parallel computing resources to allow simulation of cellular-scale systems. This efficient framework allows both modeling the individual building cytoskeletal building blocks (filaments, motors) and gathering mesoscale statistical information such as stress and order parameters from a large system. This multiscale capability will make it possible to directly compare simulations with experimental observations on mesoscopic and macroscopic scales over timescales from seconds to minutes.

The *aLENS* framework is not limited to a specific motor model. Because of the modular design of the motor code, the motor model can be extended to include additional physics such as force-dependent binding and unbinding rates, or even entirely replaced, say, with a passive crosslinker or other model. Dynamic instability and branching of cytoskeletal filaments can also be integrated with the constraint minimization problem, as we showed previously in modeling the division-driven growth of bacterial colonies (*Yan et al., 2019*). Long and flexible polymers can be simulated by chaining short and rigid segments together with flexible connections (Appendix H), even with nonlocal interactions mediated by hydrodynamics, electrostatics, or other fields (*Shelley, 2016*; *Nazockdast et al., 2017*; *Maxian et al., 2021*). For example, in ongoing work we have used *aLENS* to simulate chromatin in the nucleus as a bead-spring chain moving through the nucleoplasmic fluid, and confined by the nuclear envelope.

Recent years have seen considerable innovation in computational approaches to cytoskeletal modeling, implemented in powerful simulation packages including Cytosim (*Nedelec and Foethke, 2007*), MEDYAN (*Popov et al., 2016*), and AFINES (*Freedman et al., 2017*). These packages utilize a variety of coarse-grained representations of cytoskeletal elements and numerical simulation schemes, with the diversity of approaches in part reflecting the diversity of cytoskeletal systems and phenomena of interest. *aLENS* brings a powerful set of new capabilities to the table, significantly expanding the range of accessible time and length scales in simulations of systems in which excluded volume and crosslink-mediated interactions play an important role.

*aLENS* has been open-sourced on GitHub: https://github.com/flatironinstitute/aLENS (copy archived at swh:1:rev:f2dd484f82443735562ad7b480fe7ed9fc020fb0; *Adam, 2022*) and precompiled binary executable is available on DockerHub: https://hub.docker.com/r/wenyan4work/alens. Our GitHub documentation provides a clear roadmap for developing additional user-specific modules.

## Acknowledgements

MJS acknowledges support from NSF grants DMR-2004469 and CMMI-1762506. SA, ARL, MAG, and MB acknowledge support from NSF grants DMS-1821305, and NIH grant RGM124371A. This work utilized resources from the University of Colorado Boulder Research Computing Group, which is supported by the National Science Foundation (awards ACI-1532235 and ACI-1532236), the University of Colorado Boulder, and Colorado State University. We thank Prof. Dimitrios Vavylonis for discussions on implementing flexible filaments.

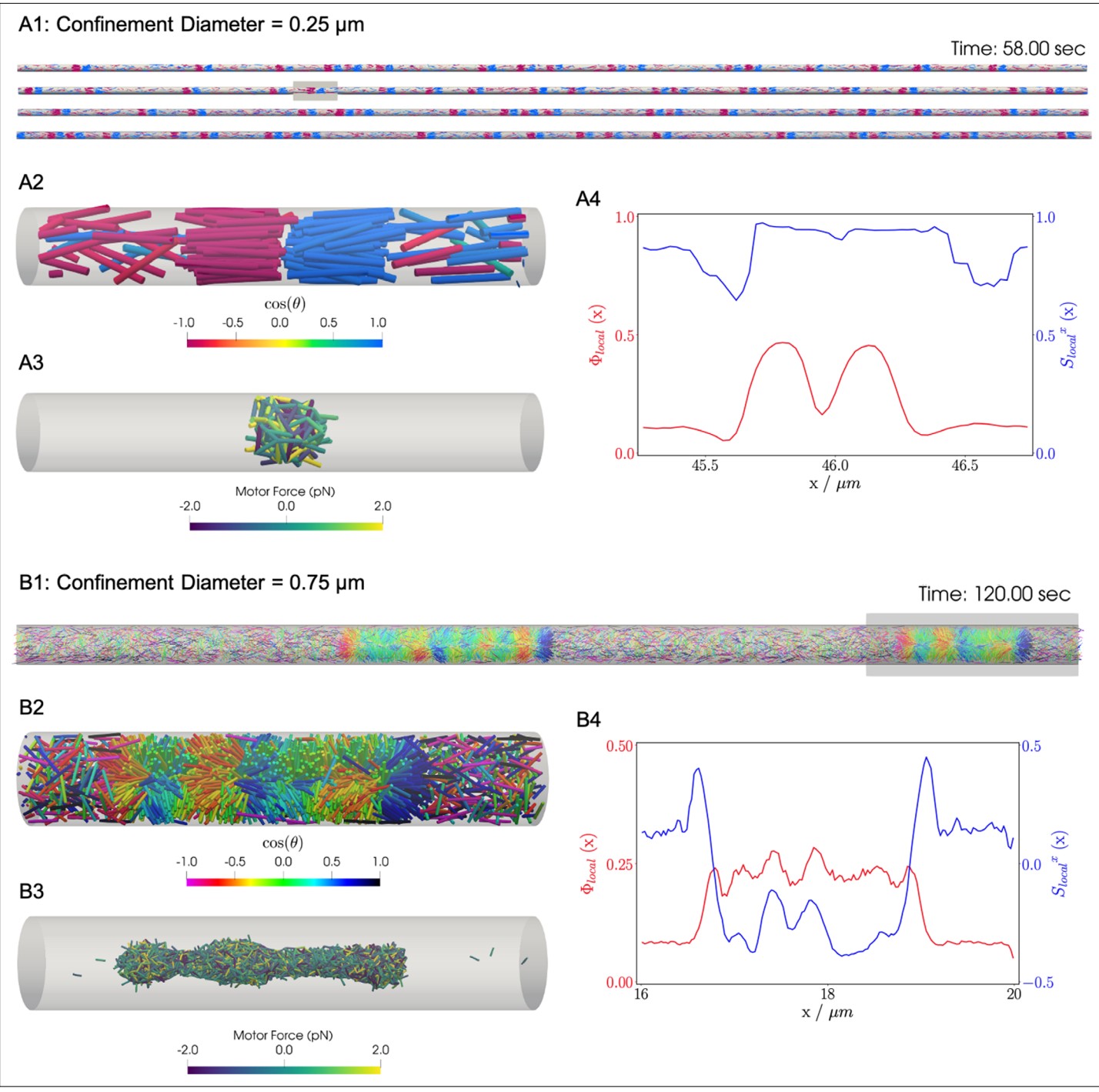

**Figure 8.** Results for the confined filament-motor protein assembly simulations with $9,216$ filaments modeling microtubules and $27,648$ motors modeling crosslinking kinesin-like proteins at a cylinder diameter of $D_{cyl} = 0.25\,\mu m$ and $0.75\,\mu m$. Initially, $0.25\,\mu m$ long filaments are uniformly distributed and aligned along the $x$-axis, with equal numbers oriented in the $+x$ and $-x$ directions. Crosslinking motor proteins are initially unbound and distributed uniformly as well. A and B: Snapshots of the simulation with $D_{cyl} = 0.25\,\mu m$ and $0.75\,\mu m$ at $t = 58\,s$ and $t = 120\,s$. A1 and B1: All 9,216 simulated filaments. In A1, the the cylinder is too long to be displayed contiguously, therefore a stacked representation is shown. The filaments are colored by the value of $\cos\theta$ where $\theta$ is the angle between the filament direction vector $\boldsymbol{p}$ (oriented from the minus-end to the plus-end) and the positive $x$-axis (pointing to the right). A2 and B2: Zoomed-in view of the filaments in the boxed regions in A1 and B1. A3 and B3: Doubly-bound motors in the boxed regions, colored by their binding force. Negative values represent contractile force while positive values indicate extensile force. A4 and B4: The local packing fraction (red line) and the local nematic order parameter (blue line), $S^x_{local}(x) = \sum_i^{N(x)} W_i(x) S^x_{local}(x)_i$ where a filament contributes $S^x_{local}(x)_i = \frac{1}{2}(3\cos^2\theta_i - 1)$ to the local order at $x$. Filament contributions are weighted by $W_i(x)$ and summed over all filaments at $x$.

*Figure 8 continued on next page*

*Figure 8 continued*

Line plots represent an average over $1\,\mathrm{s}$ for the snapshots in A2 and B2. Detailed parameters and calculations for the crosslinking motor proteins are presented in Appendix G.

The online version of this article includes the following video for figure 8:

**Figure 8—video 1.** Filament-motor assembly for the  case.
https://elifesciences.org/articles/74160/figures#fig8video1

**Figure 8—video 2.** Filament-motor assembly for the  case.
https://elifesciences.org/articles/74160/figures#fig8video2

# Additional information

### Funding

| Funder | Grant reference number | Author |
| --- | --- | --- |
| National Science Foundation | DMR-2004469 | Michael Shelley |
| National Science Foundation | CMMI-1762506 | Michael Shelley |
| National Science Foundation | DMS-1821305 | Saad Ansari<br>Matthew A Glaser<br>Meredith D Betterton |
| National Science Foundation | ACI-1532235 | Saad Ansari<br>Matthew A Glaser<br>Meredith D Betterton |
| National Science Foundation | ACI-1532236 | Saad Ansari<br>Matthew A Glaser<br>Meredith D Betterton |
| National Science Foundation | RGM124371A | Saad Ansari<br>Matthew A Glaser<br>Meredith D Betterton |

The funders had no role in study design, data collection and interpretation, or the decision to submit the work for publication.

### Author contributions

Wen Yan, Data curation, Formal analysis, Investigation, Methodology, Project administration, Resources, Software, Validation, Visualization, Writing - original draft, Writing - review and editing; Saad Ansari, Data curation, Investigation, Validation, Writing - original draft, Writing - review and editing; Adam Lamson, Formal analysis, Investigation, Methodology, Software, Validation, Writing - original draft, Writing - review and editing; Matthew A Glaser, Supervision, Writing - review and editing; Robert Blackwell, Data curation, Validation; Meredith D Betterton, Michael Shelley, Conceptualization, Funding acquisition, Investigation, Project administration, Resources, Supervision, Writing - review and editing

### Author ORCIDs

Wen Yan http://orcid.org/0000-0002-9189-0840
Adam Lamson http://orcid.org/0000-0003-2616-2801
Matthew A Glaser http://orcid.org/0000-0002-8366-5598
Meredith D Betterton http://orcid.org/0000-0002-5430-5518

### Decision letter and Author response

Decision letter https://doi.org/10.7554/eLife.74160.sa1
Author response https://doi.org/10.7554/eLife.74160.sa2

### Data availability

The current manuscript is a computational study, so no data have been generated for this manuscript. The open-source modeling software is hosted at GitHub.

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

# Appendix 1

## Crosslinker and motor properties

**Appendix 1—table 1.** The parameters of the two springs controlling extension and bending, respectively.

| Parameter | Explanation | Unit |
|---|---|---|
| End-pausing | True or False | ND |
| One head fixed | True or False | ND |
| $\lambda$ | energy factor | ND |
| $\lambda_{P,AP}$ | parallel to anti-parallel factor | ND |
| $\ell_0$ | free length | $\mu m$ |
| $r_c$ | capture radius | $\mu m$ |
| $\kappa$ | Hookean spring constant | $\mathrm{pN}\,\mu m^{-1}$ |
| $F_{stall}$ | stall force | pN |
| $d_U$ | unbound diffusivity | $\mu m^2\,\mathrm{s}^{-1}$ |
| $\epsilon$ | binding site density | $\mu m^{-1}$ |
| $v_m$ | max walking velocity | $\mu m\,\mathrm{s}^{-1}$ |
| $K_a$ | association constant ($U \rightleftharpoons S$) | $\left(\mu\mathrm{mol/L}^{-1}\right.$ |
| $k_{o,S}$ | off-rate constant ($U \rightleftharpoons S$) | $\mathrm{s}^{-1}$ |
| $K_e$ | effective association constant ($S \rightleftharpoons D$) | ND |
| $k_{o,D}$ | force-independent off-rate constant ($S \rightleftharpoons D$) | $\mathrm{s}^{-1}$ |
| $d_S$ | singly bound head diffusivity | $\mu m^2\,\mathrm{s}^{-1}$ |
| $d_D$ | doubly bound head diffusivity | $\mu m^2\,\mathrm{s}^{-1}$ |
| $v_S$ | singly bound walking velocity | $\mu m\,\mathrm{s}^{-1}$ |
| $x_c$ | force-dependent unbinding length | $\mu m$ |

**Appendix 1—table 2.** Properties of crosslinkers used in the main text.
ND means dimensionless. Parameters given as an array $[a, b]$ means the two values are used for each each of a crosslinker, respectively. Kinesin-5 parameters are adapted from *Blackwell et al., 2017*. Dynein parameters are adapted from *Foster et al., 2017*. Kinesin-1 parameters are adapted from *Scharrel et al., 2014*.

| Parameter | Kinesin-5 | Dynein | Kinesin-1 | Inactivated Kinesin-1 |
|---|---|---|---|---|
| End-pausing | True | False | False | False |
| One head fixed | False | True | True | True |
| $\lambda$ | 0.258 | 0.5 | 0.5 | 0.5 |
| $\lambda_{P,AP}$ | 1 | 1 | 1 | 1 |
| $\ell_0$ | 0.053 | 0.040 | 0.05 | 0.05 |
| $r_c$ | 0.039 | 0.033 | 0.038 | 0.038 |
| $\kappa$ | 300.0 | 100.0 | 100.0 | 100.0 |

*Appendix 1—table 2 Continued on next page*

*Appendix 1—table 2 Continued*

| Parameter | Kinesin-5 | Dynein | Kinesin-1 | Inactivated Kinesin-1 |
|---|---|---|---|---|
| $F_{stall}$ | 5.0 | 1.0 | 7.0 | 7.0 |
| $d_U$ | 1.0 | 1.0 | 1.0 | 1.0 |
| $d_S$ | 0 | 0 | $[0, 10^{-2}]$ | $[0, 0]$ |
| $d_D$ | 0 | 0 | $[0, 10^{-2}]$ | $[0, 0]$ |
| $\epsilon$ | 1,625 | 400 | 400 | 400 |
| $v_m$ | $[-0.1, -0.1]$ | $[0, -1.0]$ | $[0, 1.0]$ | $[0, 0]$ |
| $K_a$ | $[90.9, 90.9]$ | $[100.0, 100.0]$ | $[0, 10.0]$ | $[0, 10.0]$ |
| $k_{o,S}$ | $[0.11, 0.11]$ | $[0.1, 0.1]$ | $[0, 1.0]$ | $[0, 0.1]$ |
| $K_e$ | $[90.9, 90.9]$ | $[100.0, 100.0]$ | $[0, 10.0]$ | $[0, 10.0]$ |
| $k_{o,D}$ | $[0.11, 0.11]$ | $[0.1, 0.1]$ | $[0, 1.0]$ | $[0, 0.1]$ |

## Appendix 2

### Crosslinker binding and unbinding
Kinetic Monte-Carlo: crosslinking protein-filament interactions

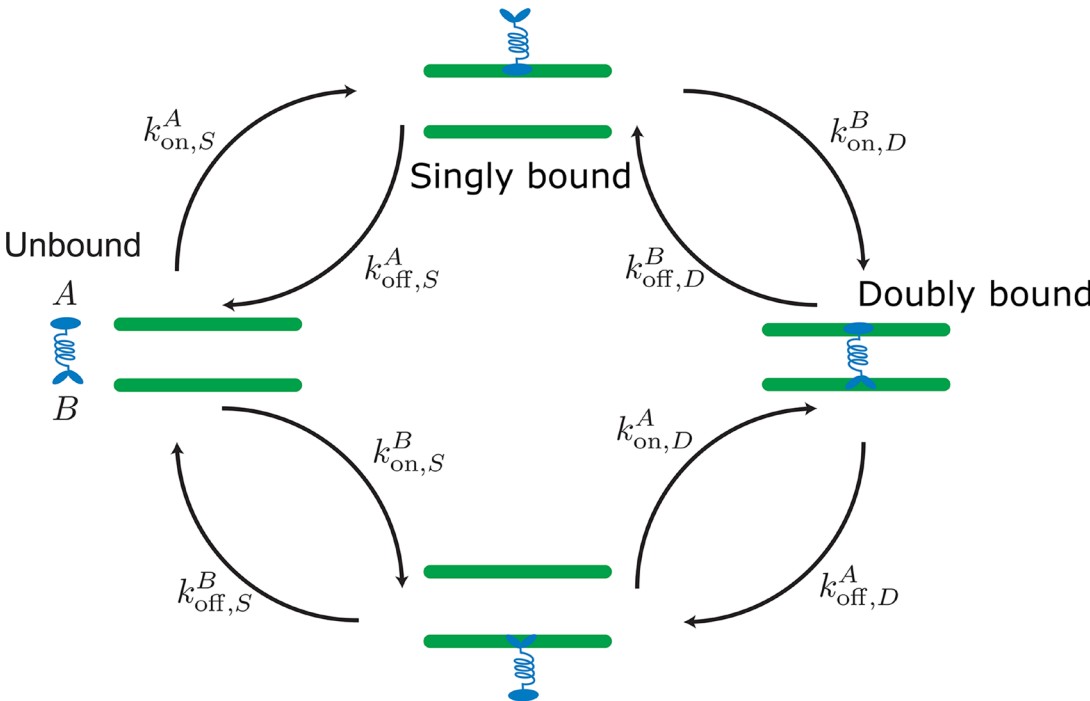

**Appendix 2—figure 1.** Labels and definition of kinetic rates for crosslinking proteins binding to filaments (green) implemented in the kinetic Monte Carlo algorithm. Crosslinking proteins (blue) exist in three different states: neither head attached to a filament (unbound), bound with one head attached to a filament (singly bound), and crosslinking two filaments (doubly bound). Motors and crosslinkers may have different rates for separate binding heads (**A,B**).

Our molecular model simulates distinct filaments and crosslinking proteins (crosslinking motor proteins, passive crosslinkers, etc.). This model includes fluctuations in bound protein number and binding kinetics that recovers the equilibrium distribution of static crosslinking proteins *Gao et al., 2015a*; *Blackwell et al., 2017*; *Rincon et al., 2017*; *Lamson et al., 2019*; *Edelmaier et al., 2020*. Modeled crosslinking proteins in solution bind to one filament and then crosslink two filaments (*Appendix 2—figure 1*). In dense filament networks, the spatial variation of unbound proteins play an important part in the network's reorganization. To account for inhomogeneous concentrations, we explicitly model unbound crosslinkers and develop a method that reproduces one head bound and doubly bound distributions consistent with a mean-field model (Appendix C.2). All binding and unbinding rate calculations are summarized in *Appendix 2—table 1*.

Unbound crosslinking proteins rapidly diffuse in the surrounding fluid until a head binds to a filament. Heads of modeled crosslinking proteins in solution bind to filaments described by the reversible chemical reaction

$$H + B = HB,  \tag{10}$$

where H is a head and B is a binding site on a filament. The association constant $K_a$ of the heads to binding sites is described by the equilibrium equation

$$K_a = \frac{[HB]}{[H][B]} = \frac{k_{on,S}}{k_{off,S}},  \tag{11}$$

where $[X]$ defines the concentration of substance $X$. The association constant has units of inverse molarity, and relates the the on- and off-rate constants $k_{on,S}$ and $k_{off,S}$.

Unbound crosslinkers are modeled as diffusing points with center of mass positions $x_o(t)$ and diffusion constants $d_u$. The heads of a crosslinker have spatial- and time-dependent concentrations $[H] = c(\boldsymbol{x}, t)$. The head binding rate is the volume integral over the product of the on-rate constant, binding site density, and crosslinker concentration

$$R_{\text{on},S}^A(t) = k_{\text{on},S}^A \sum_i \int_{L_i} ds \int dx^3 \epsilon \delta^3(\boldsymbol{x} - \boldsymbol{x}_i(s)) c(\boldsymbol{x}, t), \tag{12}$$

where $k_{\text{on},S}^A = K_a^A k_{\text{off},S}^A$ and $\epsilon$ is the linear binding site density along filaments. The lab position along the $i$th filament $\boldsymbol{x}_i(s)$ is parameterized by $s$.

The binding probability in a timestep $\Delta t$ is an in-homogeneous Poisson process with the cumulative probability function

$$P_{\text{on},S}(\Delta t) = 1 - \exp\left(-\int_0^{\Delta t} dt R_{\text{on},S}(t)\right). \tag{13}$$

We assume the tight binding limit $k_{\text{on},S} \gg k_{\text{off},S}$ and do not consider multiple binding and unbinding events of one crosslinker during a timestep $\Delta t$. The average number of multiple events may be calculated from binding parameters and the timestep allowing one to set a probability threshold (*Lamson et al., 2021*). Heads of the same crosslinking protein are forbidden to be bound to the same filament at the same time.

To describe $c(\boldsymbol{x}, t)$ during a timestep, we first consider a crosslinking protein with two heads connected by a flexible but relatively stiff polymer tether with length $\ell_o$. Because of the tether's stiffness, the radius of gyration of an unbound protein is assumed to be $r_g = \ell_o/2$. The binding heads at the tether's ends move by the tether's rotation and translation. Depending on the timestep's length, either rotation or translation will dominate the evolution of the head distributions.

For most biological crosslinking proteins, the rotational diffusion is fast compared to translational diffusion. When the crosslinker's center does not diffuse far from its position at the beginning of a timestep, that is, $\sqrt{6d_U\Delta t} < r_g$, rotational diffusion dominates and we approximate heads to be within a sphere of radius $r_{c,S} = r_g$ centered at $x_o$. Realistically, the head distributions can vary within this volume but because $\ell_o$ is small compared to filament lengths, we approximate the head distribution as being uniform, that is, $c(\boldsymbol{x}, t) = (4\pi r_{c,S}^3/3)^{-1}(1 - \Theta(|\boldsymbol{x}| - r_{c,S}))$, where $\Theta(x)$ is the Heaviside step function. For larger crosslinking proteins, more detailed spatial distributions may be calculated using freely-jointed or worm-like chain models.

For uniformly distributed heads, the head binding rate is

$$R_{\text{on},S}(t) = \frac{3\epsilon k_{\text{on},S}}{4\pi(r_{c,S})^3} \sum_i L_{\text{in},i}(t), \tag{14}$$

where $L_{in}$ is the filament 's length segment within $r_{c,S}$. To account for cylindrical filaments with diameter $D_{\text{fil}}$, we augment the binding radius such that $r_{c,S} = r_g + D_{\text{fil}}/2$. Since this scenario exists within a regime where the crosslinker or motor does not diffuse far from its initial position in a timestep, we approximate $R_{\text{on},S}(t) \approx R_{\text{on},S}(t_i)$, for $t \in (t_i, t_i + \Delta t)$.

However, it is uncommon that an unbound crosslinker or motor will diffuse less than $r_g$ in a timestep, and so we must account for the protein's translational diffusion. The diffusion equation models the mean spatial distribution of a unbound crosslinking protein's center

$$\frac{\partial c_o(\boldsymbol{x}, t)}{\partial t} = d_U \nabla^2 c_o(\boldsymbol{x}, t), \tag{15}$$

which has the solution

$$c_o(\boldsymbol{x}, t) = \frac{1}{(4\pi d_U)^{3/2}} \exp\left[\frac{-|\boldsymbol{x} - \boldsymbol{x}_o|^2}{4d_U t}\right]. \tag{16}$$

If the characteristic diffusion length $\sqrt{d_U \Delta t} \gg r_g$, then *equation (14)* underestimates binding (*Appendix 2—figure 2*). The large diffusion distance also allows us to approximate the head distribution as matching the protein's spatial distribution, $c_o(\boldsymbol{x}, t) \approx c(\boldsymbol{x}, t)$. Substituting the binding

rate *equation (12)* and the solution to the diffusion *equation (16)* into the integral of *equation (13)* gives

$$\int_0^{\Delta t} dt R_{\text{on},S}(t) = \sum_i \int_0^{\Delta t} dt \frac{k_{\text{on},S}\epsilon}{(4\pi d_U t)^{3/2}} \int_{L_i} ds_i \int dx^3 \delta^3(\boldsymbol{x} - \boldsymbol{x}_i'(s_i)) \exp\left[\frac{-|\boldsymbol{x} - \boldsymbol{x}_o|^2}{4 d_U t}\right]. \tag{17}$$

For straight, rigid filaments, we take the volume and time integrals while reparameterizing $|\boldsymbol{x} - \boldsymbol{x}_o|^2$ by the crosslinker's perpendicular $h$ and parallel $s$ distances from a filament segment's center. This gives the linear binding probability density for a filament

$$p_{\text{on},S}(h_\perp, s_i, \Delta t) = \int_0^{\Delta t} dt \frac{\partial R_{\text{on},S}}{\partial s_i} = \frac{K_a \epsilon k_{o,S}}{4\pi d_U}\left(\frac{1}{\sqrt{h_i^2 + s_i^2}} \operatorname{erfc}\left[\frac{\sqrt{h_i^2 + s_i^2}}{\sqrt{4 d_U \Delta t}}\right]\right). \tag{18}$$

Integrating over $s_i$ gives the binding probability of one crosslinker head to a single filament. The total binding probability is then

$$P_{\text{on},S}(\Delta t) = 1 - \exp\left(-\sum_i^N \int_{L_i} ds_i p_{\text{on},S}(h_i, s_i, \Delta t)\right), \tag{19}$$

where $N$ is the number of filaments surrounding the crosslinker head.

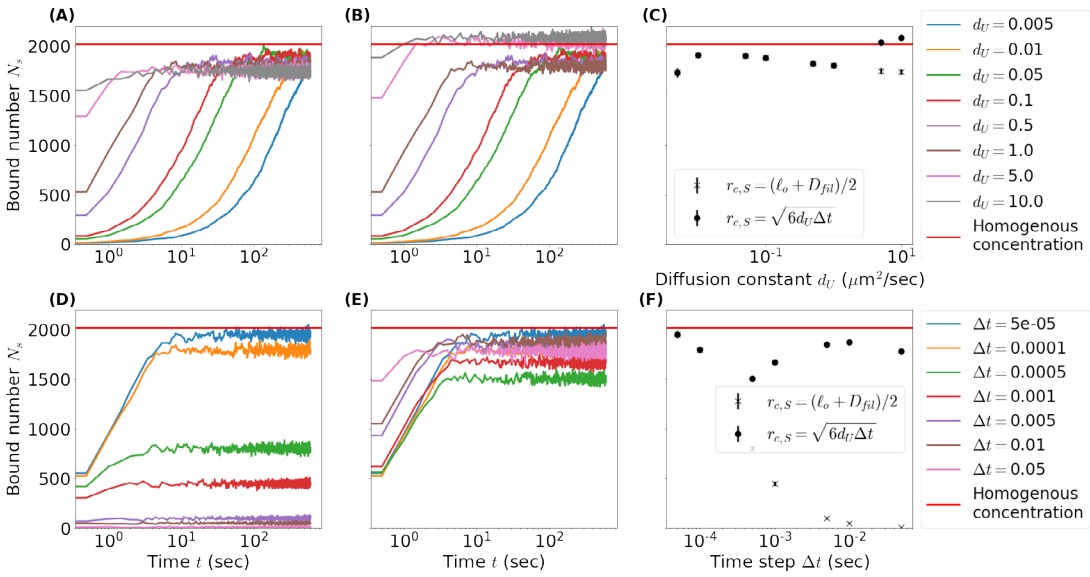

**Appendix 2—figure 2.** Comparison of initially unbound passive crosslinkers binding to a $1\,\mu m$ filament with binding radii set to a crosslinker's radius of gyration versus a binding radius $\sim \sqrt{d_U \Delta t}$. (**A, D**) Number of singly bound crosslinkers over time as the unbound diffusion constant $d_U$ (**A**) and time step $\Delta t$ (**D**) vary while binding radius remains unchanged $r_{c,S} = (\ell_o + D_{\text{fil}})/2$. Red lines mark the steady-state number of singly bound crosslinkers for a homogeneous reservoir calculated from *equations (26)-(30)*. (**B, E**) Same as A and D but binding radius scales as the root mean square of diffused distance in a time step $r_{c,S} = \sqrt{6 d_U \Delta t}$. (**C, F**) Comparison of the steady-state number of singly bound crosslinkers as a function of $d_U$ (**C**) and $\Delta t$ (**F**) for both definitions of $r_{c,S}$. Simulation parameters: periodic box length $=2\,\mu m$, filament length $L = 1\,\mu m$, linear binding site density $\epsilon = 27\,\mu m^{-1}$, crosslinker number $N = 4000$, crosslinker length $\ell_o = 50\,\text{nm}$, association constant $K_a = 90.9\,(\mu \text{mol/L})^{-1}$, unbinding rate $k_{o,S} = 5\,\text{s}^{-1}$. Unless otherwise stated unbound diffusion constant $d_U = 1\,\mu\text{m}^2/\text{s}$ and timestep $\Delta t = 0.0001\,\text{s}$

Calculating the binding probability from this function and ensuring that the protein unbinds so that detailed-balance is satisfied is computationally prohibitive. Instead, setting $r_{c,S}$ to the root mean square diffusion distant $\sqrt{6 d_U \Delta t}$ and using the rate *equation 14*, we obtain a useful approximation to *equation 19*. This is computationally efficient and mitigates the low binding rates when diffusion or times steps are large (*Appendix 2—figure 2*). We note that the accuracy of this approximation

is dependent on the length of filaments in the simulation with longer filaments reducing the error from edge effects. Future work will focus on developing methods to more accurately reproduce the above binding distribution.

Once bound, a crosslinking protein's head unbinds with a constant rate

$$k_{\text{off},S} = k_{\text{o},S}.$$ (20)

If implementing *equation 14*, the protein unbinds into a uniform sphere of radius $r_{c,S}$. This ensures crosslinkers bind to and from regions in a way that satisfies detailed-balance.

With one head bound, a crosslinker's tether deforms to bind its other head to adjacent filaments. Deforming a tether requires energy, implying crosslinking kinetics depend on tether deformation. For passive crosslinkers and motors with rapid kinetic rates compared to stepping rate, the ratio of binding and unbinding rates to and from a position on a filament is proportional to the Boltzmann factor of the binding free energy.

With one head bound to filament at position $s_i$, the binding rate constant $k_{\text{on},[}, D]$ for binding to a location $s_j$ on filament $j$ is

$$k_{\text{on},D}(s_i, s_j) = K_e k_{\text{o},D} e^{-\beta U_{i,j}(s_i, s_j)},$$ (21)

where $k_{\text{o},[}, D] = k_{\text{off},[}, D](U_{i,j} = 0)$ is the unbinding-rate of crosslinking proteins when no force is applied, $K_e$ is a binding association constant similar to $K_a$, and $U_{i,j}$ is the free energy contribution from the tether

$$U_{i,j} = \frac{\kappa_{\text{xl}}}{2}(\ell(s_i, s_j) - \ell_o - D_{\text{fil}})^2$$ (22)

Before crosslinking, the unbound motor head explores an effective volume $V_{\text{bind}}$ centered around the bound motor head. Not considering steric interactions with filaments, this volume is the free head's position weighted by the Boltzmann factor integrated over all space.

$$V_{\text{bind}} = \int e^{-\beta U_{i,j}} dr^3 = 4\pi \int_0^{r_{c,D}} e^{-\beta U_{i,j}} r^2 dr.$$ (23)

We impose an integration cutoff radius $r_{c,D}$ where the integrand becomes sufficiently small, making the factor consistent with a finite lookup table *Lamson et al., 2021*. The binding head's positional distribution must also satisfy the Boltzmann factor. We recover the proper binding distribution through inverse transformation sampling of *equation (21) Lamson et al., 2021*.

Theory and experimental evidence suggests that binding rates depend not only on energy but also force *Evans and Ritchie, 1997*; *Dudko et al., 2006*; *Walcott, 2008*; *Guo et al., 2019*. This allows for catch-bond-like behavior where proteins remain crosslinked for longer if under tension and release quicker if compressed. We replicate this behavior with the function

$$f_F(s_i, s_j) = \kappa_{\text{xl}}\left(\frac{\lambda}{2}(\ell(s_i, s_j) - \ell_o)^2 + x_c(\ell(s_i, s_j) - \ell_o)\right)$$ (24)

where $\lambda$ and $x_c$ are the energy factor and characteristic length specifying the behavior of the energy- and force-dependent binding/unbinding, respectively. For values of $x_c < 0$, you see catch-bond like behavior whereas values of $x_c > 0$ exhibit slip bond behavior *Walcott, 2008*; *Edelmaier et al., 2020*. This formalism can also be used to add in angle dependence.

When we include an effective energy and/or force dependence, the unbinding rate becomes

$$k_{\text{off},D}(s_i, s_j) = k_{\text{o},D} e^{\beta f_F(s_i, s_j)}.$$ (25)

This does not change the final stored energy of either bound state but does effect the frequency at which the motors will switch between having one head bound and crosslinking.

**Appendix 2—table 1.** The transition rates between all possible states of a crosslinker $U \rightleftharpoons (S_A, S_B) \rightleftharpoons D$.

$(S_A, S_B)$ means either head $A$ or $B$ is bound but the other is unbound. All binding rates account for the linear binding density $\epsilon$ is the length of filament with center-of-mass position $x_i$ and orientation $p_i$ inside the capture sphere with cutoff radius $r_{c,s}$ relative to position of motor/crosslinker $x$. The

sum is over all possible candidate filaments . The unbound-singly bound transition $U \rightleftharpoons (S_A, S_B)$ is determined by the association constant $K_a$ and the force-independent off rate $k_{o,s}$. Similarly, the singly bound-doubly bound transition $(S_A, S_B) \rightleftharpoons D$ is determined by the association constant $K_e$ and force-independent off rate $k_{o,d}$ with an additional factor $V_{\text{bind}}$, the effective volume explored by the unattached head while the motor/crosslinker is singly bound. Energy dependence in the $(S_A, S_B) \rightleftharpoons D$ transition rates is imposed by the Boltzmann factor that is a function of $\beta = 1/(k_B T)$, the tether length of the motor/crosslinker attached to filaments and $j$ at locations $s_i$ and $s_j \ell(s_i, s_j, \boldsymbol{x}_i, \boldsymbol{p}_i, \boldsymbol{x}_j, \boldsymbol{p}_j)$, the characteristic length of the tether not under load $\ell_o$, and the filament diameter $D_{\text{fil}}$. The dimensionless factor $\lambda$ determines the energy dependence in the unbinding rate while the $x_c$ is the characteristic length that determines the force dependence. The latter is not used in the simulations of the main text but is implemented in the code base.

| Process | Rate | Value |
|---|---|---|
| $U \rightarrow (S_A, S_B)$ | $R_{\text{on},S}(\boldsymbol{x}, t)$ | $\dfrac{3\epsilon K_a k_{o,S}}{4\pi r_{c,S}^3} \sum_i L_{\text{in},i}(\boldsymbol{x}, t)$ |
| $(S_A, S_B) \rightarrow U$ | $R_{\text{off},S}$ | $k_{o,S}$ |
| $(S_A, S_B) \rightarrow D$ | $R_{\text{on},D}(s_i, t)$ | $\dfrac{\epsilon K_e k_{o,D}}{V_{\text{bind}}} \sum_j \int_{L_j} ds_j \exp\left[ -\beta \kappa_{\text{xl}} \left( \dfrac{1-\lambda}{2} \left( \ell - \ell_o - D_{\text{fil}} \right)^2 - x_c \left( \ell - \ell_o - D_{\text{fil}} \right) \right) \right]$ |
| $D \rightarrow (S_A, S_B)$ | $R_{\text{off},D}(s_i, s_j, t)$ | $k_{o,D} \exp\left[ \beta \kappa_{\text{xl}} \left( \dfrac{\lambda}{2} \left( \ell - \ell_o - D_{\text{fil}} \right)^2 + x_c \left( \ell - \ell_o - D_{\text{fil}} \right) \right) \right]$ |

## Mean-field theory for crosslinking proteins

We expand on our previous mean-field motor density model to include motors that have dissimilar heads, diffusion and walking in singly and doubly bound states, and a time-dependent homogeneous concentration of unbound crosslinking proteins *Lamson et al., 2021*. This last addition imposes the condition that the total number of proteins when all bound and unbound states are accounted for remains constant.

This requires a system of equations with $N(N-1)$ crosslinking densities $\psi_{i,j}^{A,B}$, $2N$ singly bound densities $\chi_i^A$ and $\chi_i^B$, and an unbound density $C$ to model all crosslinking proteins between $N$ filaments. By convention, $\psi_{i,j}^{A,B} = \psi_{j,i}^{B,A}$

$$
\frac{\partial \psi_{i,j}^{A,B}}{\partial t} + \frac{\partial}{\partial s_i}\left[ -d_{i,j}^A \frac{\partial \psi_{i,j}^{A,B}}{\partial s_i} + (v_{drag,i,j}^A + v_{walk,i,j}^A)\psi_{i,j}^{A,B} \right] + \frac{\partial}{\partial s_j}\left[ -d_{i,j}^B \frac{\partial \psi_{i,j}^{A,B}}{\partial s_j} + (v_{drag,i,j}^B + v_{walk,i,j}^B)\psi_{i,j}^{A,B} \right]
$$
$$
= \epsilon(k_{on,i,j}^A \chi_i^B + k_{on,i,j}^B \chi_j^A) - (k_{off,i,j}^A + k_{off,i,j}^B)\psi_{i,j}^{A,B},
\tag{26}
$$

$$
\frac{\partial \psi_{i,j}^{B,A}}{\partial t} + \frac{\partial}{\partial s_i}\left[ -d_{i,j}^B \frac{\partial \psi_{i,j}^{B,A}}{\partial s_i} + (v_{drag,i,j}^B + v_{walk,i,j}^B)\psi_{i,j}^{B,A} \right] + \frac{\partial}{\partial s_j}\left[ -d_{i,j}^A \frac{\partial \psi_{i,j}^{B,A}}{\partial s_j} + (v_{drag,i,j}^A + v_{walk,i,j}^A)\psi_{i,j}^{B,A} \right]
$$
$$
= \epsilon(k_{on,i,j}^B \chi_i^A + k_{on,i,j}^A \chi_j^B) - (k_{off,i,j}^A + k_{off,i,j}^B)\psi_{i,j}^{B,A},
\tag{27}
$$

$$
\frac{\partial \chi_i^A}{\partial t} + \frac{\partial}{\partial s_i}\left[ -d_i^A \frac{\partial \chi_i^A}{\partial s_i} + v_{walk,i}^A \chi_i^A \right] = \epsilon k_{on,S}^A C - k_{off,S}^A \chi_i^A + \sum_j \int_{L_j} ds_j \left( k_{off,i,j}^B \psi_{i,j}^{A,B} - \epsilon k_{on,i,j}^B \chi_i^A \right),
\tag{28}
$$

$$
\frac{\partial \chi_i^B}{\partial t} + \frac{\partial}{\partial s_i}\left[ -d_i^B \frac{\partial \chi_i^B}{\partial s_i} + v_{walk,i}^B \chi_i^B \right] = \epsilon k_{on,S}^B C - k_{off,S}^B \chi_i^B + \sum_j \int_{L_j} ds_j \left( k_{off,i,j}^A \psi_{i,j}^{B,A} - \epsilon k_{on,i,j}^A \chi_i^B \right),
\tag{29}
$$

$$\frac{\partial C}{\partial t} = \sum_i \int_{L_i} ds_i \left[ \frac{k_{\mathrm{off},S}^A \chi_i^A + k_{\mathrm{off},S}^B}{V} - \epsilon \left( k_{\mathrm{on},S}^A + k_{\mathrm{on},S}^B \right) C \right]. \tag{30}$$

For heads $A$ and $B$, crosslinking diffusion constants $d_{i,j}^A$ and $d_{i,j}^B$, the drag speeds $v_{drag,i,j}^A$ and $v_{drag,i,j}^B$, and walking speeds $v_{walk,i,j}^A$ and $v_{walk,i,j}^B$ have been shown to depend on the force exerted on the binding heads and thus the stretch of the tether $\ell(s_i, s_j)$. No tether force acts on singly bound motor proteins but the singly bound diffusion constants $d_i^A$ and $v_i^B$ and walking speeds $v_{drag,i}^A$ and $v_{drag,i}^B$ may depend on $s_i$ through some other physical mechanism such as crowding or state of the filament's lattice. The total number of crosslinking proteins of the system is

$$N = \sum_i \sum_j \int_{L_i} ds_i \int_{L_j} ds_j \psi_{i,j}^{A,B} + \sum_i \int_{L_i} ds_i \left( \chi_i^A + \chi_i^B \right) + CV \tag{31}$$

and is constant in time.

## Appendix 3

### Filament dynamics

Constraint quadratic programming

In the main text we discussed specifically filament. In fact, our method is applicable to rigid bodies of arbitrary shapes. Here we derive the detailed equations.

The configuration of each particle is tracked by its center location $x$ in the lab frame and its orientation $\theta = [s, p] \in \mathbb{R}^4$ as a quaternion *Delong et al., 2015*. This $p$ is the vector component of the quaternion, not the unit orientation vector. There are other choices to specify the orientation, such as Euler-angles and rotation matrices, but we prefer quaternions for simplicity. The geometric configuration $\mathcal{C}$ for all $N$ filaments can be written as a column vector:

$$\mathcal{C} = \left[x_1, \theta_1, \ldots, x_N, \theta_N\right]^T \in \mathbb{R}^{7N}, \tag{32}$$

which is a function of time: $\mathcal{C}(t)$. The translational and angular velocity $U, \Omega$ of all filaments can also be written as a column vector:

$$\mathcal{U} = \left[U_1, \Omega_1, \ldots, U_N, \Omega_N\right]^T \in \mathbb{R}^{6N}. \tag{33}$$

Similarly we can write the force and torque $F, T$ applied on all filaments as a column vector:

$$\mathcal{F} = \left[F_1, T_1, \ldots, F_N, T_N\right]^T \in \mathbb{R}^{6N}. \tag{34}$$

The kinematic equation of motion 35 maps $\mathcal{U}$ to $\dot{\mathcal{C}}(t) = \partial \mathcal{C}/\partial t$, via a geometric matrix $\mathcal{G}$.

$$\dot{\mathcal{C}}(t) = \mathcal{G}\mathcal{U}. \tag{35}$$

$\mathcal{G} \in \mathbb{R}^{7N \times 6N}$ is a block diagonal matrix, with one $3 \times 3$ and one $4 \times 3$ block for each particle:

$$\mathcal{G} = \begin{bmatrix} I^3 & & & & \\ & \Psi_1 & & & \\ & & I^3 & & \\ & & & \Psi_2 & \\ & & & & \ddots \end{bmatrix}. \tag{36}$$

$I^3$ is the $3 \times 3$ identity matrix, same for every particle. Each $I^3$ block simply corresponds to the translational motion $\dot{x}_j = U_j$ of each particle $j$. Each $\Psi_j \in \mathbb{R}^{4 \times 3}$ refers to the rotational motion $\dot{\theta}_j = \Psi_j \Omega_j$ of each particle, where for each $j$:

$$\Psi(\theta) = \frac{1}{2} \begin{bmatrix} -p^T \\ sI - P \end{bmatrix}, \quad P_{ij} = \epsilon_{ikj} p_k. \tag{37}$$

Here $\epsilon_{ikj}$ is the Levi-Civita symbol for cross-product in 3D space.

The biological filaments we consider mostly have lengths on the nm to $\mu m$ scales. At these scales, solvent viscosity dominates and inertia effects can be ignored, which is the so-called Stokes regime where the mobility matrix $\mathcal{M}$ maps the force $\mathcal{F}$ linearly to the velocity $\mathcal{U}$:

$$\mathcal{U} = \mathcal{M}\mathcal{F}, \quad \mathcal{F} = \mathcal{F}_C + \mathcal{F}_L + \mathcal{F}_B + \mathcal{F}_E. \tag{38}$$

$\mathcal{F}$ includes collision force $\mathcal{F}_C$ between particle-particle and particle-container pairs, linker force between particle pairs $\mathcal{F}_L$ generated by doubly bound crosslinkers, Brownian force on each particle $\mathcal{F}_B$ generated by thermal fluctuations, and other externally applied forces $\mathcal{F}_E$ through gravity and electrostatic fields.

In principal, *Equation 35* together with *Equation 38* can be integrated directly because both $\mathcal{M}$ and $\mathcal{F}$ are functions of the geometry $\mathcal{C}$ and time only. However, this approach is usually impractical, because $\mathcal{F}_C$ or $\mathcal{F}_L$ is usually very stiff functions of the geometry. For example, the collision force

$\mathcal{F}_C$ is usually computed by assuming a very stiff pairwise potential between filaments, such as the Lennard-Jones or WCA potential. This stiffness poses severe limits on the stability of all explicit temporal integrators. We discussed this problem in detail for collision forces $\mathcal{F}_C$ in our previous work on Brownian spherocylinders *Yan et al., 2019* and rigid spheres in Stokes flow *Yan et al., 2020*. Instead of computing $\mathcal{F}_C$ using repulsive potentials, we imposed non-overlapping constraints on the geometry $\mathcal{C}$ while integrating *Equation 35*.

## Equation of motion with geometric constraints

In the following, the subscript $_c$ refers to constraints, which includes both unilateral (with subscript $_u$) and bilateral (with subscript $_b$) constraints. Unilateral constraints refer to those inequality constraints, i.e., constraints imposed from one side, while bilateral constraints refer to equality constraints. In our system, unilateral constraints come from collisions and bilateral constraints come from doubly bound crosslinkers. The subscript $_{nc}$ refers to non-constraint, i.e., physical components that are independent of the constraints.

For unilateral constraints, we define the grand distance function $\mathbf{\Phi}_u$ between every pair of particles as a column vector:

$$\mathbf{\Phi}_u = \left[ \Phi_{u,P_1Q_1}, \Phi_{u,P_2Q_2}, \cdots, \Phi_{u,P_{N_u}Q_{N_u}} \right]^T \in \mathbb{R}^{N_u}, \tag{39}$$

where each $\Phi_{u,P_jQ_j}$ is the minimal distance between particles with indices $P_j$ and $Q_j$. Similarly, we define the grand distance function $\mathbf{\Phi}_b$ for bilateral constraints:

$$\mathbf{\Phi}_b = \left[ \Phi_{b,P_1Q_1}, \Phi_{b,P_2Q_2}, \cdots, \Phi_{b,P_{N_b}Q_{N_b}} \right]^T \in \mathbb{R}^{N_b}, \tag{40}$$

where each $\Phi_{b,P_jQ_j}$ is the distance between two fixed points on particles $P_j$ and $Q_j$, respectively. Physically, $\Phi_{b,P_j,Q_j}$ is simply the length of each doubly bound crosslinker. With this definition, there are in total $N_u$ unilateral and $N_b$ bilateral constraints in the system. In other words, there are in total $N_u$ possibly colliding pairs of filaments and $N_b$ doubly bound crosslinkers. Both kinds of constraints are functions of the system geometry, so we shall write them as $\mathbf{\Phi}_b(\mathcal{C})$ and $\mathbf{\Phi}_u(\mathcal{C})$ in the following when necessary.

The force magnitude between all pairs of particles for unilateral and bilateral constraints can be written similarly as column vectors:

$$\boldsymbol{\gamma}_u = \left[ \gamma_{u,1}, \gamma_{u,2}, \cdots, \gamma_{u,N_u} \right]^T \in \mathbb{R}^{N_u}, \tag{41}$$

$$\boldsymbol{\gamma}_b = \left[ \gamma_{b,1}, \gamma_{b,2}, \cdots, \gamma_{b,N_b} \right]^T \in \mathbb{R}^{N_b}. \tag{42}$$

For each $\Phi_{u,P_jQ_j}$ or $\Phi_{b,P_jQ_j}$, there is a corresponding force magnitude $\gamma_{u,j}$ or $\gamma_{b,j}$, the (normalized) direction vector $\hat{\boldsymbol{e}}_{P_j} = -\hat{\boldsymbol{e}}_{Q_j}$ of this force, and the location $\boldsymbol{y}_{P_j}$ and $\boldsymbol{y}_{Q_j}$ where this force is applied on the filament $P_j$ and $Q_j$ respectively, as shown in *Appendix 3—figure 1*. With norm vectors defined in this way, $\gamma_u$ or $\gamma_b$ is positive when the force is repulsive between two filaments.

For unilateral constraints $\mathbf{\Phi}_u$ and $\boldsymbol{\gamma}_u$ satisfy this complementarity condition:

$$0 \leq \mathbf{\Phi}_u(\mathcal{C}) \perp \boldsymbol{\gamma}_u \geq 0 \tag{43}$$

This condition means $\mathbf{\Phi}_u(\mathcal{C})$ and $\boldsymbol{\gamma}_u$ are orthogonal to each other, and all components of $\mathbf{\Phi}_u(\mathcal{C})$ and $\boldsymbol{\gamma}_u$ are non-negative *Yan et al., 2019*.

For bilateral constraints $\mathbf{\Phi}_b$ and $\boldsymbol{\gamma}_b$ satisfy this linear equality condition because they are modeled as Hookean springs:

$$\mathcal{K}\left[ \mathbf{\Phi}_b(\mathcal{C}) - \mathbf{\Phi}_b^0 \right] = -\boldsymbol{\gamma}_b. \tag{44}$$

$\mathcal{K} \in \mathbb{R}^{N_b \times N_b}$ is a diagonal matrix, with the stiffness constant κ for each spring on its diagonal $\left[ \kappa_1, \kappa_2, \ldots \right]$. Obviously every constant $\kappa_j$ is positive. $\mathbf{\Phi}_b(\mathcal{C})$ and $\mathbf{\Phi}_b^0$ represent the current and free length of every spring.

Both unilateral and bilateral constraints change over time, as particles move and springs attach to and detach from particles.

All combined together, we reach the equation of motion with geometric constraints:

$$\dot{\mathcal{C}}(t) = \mathcal{G}(\mathcal{C})\mathcal{U}, \tag{45a}$$

$$\mathcal{U} = \mathcal{M}\mathcal{F} = \mathcal{M}\left(\mathcal{F}_u + \mathcal{F}_b + \mathcal{F}_{nc}\right), \tag{45b}$$

$$0 \leq \boldsymbol{\Phi}_u(\mathcal{C}) \perp \gamma_u \geq 0, \tag{45c}$$

$$\mathcal{K}\left[\boldsymbol{\Phi}_b(\mathcal{C}) - \boldsymbol{\Phi}_b^0\right] = -\gamma_b. \tag{45d}$$

These equations are solvable when closed by a geometric relation, which maps the force magnitude $\gamma_u$ and $\gamma_b$ to the force vectors $\mathcal{F}_u$ and $\mathcal{F}_b$:

$$\mathcal{F}_u = \mathcal{D}_u\gamma_u, \quad \mathcal{F}_b = \mathcal{D}_b\gamma_b, \tag{46}$$

where $\mathcal{D}_u$ and $\mathcal{D}_b$ are sparse matrices containing all orientation vectors of unilateral and bilateral forces, i.e., all $\hat{e}$ vectors as shown in *Appendix 3—figure 1*. More details about the definition of $\mathcal{D}$ can be found in the following.

Both $\mathcal{D}_u$ and $\mathcal{D}_b$ depend only on the geometry norm vectors $\hat{e}_{P_j}, \hat{e}_{Q_j}$ and location of constraints $y_{P_j}, y_{Q_j}$, together with the particle indices $P_j, Q_j$, i.e., which particles appear within the vicinity of each other and which are bound to each other by springs.

Further, this constraint formulation is also applicable to the case where one constraint is not between a pair of particles but between one particle and one externally imposed confinement or boundary, for example, a flat substrate or a spherical shell. The only necessary modification in this case is to ignore one side of the collision geometry when constructing the matrix $\mathcal{D}_u$ and $\mathcal{D}_b$. For example, if a particle $P$ collides with a fixed substrate, we only include $\hat{e}_P$ and $y_P$ in $\mathcal{D}_u$, because this substrate does not appear in the mobility matrix $\mathcal{M}$.

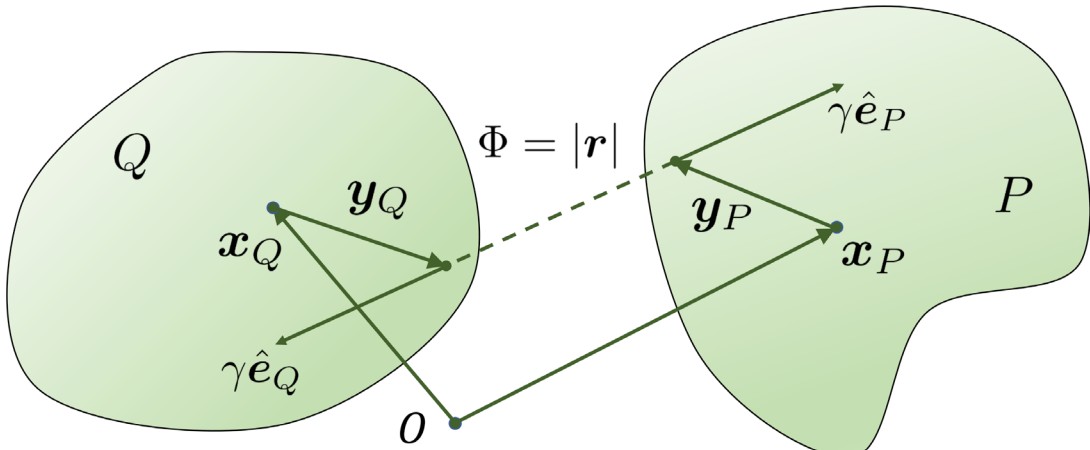

**Appendix 3—figure 1.** The geometry for a pair of rigid particles. The distance between two marked points $\Phi = |\boldsymbol{r}|$, where $\boldsymbol{r} = x_P + y_P - x_Q - y_Q$.

## Temporal discretization and convex quadratic programming

*Equations 45 and 46* generate a differential variational inequality (DVI), which can be solved when equipped with a timestepping scheme. In this work we use the linearized implicit Euler timestepping scheme, similar to our previous work *Yan et al., 2020*; *Yan et al., 2019*, for three reasons:

- It is straightforward to integrate with both the Brownian motion and the stochastic binding and unbinding of crosslinkers into an Euler scheme.
- The scheme cannot be explicit. Otherwise $\Delta t$ is limited to be tiny by the temporal stiffness of collision and doubly bound crosslinkers.
- The implicit scheme is linearized to avoid expensive large-scale non-linear problems.

With timestep $\Delta t = h$, *Equations 45 and 46* are discretized at timestep $k$ as:

$$\frac{1}{h}(\mathcal{C}^{k+1} - \mathcal{C}^k) = \mathcal{G}^k \mathcal{U}^k, \tag{47a}$$

$$\mathcal{U}^k = \mathcal{M}^k \left( \mathcal{F}_u^k + \mathcal{F}_b^k + \mathcal{F}_{nc}^k \right), \tag{47b}$$

$$\mathcal{F}_u^k = \mathcal{D}_u^k \gamma_u^k, \quad \mathcal{F}_b^k = \mathcal{D}_b^k \gamma_b^k, \tag{47c}$$

$$0 \leq \Phi_u^{k+1} \perp \gamma_u^k \geq 0, \tag{47d}$$

$$\mathcal{K}^k \left[ \Phi_b^{k+1} - \Phi_b^{0,k} \right] = -\gamma_b^k. \tag{47e}$$

The unknowns to be solved at every timesteps are the constraint force magnitude $\gamma_u^k, \gamma_b^k$. *Equations 47*d and e are nonlinear because $\Phi_u^{k+1}$ and $\Phi_b^{k+1}$ are nonlinear functions of $\mathcal{C}^{k+1}$. Therefore, we linearize these two terms:

$$0 \leq \Phi_u^k + h\nabla_C \Phi_u^k \mathcal{G}^k \mathcal{M}^k \left[ \mathcal{F}_{nc}^k + \mathcal{D}_u^k \gamma_u^k + \mathcal{D}_b^k \gamma_b^k \right] \perp \gamma_u^k \geq 0, \tag{48a}$$

$$0 = \Phi_b^k - \Phi_b^0 + h\nabla_C \Phi_b^k \mathcal{G}^k \mathcal{M}^k \left[ \mathcal{F}_{nc}^k + \mathcal{D}_u^k \gamma_u^k + \mathcal{D}_b^k \gamma_b^k \right] + \mathcal{K}^{-1} \gamma_b^k \perp \gamma_b^k \in \mathbb{R}. \tag{48b}$$

Here we have also rewritten the *Equation 47* e into a equivalent form, similar to *Equation 47* d. The right side, $\gamma_u \geq 0$ and $\gamma_b \in \mathbb{R}$ should be understood in the component-wise sense. *Equations 48* a and b are now closed and $\gamma_u^k, \gamma_b^k$ can be solved. We shall drop the superscript $k$ in the following derivations because we shall repeat this solution process at every timestep.

Then *equation (48)* can be written in the block-matrix form:

$$0 \leq \begin{bmatrix} A \\ 0 = \end{bmatrix} \begin{bmatrix} A \\ D \end{bmatrix} + \begin{bmatrix} B & C \\ E & F \end{bmatrix} \begin{bmatrix} \gamma_u \\ \gamma_b \end{bmatrix} \perp \begin{bmatrix} \gamma_u \\ \gamma_b \end{bmatrix} \begin{matrix} \geq 0 \\ \in \mathbb{R} \end{matrix} \tag{49}$$

where the blocks are clear from *equation (48)*

$$A \quad = \frac{1}{h}\Phi_u + \mathcal{D}_u^T \mathcal{M} \mathcal{F}_{nc} \tag{50a}$$

$$B \quad = \nabla_{\mathcal{C}} \Phi_u \mathcal{G}^k \mathcal{M} \mathcal{D}_u = \mathcal{D}_u^T \mathcal{M} \mathcal{D}_u \tag{50b}$$

$$C \quad = \nabla_{\mathcal{C}} \Phi_u \mathcal{G} \mathcal{M} \mathcal{D}_b = \mathcal{D}_u^T \mathcal{M} \mathcal{D}_b \tag{50c}$$

$$D \quad = \frac{1}{h}\left( \Phi_b - \Phi_b^0 \right) + \mathcal{D}_b^T \mathcal{M} \mathcal{F}_{nc} \tag{50d}$$

$$E \quad = \nabla_{\mathcal{C}} \Phi_b \mathcal{G} \mathcal{M} \mathcal{D}_u = \mathcal{D}_b^T \mathcal{M} \mathcal{D}_u \tag{50e}$$

$$F \quad = \nabla_{\mathcal{C}} \Phi \mathcal{G} \mathcal{M} \mathcal{D}_b = \mathcal{D}_b^T \mathcal{M} \mathcal{D}_b + \frac{1}{h}\mathcal{K}^{-1} \tag{50f}$$

Here we used the fact that:

$$\nabla_{\mathcal{C}} \Phi_u \mathcal{G} \quad = \mathcal{D}_u^T, \tag{51a}$$

$$\nabla_{\mathcal{C}} \Phi_b \mathcal{G} \quad = \mathcal{D}_b^T. \tag{51b}$$

The first relation has been well known in the problem of collision constraints *Anitescu et al., 1996*. In this work we extend this result to bilateral constraints. A proof of this is detailed in Section Symmetry of the geometrically constrained optimization problem.

This formulation means that the coefficient matrix is Symmetric-Positive-Semi-Definite (SPSD), because the mobility matrix $\mathcal{M}$ is SPD and $\frac{1}{h}\mathcal{K}^{-1}$ is positive & diagonal:

$$\begin{bmatrix} B & C \\ E & F \end{bmatrix} = \begin{bmatrix} \mathcal{D}_u^T \\ \mathcal{D}_b^T \end{bmatrix} \mathcal{M} \begin{bmatrix} \mathcal{D}_u & \mathcal{D}_b \end{bmatrix} + \begin{bmatrix} 0 & 0 \\ 0 & \frac{1}{h}\mathcal{K}^{-1} \end{bmatrix} \tag{52}$$

Because of this SPSD property, solving *Equations 48* is equivalent to solving a constrained quadratic programming (CQP) due to the Karush-Kuhn-Tucker condition *Nocedal and Wright, 2006*:

$$\min_\gamma f(\gamma) \quad = \tfrac{1}{2}\gamma^T M \gamma + \gamma^T q, \tag{53a}$$

$$\text{subject to} \quad \begin{bmatrix} I^{N_u \times N_u} & 0 \end{bmatrix} \gamma \quad \geq 0. \tag{53b}$$

Here $\gamma = [\gamma_u, \gamma_b] \in \mathbb{R}^{N_u + N_b}$ is a column vector, and

$$M \quad = \begin{bmatrix} B & C \\ E & F \end{bmatrix}, \quad q = \begin{bmatrix} A \\ D \end{bmatrix}. \tag{54}$$

This can be conveniently understood as following. $q$ represent the current values of the constraint functions $\Phi$ plus the (linearized) changes due to non-constraint forces $\mathcal{F}$, such as Brownian fluctuations. $M$ represent the linearized relation between the unknown constraint force $\gamma$ and the changes of the constraint functions $\Phi$.

Solving one global optimization problem at every timestep is usually expensive, because the dimension of this problem (53) can be very large in a system with many particles and constraints. However, this CQP. (53) is a class of well understood optimization problem and fast algorithms exist. We previously developed a fully parallel Barzilai-Borwein projected gradient descent (BBPGD) method *Yan et al., 2020*; *Yan et al., 2019* to efficiently solve this problem for unilateral constraints only. In this work we found that the same BBPGD method also works very well for the current problem.

One way to understand the constraint optimization method is that the temporal integration 'jumps' on a timescale that the relaxation timescales of unilateral and bilateral constraints (collisions and crosslinker springs) are bypassed. As a special case, in the limit of infinitely stiff springs where $\mathcal{K}^{-1} \to 0$ the quadratic term matrix $M$ is still SPSD and the *Equation 53* is still easily solved. Physically speaking, in this case the bilateral constraints degenerate from deformable springs to non-compliant joints.

Last but not least, due to the linearization in *Equations 48* our geometric constraint method has some inevitable numerical errors in imposing both types of constraints for any finite timestep size $\Delta t = h$. In other words, there may be some slight residual overlaps between filaments even if *Equation 48* are exactly solved. Such residual overlaps converge to zero as the timestep size $\Delta t = h$ decreases to zero, which follows the typical first order numerical convergence. In principal, such residual overlaps due to linearization errors can be eliminated if the full nonlinear constraint problem is solved. However, the cost for a full nonlinear solution is prohibitive. Therefore, in our implementation we do not pursue the elimination of such residual overlaps. Instead, we focus on the stability of temporal integration, i.e., the temporal integration of trajectory is stable even if very large forces suddenly appear on some particles due to, for example, Brownian noise or a large number of doubly bound crosslinkers. We have also benchmarked our algorithm such that the average physical properties of the entire suspension converge to the reference values. For example, our method accurately captures the system stress, the equation of state and isotropic-nematic phase transition of rigid Brownian spherocylinders *Yan et al., 2019*.

## Symmetry of the geometrically constrained optimization problem

We briefly prove the symmetry of *Equation 51*. The derivation in this section is applicable to rigid particles with arbitrary shapes.

The configuration of each particle is tracked by its center location $x$ in the lab frame and its orientation as a unit quaternion $\theta = [s, p] \in \mathbb{R}^4$. For an arbitrary 3D vector $Y$ which is attached to a particle and follows the particle's motion, its image $y$ in the lab frame following the particle's rotation is:

$$y = RY. \tag{55}$$

where $\boldsymbol{R} \in \mathbb{R}^{3\times3}$ denotes the rotation matrix generated by the unit quaternion $\boldsymbol{\theta}$.

For both unilateral and bilateral constraints, $\mathcal{D}$ has a sparse column structure:

$$\mathcal{D} = \left[\boldsymbol{D}_{P_1Q_1}, \boldsymbol{D}_{Q_2Q_2}, \cdots\right], \tag{56}$$

where $P_i, Q_i$ are particle indices for the -th column. For example, for a system with 4 particles $0, 1, 2, 3$ and two possible collision pairs $0, 1$ and $1, 3$, the $\mathcal{D}_u$ matrix for collision (unilateral) constraints is:

$$\mathcal{D} = \left[\boldsymbol{D}_{0,1}, \boldsymbol{D}_{1,3}, \cdots\right]. \tag{57}$$

Because of this structure, to prove *Equation 51* we only need to prove the equality $\boldsymbol{D}_{PQ} = \nabla_{\mathcal{C}}\boldsymbol{\Phi}_{PQ}\mathcal{G}$ for a pair of particles $P, Q$, as shown in *Appendix 3—figure 1*.

We consider two rigid particles centered at $\boldsymbol{x}_P, \boldsymbol{x}_Q$, each has a point fixed on the body (not necessarily on the surface). $\boldsymbol{y}_P$ and $\boldsymbol{y}_Q$ are vectors in the lab frame from the particle centers to the points. The distance between these two points follows the rigid body motion of both particles:

$$\boldsymbol{r} = \boldsymbol{x}_P + \boldsymbol{y}_P - \boldsymbol{x}_Q - \boldsymbol{y}_Q = \boldsymbol{x}_P + \boldsymbol{R}_P\boldsymbol{Y}_P - (\boldsymbol{x}_Q + \boldsymbol{R}_Q\boldsymbol{Y}_Q), \tag{58}$$

where $\boldsymbol{R}_P$ and $\boldsymbol{R}_Q$ are the well-known rotation matrices. $\boldsymbol{Y}_P$ and $\boldsymbol{Y}_Q$ are locations of those two points in their intrinsic coordinate systems. $\Phi_{PQ} = |\boldsymbol{r}|$ is simply the distance between the two points, dependent on the motion of the two rigid particles.

According to our definition, $\boldsymbol{D}_{PQ}$ maps the force magnitude γ between the two particles to force and torque vectors on each particle:

$$\boldsymbol{D}_{PQ} = [\boldsymbol{e}_P, \boldsymbol{y}_P \times \boldsymbol{e}_P, \boldsymbol{e}_Q, \boldsymbol{y}_Q \times \boldsymbol{e}_Q]^T, \tag{59}$$

$$\boldsymbol{e}_P = \boldsymbol{r}/|\boldsymbol{r}| = -\boldsymbol{e}_Q \tag{60}$$

$(\nabla_{\mathcal{C}}\boldsymbol{\Phi})\,\mathcal{G}$ can also be explicitly written as follows:

$$\nabla_{\mathcal{C}}\boldsymbol{\Phi}_{PQ}\mathcal{G} = \begin{bmatrix} \partial\Phi/\partial\boldsymbol{x}_P \\ \partial\Phi/\partial\boldsymbol{\theta}_P \\ \partial\Phi/\partial\boldsymbol{x}_Q \\ \partial\Phi/\partial\boldsymbol{\theta}_Q \end{bmatrix} \begin{bmatrix} \boldsymbol{I}^3 & & & \\ & \boldsymbol{\Psi}_P & & \\ & & \boldsymbol{I}^3 & \\ & & & \boldsymbol{\Psi}_Q \end{bmatrix} \tag{61}$$

Further, we notice the symmetry of P and Q in the above equations of $D_{PQ}$ and $\nabla_{\mathcal{C}}\boldsymbol{\Phi}_{PQ}\mathcal{G}$, we only need to prove the following equality for P:

$$\begin{bmatrix} \partial\Phi/\partial\boldsymbol{x}_P \\ \partial\Phi/\partial\boldsymbol{\theta}_P \end{bmatrix} \begin{bmatrix} \boldsymbol{I}^3 & \\ & \boldsymbol{\Psi}_P \end{bmatrix} = \begin{bmatrix} \boldsymbol{e}_P \\ \boldsymbol{y}_P \times \boldsymbol{e}_P \end{bmatrix} \tag{62}$$

In *Equation 62* the only difference between unilateral and bilateral constraints are how the two points on particles P and Q are picked. For unilateral (collision) constraints, the two points are where the distance $\Phi$ reaches the minimal distance between the two particles. For bilateral constraints, there is no such restriction and the two points are arbitrary. Obviously, we only need to prove this latter case, i.e., to prove *Equation 62* when $\boldsymbol{Y}_P$ is an arbitrary vector.

The first row of *Equation 62* is straightforward because

$$\partial\Phi/\partial\boldsymbol{x}_P = \partial|\boldsymbol{r}|/\partial\boldsymbol{x}_P = \boldsymbol{r}/|\boldsymbol{r}| = \boldsymbol{e}_P \tag{63}$$

The second row can be proved as follows. We first derive some general results about quaternions and rotation matrices, dropping the subscript $P$ to simplify equations. When the particle rotates with an angular velocity $\omega$, the motion of $\boldsymbol{y}$ satisfies

$$\dot{\boldsymbol{y}} = \boldsymbol{\omega} \times \boldsymbol{y}, \quad \text{i.e.,} \quad \dot{y}_i = \frac{\partial y_i}{\partial t} = \epsilon_{i\alpha\beta}\omega_\alpha y_\beta = \epsilon_{i\alpha\beta}\omega_\alpha R_{\beta\gamma}Y_\gamma \tag{64}$$

$\dot{\boldsymbol{y}}$ can also be directly computed by applying the chain rule on *Equation 55*, because $\boldsymbol{Y}$ is intrinsic to the particle invariant over time:

$$\dot{y}_i = \frac{\partial R_{ij}}{\partial \theta_k}\dot{\theta}_k Y_j \tag{65}$$

The matrix $\Psi$ bridges angular velocity and quaternion by definition:

$$\dot{\theta}_k = \Psi_{kl}\omega_l, \quad \Psi_{kl} \in \mathbb{R}^{4\times3} \tag{66}$$

We have

$$\omega_l \frac{\partial R_{ij}}{\partial \theta_k}\Psi_{kl}Y_j = \epsilon_{i\alpha\beta}R_{\beta\gamma}Y_\gamma \omega_\alpha \tag{67}$$

This must be valid for arbitrary $\boldsymbol{\omega}$, which is only possible when

$$\frac{\partial R_{ij}}{\partial \theta_k}\Psi_{kl}Y_j = \epsilon_{il\beta}R_{\beta\gamma}Y_\gamma \tag{68}$$

Now for another arbitrary vector $\boldsymbol{r}$:

$$r_i\frac{\partial R_{ij}}{\partial \theta_k}\Psi_{kl}Y_j = r_i\epsilon_{il\beta}R_{\beta\gamma}Y_\gamma = \epsilon_{l\beta i}R_{\beta\gamma}Y_\gamma r_i = \left[(\boldsymbol{RY})\times\boldsymbol{r}\right]_l \tag{69}$$

Using *Equation 69* we can prove the second row of *Equation 62*. We first calculate the derivatives of *Equation 62* using dummy indices:

$$\frac{\partial \Phi}{\partial \theta_k} = \frac{\partial \Phi}{\partial r_i}\frac{\partial r_i}{\partial \theta_k} = \frac{1}{\Phi}r_i\frac{\partial r_i}{\partial \theta_k} = \frac{1}{\Phi}r_i\frac{\partial R_{ij}}{\partial \theta_k}Y_j \tag{70}$$

Multiply the matrix $\Psi_{kl}$ on both sides:

$$\frac{\partial \Phi}{\partial \theta_k}\Psi_{kl} = \frac{1}{\Phi}r_i\frac{\partial R_{ij}}{\partial \theta_k}Y_j\Psi_{kl} \tag{71}$$

Substitute the right side by *Equation 69*, we get:

$$\frac{\partial \Phi}{\partial \theta_k}\Psi_{kl} = \frac{1}{\Phi}\epsilon_{l\beta i}R_{\beta\gamma}Y_\gamma r_l = \frac{1}{\Phi}\left[(\boldsymbol{RY})\times\boldsymbol{r}\right]_l. \tag{72}$$

This is exactly the right side of *Equation 62* because by definition $\boldsymbol{y}_P = \boldsymbol{R}_P\boldsymbol{Y}_P$ and $\boldsymbol{e}_P = \boldsymbol{r}/\Phi$. Therefore *Equation 62* holds and the equality *Equation 51* holds.

## Implementation

As mentioned above, at each timestep we first update the crosslinkers and then the filaments. We implement the two steps in a fully parallelized C ++ codebase, utilizing MPI and OpenMP and scalable to hundreds of CPU cores.

In the crosslinker-update step, we have assumed that every crosslinker has binding-unbinding probabilities independent of other crosslinkers. Therefore, it is straightforward to parallelize this step, we only need to search the vicinity of each crosslinker to find the candidate filaments that this crosslinker may bind to. This can be conveniently accomplished by a standard near neighbor detection operation based on bounding volume hierarchy *Iwasawa et al., 2016*, where the search radius is determined by the maximum stretch of each crosslinker. Once the candidate filaments for each crosslinker have been found, we compute the k-MC probabilities using a precomputed lookup table with interpolation to speed up the numerical integration while maintaining accuracy. This step is also parallel on all CPU cores.

After the positions of crosslinkers have been updated, we update the set of bilateral constraints $\Phi_b$ in the constraint solver. If one crosslinker has changed its status from doubly bound to singly

bound, the corresponding constraint is removed from $\Phi_b$, and vice versa. The geometric matrix $\mathcal{D}_b$ is also updated according to the current geometry, that is, those locations $y_P, y_Q$ and norm vectors $\hat{e}_P, \hat{e}_Q$. Then, a near neighbor detection operation is performed for all filaments to determine the unilateral constraints $\Phi_u$ and its geometry $\mathcal{D}_u$. If two filaments are far away from each other, there is no need to include this pair in the constraint solver because it is impossible for them to collide within this time step $\Delta t$. Therefore, we include only close pairs whose minimal distance is below some threshold value $\delta_c$. $\delta_c$ is controlled by system dynamics, that is, how far each filament may move within each timestep. Empirically, we take $\delta_c$ to be the diameter of each filament.

Once the constraint problem *Equation 48* has been constructed, we run a fully parallel iterative Barzilai-Borwein Projected Gradient Descent (BBPGD) solver *Yan et al., 2019* to solve for constraint forces $\gamma_b$ and $\gamma_u$, together with the velocities $\mathcal{U}_b$ and $\mathcal{U}_c$ due to constraint forces $\mathcal{F}_b$ and $\mathcal{F}_u$ by solving the equivalent CQP 53. The cost of every BBPGD iteration scales as $O(N_u + N_b)$, that is, the total dimension of the linear constraint problem. The number of iterations needed depends on the complexity of the structure. For example, if all filaments are far from each other such that almost no collisions or no doubly bound crosslinkers exist, the solution converges almost immediately. If all filaments are densely packed and many doubly bound crosslinkers form between the filaments, many iterations may be necessary. Empirically, the solution of *Equation 53* converges in a few hundred BBPGD iterative steps for common biological structures such as microtubule asters or bundles. However, each iteration of BBPGD is cheap because we only need to compute $\nabla f = M\gamma + q$. This sparse matrix-vector multiplication spmv is a well optimized standard mathematical operation. The BBPGD solver is implemented using the Trilinos package for distributed linear algebra operations. Once $\mathcal{U}_b$ and $\mathcal{U}_c$ have been solved, the filament configuration is updated and then next timestep starts.

## Appendix 4

### Performance measurements

The bundle contraction-buckling simulation runs on 2 nodes connected by Infiniband, and each node has two AMD EPYC 7742 64-Core CPUs 2.25 GHz. *Appendix 4—figure 1* shows the performance of the solver. Different from the aster formation case shown in *Appendix 4—figure 2*, computational time spent on crosslinkers is negligible. This is because as the fixed head of each dynein is permanently attached to the microtubule, we only need to update the status of the free head. Also, the free heads only experience the $S \rightleftharpoons D$ transition, which further reduces the computational cost. On the other hand, the collisions in this case is more difficult to resolve compared to the aster cases, because in nematic bundles more collisions happen and collisions may happen anywhere along the microtubule instead of only at the center of each aster. Similar to the aster case, we see that computational time for constraint solution is proportional to the number of BBPGD steps.

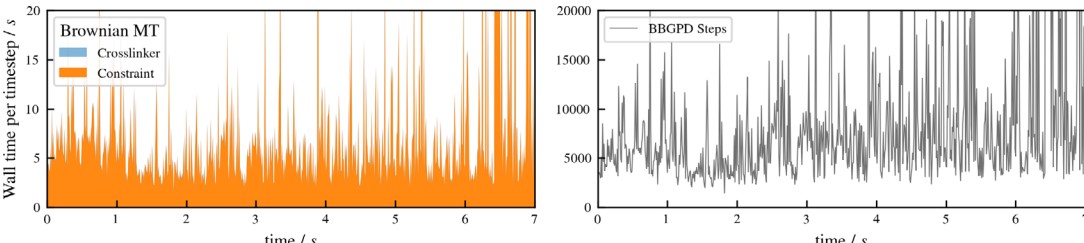

**Appendix 4—figure 1.** Performance of *aLENS* for the buckling simulation shown in *Figure 5* of main text. The left panel shows the wall clock time that every timestep takes. The right panel shows the number of BBPGD steps to solve the constraint optimization problem at every timestep.

For the aster formation in bulk problem, each case runs on 1 node of dual Intel Xeon 14-core CPUs E5-2680 v4 2.40 GHz. *Appendix 4—figure 2* shows the performance of the solver for simulations with and without thermal fluctuations. Updating the binding states of kinesin-5 motors requires roughly the same wall clock time per timestep for the entire simulation. However, the time required to solve the constraint problem grow quickly in the initial stage. The solver cost increases mostly due to the increased number of BBPGD steps (as shown in the right panels of *Appendix 4—figure 2*) even though the dimension of the constraint problem *Equation 53* grows as more kinesin-5 motors become doubly bound and more collisions occur as the asters form. The increase in BBPGD steps dominates because while the dimension of $\gamma$ increases, the dimension of $\mathcal{M}$ remains constant since the number of microtubules does not change and the cost of each BBPGD step mainly depends on the cost of applying $\mathcal{M}$ when calculating $\nabla f$ in solving *Equation 53*.

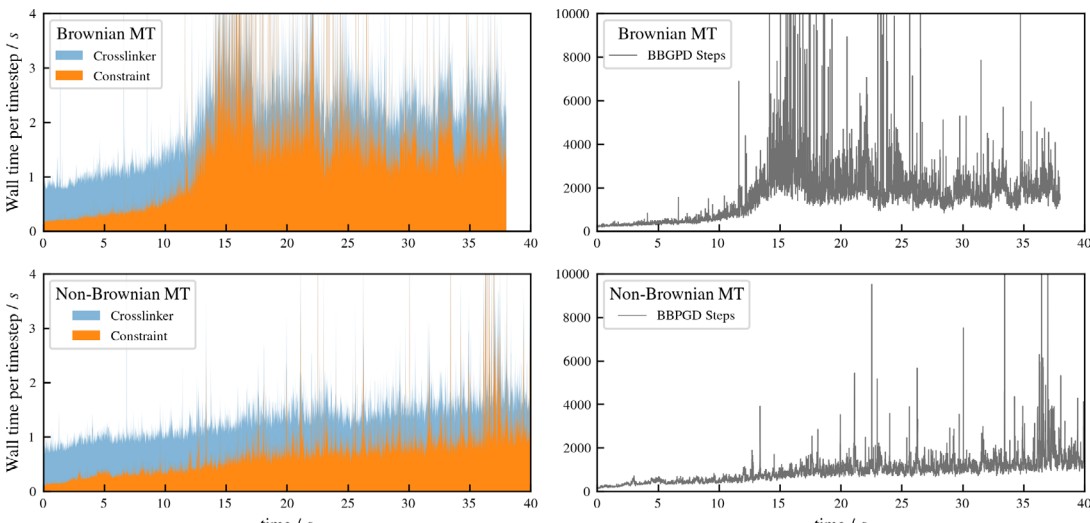

**Appendix 4—figure 2.** Performance of *aLENS* for aster formation simulations shown in *Figure 7* of main text. The left panels show the wall clock time that every timestep takes to simulate the Brownian and Non-Brownian cases. The right panels show the number of BBPGD steps to solve the constraint optimization problem at every timestep for those two cases.

## Appendix 5

### Aster center analysis of asters formation in bulk

This section provides more details about the simulation in Section Confined filament-motor protein assemblies of main text.

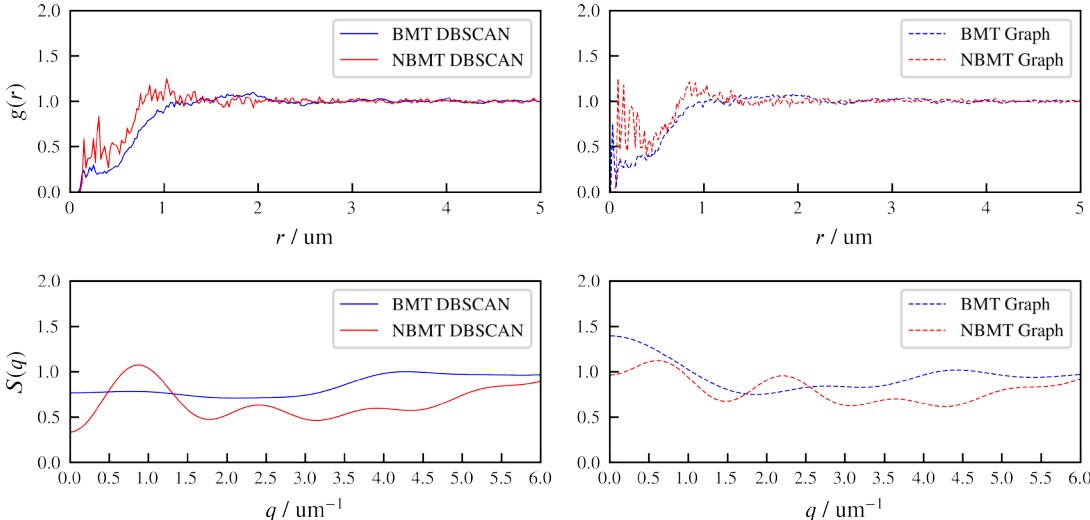

**Appendix 5—figure 1.** The radial distribution function $g(r)$ and structure factor $S(q)$ of identified aster centers at steady state, for BMT and NBMT cases. 'DBSCAN' and 'Graph' refer to two different methods of identifying aster centers, based on spatial locations of all microtubule minus ends, and the crosslinking connectivity, respectively. 500 snapshots at simulation steady state are used to compute $g(r)$ and $S(q)$, for each case.

To quantify the spatial aster center distribution, we identify aster centers for each snapshot of data. For cross validation, we use two different methods to identify the aster centers: 'DBSCAN' and 'Graph'. The implementation details are discussed in the following. Once aster centers are identified, we compute the radial distribution function $g(r)$. Then, we compute structure factor $S(q)$ based on $g(r)$ as

$$S(q) = 1 + 4\pi\rho\frac{1}{q}\int r\sin qr\left[g(r) - 1\right]dr,\tag{73}$$

because the structure of aster centers is isotropic and the orientation of $q$ does not matter.

*Appendix 5—figure 1* summarizes the results for BMT and NBMT systems. Both 'DBSCAN' and 'Graph' methods generate similar results. According to $S(q)$, there is a clearly length scale for the NBMT case at $q \approx 0.8\,\mu m^{-1}$. This reflects the spacing between individual asters at approximately $1.2\,\mu m$. This length scale is straightforward to understand. Since we used microtubules of length $0.5\,\mu m$, if two aster centers are smaller than $2L$, then the edge of two asters may touch or overlap, and are likely to be crosslinked by kinesin-5 motors and gradually merge into one bigger aster. With this length scale argument, we can estimate the total number of asters in the simulation box to be $(L_{\mathrm{box}}/(2L))^3 \approx 10^3$. This simple estimation agrees with our aster center identification results, which on average 1200 aster centers are found for each snapshot of steady state configuration.

The BMT case does not show such a significant special length scale in $S(q)$, but they do show larger spacing between asters according to $g(r)$, compared to the NBMT case. This agrees with the snapshots shown in *Figure 7* in the main text, where asters are larger but more distributed in space. Both methods identified on average 280 aster centers for each snapshot of steady state configuration.

### Identify aster centers by DBSCAN method

DBSCAN stands for *Density-Based Spatial Clustering of Applications with Noise* and is a method to identify clusters from points in space. With a given distance $\epsilon$ and a threshold $N_{\min}$ of minimal number of points, DBSCAN searches all clusters such that each cluster has no less than $N_{\min}$ points and no point in one cluster is more than distance $\epsilon$ separated from other points in the same cluster.

To apply DBSCAN, we first create a point cloud using the location of all microtubule minus ends in the system, and then run the algorithm using the function cluster.dbscan from the python package scikit-learn. Once clusters have been identified, we compute the aster centers by averaging the location of all points in each cluster.

We set $\epsilon = 100$ nm, because according to *Figure 7* in main text, the minus ends of microtubules are separated roughly $25 + 53$ nm. We also set $N_{\min} = 5$.

## Identify aster centers by Graph method

The entire microtubule-kinesin system can be abstracted as an undirected graph, where each microtubule is a node marked by their index and each doubly bound kinesin form an edge. Then, one aster is simply abstracted as a connected component of the graph. We use the `connected_components()` function in the python package networkx to find all such connected components, with minimal number of microtubules $N_{\min} = 5$. We identify aster centers by computing the average location of minus ends of these connected components.

## Appendix 6

### Confined filament-motor protein assemblies

This section provides more details about the simulation in Section Confined filament-motor protein assemblies of main text. We simulate 9,216 microtubules and 27,648 crosslinking motor proteins in a cylindrical volume. Microtubules are modeled as rigid spherocylinders with length $0.25\,\mu m$ and diameter $25\,\text{nm}$ (aspect ratio of $10$). Crosslinking motor proteins are modeled as Hookean springs. The cylinderical axis is oriented along the $+x$ direction, with a periodic boundary condition. The radial direction has a hard confinement boundary. System temperature is fixed at $300\,\text{K}$ and the simulation timestep is $10^{-4}\,\text{s}$, with the system configuration recorded every 500 steps. Solvent viscosity is set at $0.01\,\text{pN}\mu\text{m}^{-2}\text{s}$. Values for the cylinder diameter, $D_{cyl} \in \{0.25, 0.75\}\,\mu m$ are chosen to disrupt the self-assembly of an ideal aster. Initially, microtubules were aligned along the $x$ direction (cylinder axis) such that the initial nematic order parameter, $S = \langle \frac{1}{2}(3\cos^2\theta - 1)\rangle$, was 1. Here, $\theta$ is the angle between the microtubule orientation vector and the $+x$ axis, and $\langle . \rangle$ denotes an average over all microtubules. Equal numbers of microtubules are oriented in the $+x$ and the $-x$ direction such that the polar order parameter, $P = \langle \cos\theta \rangle = 0$.

### Structural quantification

To measure the structure of our steady-states, we compute the local packing fraction $\phi_{\text{local}}(x)$, local nematic order, local crosslinker density, and pair distribution functions. For the first three quantities, we start by dividing the volume into cylindrical bins with their axis in the $+x$ direction. The diameter of the bins is equal to $D_{cyl}$, and the height is chosen as $25\,\text{nm}$. The local packing fraction is computed by calculating the cumulative volume of all microtubules that fall inside each bin, and then dividing by the bin volume. For simplicity, we treat the filaments as cylinders (such that there as no hemispherical caps at their ends). For the local nematic order parameter $S_{\text{local}}^x(x)$, we find the total number of microtubules, $N(x)$, that pass through each bin at some location $x$. For each microtubule in the bin, we compute it's individual contribution to the local nematic order parameter, $S_{\text{local}}^x(x)_i = \frac{1}{2}(3\cos^2\theta_i - 1)$. We weight each $S_{\text{local}}^x(x)_i$ by a factor $W_i(x)$ that depends on the length of microtubule that falls inside the bin, normalized by the cumulative length of all other microtubules that traverse the bin. We calculate the local nematic order parameter as

$$S_{\text{local}}^x(x) = \sum_i^{N(x)} W_i(x) S_{\text{local}}^x(x)_i$$

Local crosslinker density, $C(x)$, is found by counting the number of center points of crosslinking motor proteins that fall in each bin, and then dividing by the bin volume. For this calculation, we only consider doubly-bound crosslinking motor proteins. Finally, we compute the pair distribution functions by finding the distance (using the nearest image convention in $x$) of all microtubules from a single reference microtubule. Repeating this for all microtubules as a reference, and averaging yields a pair distribution function. The useful dimensions here are $x$ and $\rho = \sqrt{y^2 + z^2}$. Due to the non-periodic nature in $\rho$, this pair distirbution function does not decay to 1.

$$D_{cyl} = 0.25\,\mu m$$

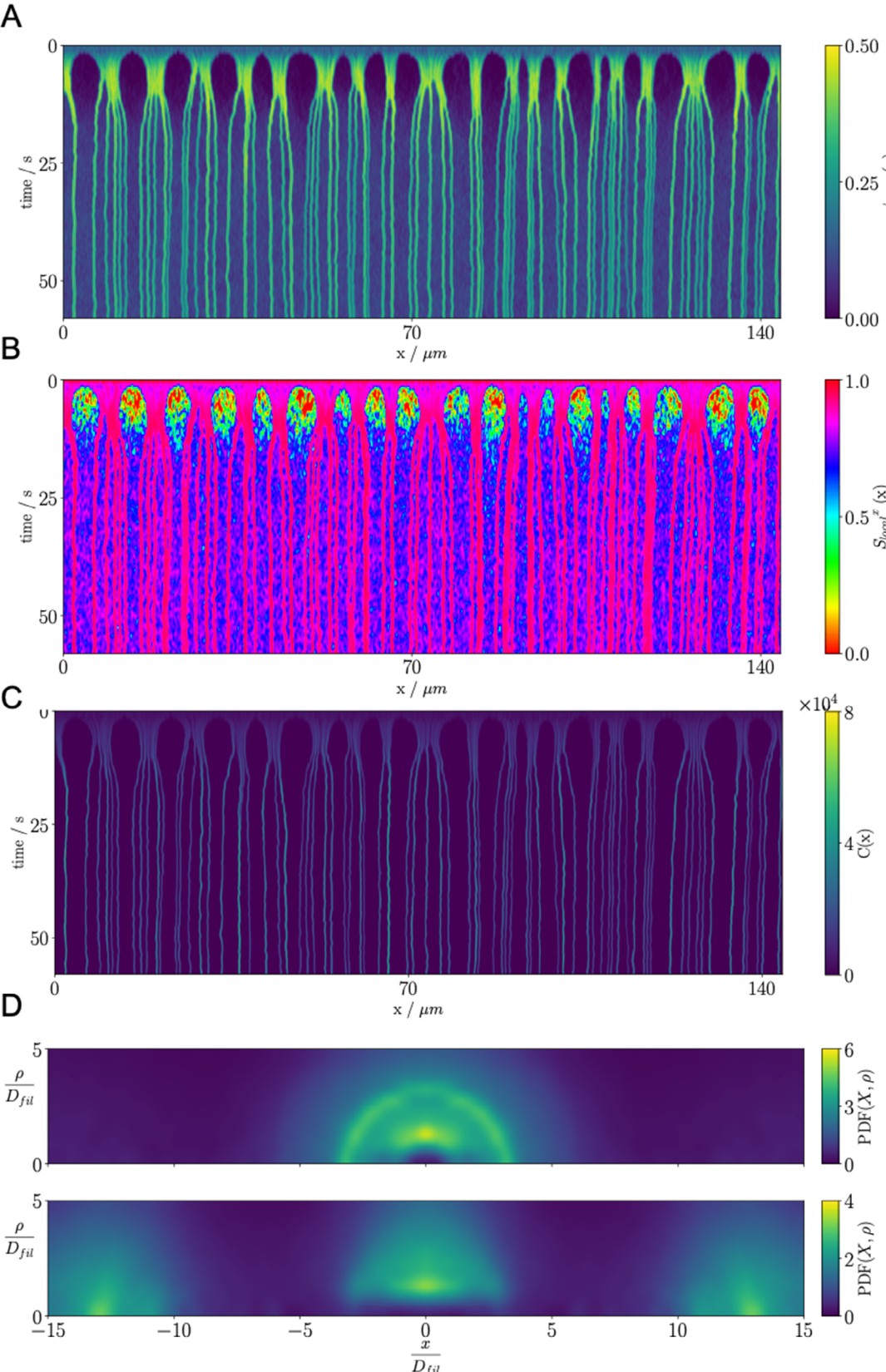

**Appendix 6—figure 1.** Results for the confined microtubule-motor protein assembly simulations with $D_{cyl} = 0.25\,\mu m$. (**A**) A kymograph of the local microtubule packing fraction $\phi_{local}(x)$. Initially, crosslinking motor
*Appendix 6—figure 1 continued on next page*

*Appendix 6—figure 1 continued*

proteins drive contraction of the system into condensed regions that break into PSBs over time. (**B**) A kymograph of the local nematic order parameter $S^x_{\text{local}}(x)$. (**C**) A kymograph of the density of the crosslinking motor proteins, $C(x)$. Condensation of microtubules coincides with condensation of the crosslinking motor proteins. (**D**) Pair distribution function for microtubule plus-ends (top) and microtubule centers (bottom).

The simulation volume is a cylinder with height $144\,\mu m$. We measure structural properties of the system over the course of the simulation. A kymograph of the local packing fraction is shown in *Appendix 6—figure 1*. The local nematic order (*Appendix 6—figure 1B*) shows that the polarity-sorted bilayers (PSBs) have a maximum order parameter equal to 1 The condensation of microtubules is mediated by the crosslinking motor proteins. In *Appendix 6—figure 1C*, we show a kymograph of the local density of the crosslinking motor proteins.

The microtubule pair distribution function at steady-state (*Appendix 6—figure 1D*) shows that plus-ends (top plot) are distributed in a ring. The ring radius is set by the length of a single crosslinking motor protein. There is negligible density away from the ring. In contrast to asters (that contain microtubules isotropically distributed around a core), microtubule centers (bottom plot) are distributed in vertically extended regions. Separation between these regions is determined by the sum of the microtubule length and the length of the crosslinking motor protein. The presence of three regions in this pair distribution plot is evidence for a pair of layers.

$$D_{cyl} = 0.75\,\mu m$$

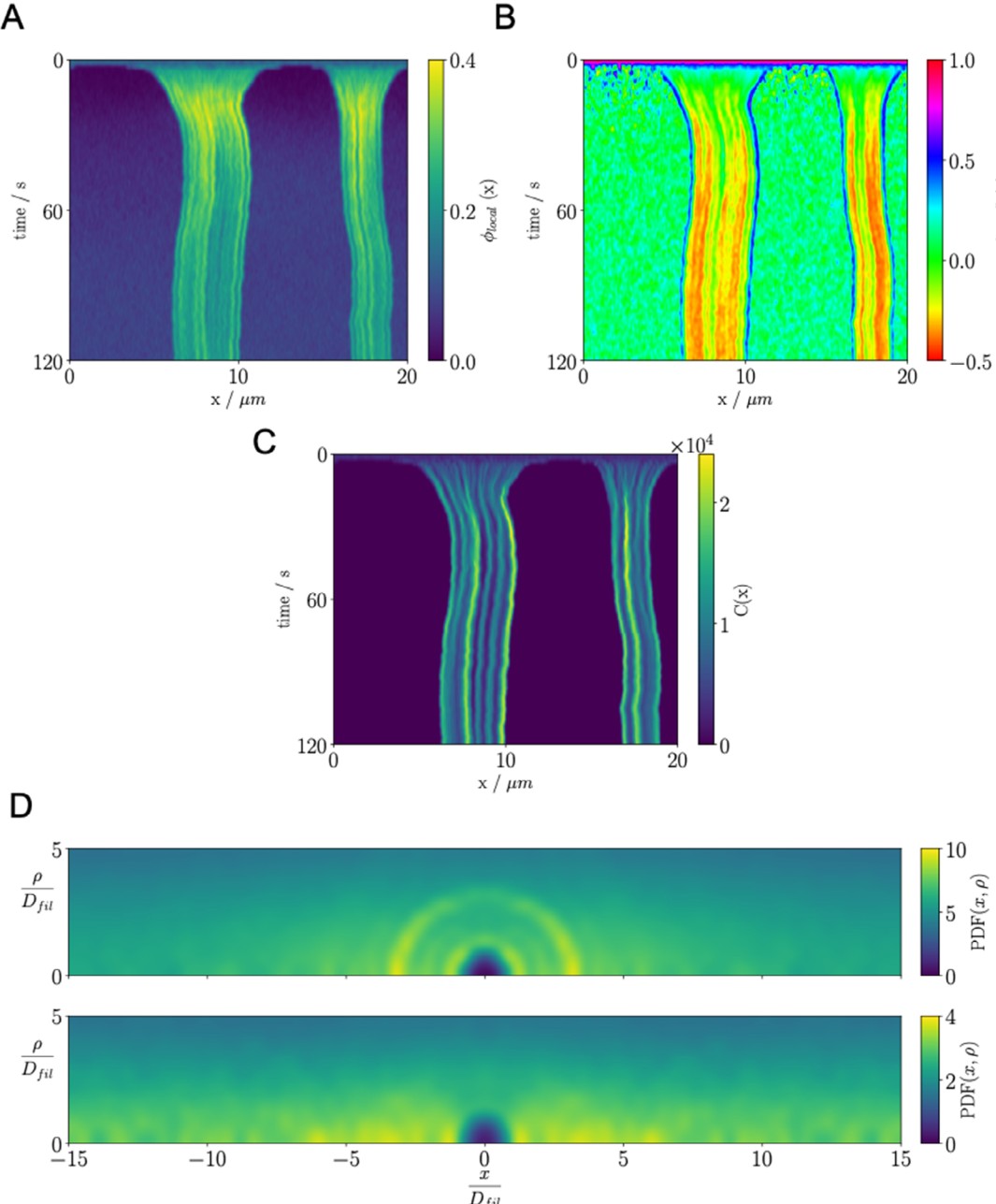

**Appendix 6—figure 2.** Results for the confined microtubule-motor protein assembly simulations with $D_{cyl} = 0.75\,\mu m$. (**A**) A kymograph of the local microtubule packing fraction $\phi_{local}(x)$. Crosslinking motor proteins drive contraction of the system. Self organization of these regions leads to emergence of the BB-like state. (**B**) A kymograph of the local nematic order parameter $S^x_{local}(x)$. The negative order parameter suggests that there is alignment of microtubules in the radial direction ($YZ$ plane). (**C**) A kymograph of the density of the crosslinking motor proteins, $C(x)$. (**D**) Pair distribution function for microtubule plus-ends (top) and microtubule centers (bottom).

In this case, the simulation volume is a cylinder with height $20\,\mu m$. Over time, microtubules condense into a bottlebrush-like (BB) state with a hedgehog line defect. This consists of microtubules having a degree of alignment in the radial direction. The ends of the BB state contain a half-aster. Crosslinking motor proteins are highly concentrated along the central axis of the BB. Here is a kymograph for the local microtubule packing fraction, $\phi_{local}(x)$ (**Appendix 6—figure 2A**). We show the evolution of the local nematic order parameter, $S^x_{local}(x)$, in **Appendix 6—figure 2B**. A negative $S^x_{local}$ indicates a significant degree of radial alignment. Maximum radial alignment (the ideal bottlebrush

state) is evidenced by a nematic order parameter value of $0.5$ The condensation of microtubules is mediated by crosslinking motor proteins. In *Appendix 6—figure 2C*, we show a kymograph of the local density of the crosslinking motor proteins. *Appendix 6—figure 2D* depicts the microtubule pair distribution function. While there is a ring clearly visible for microtubule plus-ends (top plot), showing that this state is aster-like, there is significant density present along the X axis. This indicates that there is an accumulation of plus-ends throughout the line defect. Microtubule centers (bottom plot) are distributed uniformly along $x$ while there is a decay in density along $\rho$. The absence of a ring indicates that this state is not aster-like. High density at $\rho = 0$ suggests that microtubule centers tend to be stacked in $x$.

## Ideal bottle-brush state

The ideal bottle-brush state (BB) consists of microtubules aligned in the radial direction directed away from a central line defect. A schematic and different views are shown in *Appendix 6—figure 3*. Microtubule orientation is indicated by the color wheel. For such a state, the local nematic order parameter along $x$ has a value of $-0.5$ along the length of the BB.

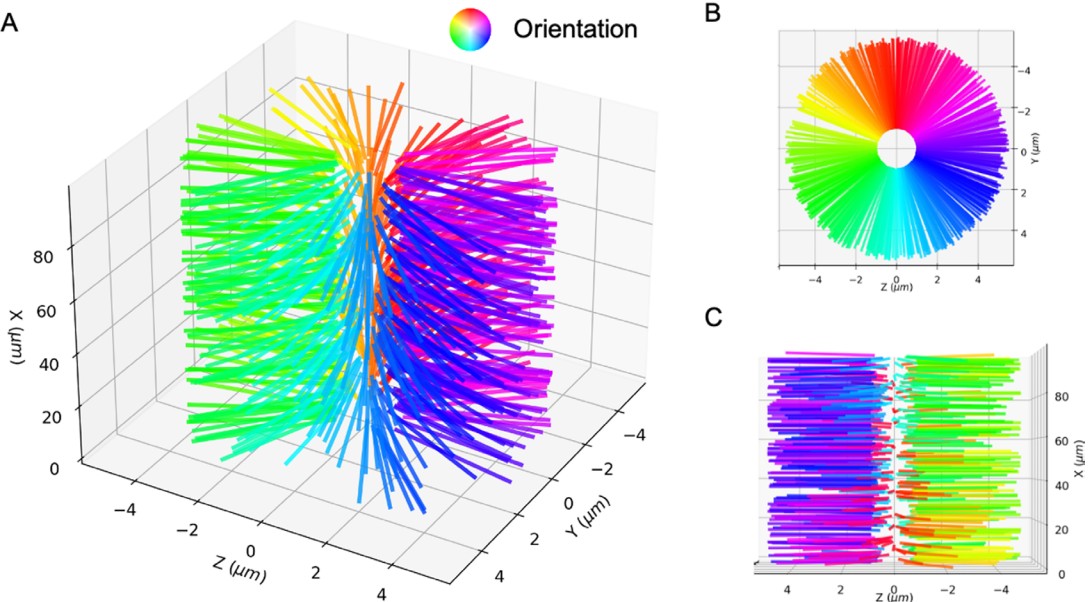

**Appendix 6—figure 3.** The perfect bottle-brush state. Microtubules are aligned in the $XY$ planes such that there is a line defect along the $z$ axis. (**A**) 3D view. (**B**) Side view. (**C**) Top view. Microtubule orientation is shown by the color wheel.

# Appendix 7

## Bending rigidity

A flexible long fiber can be implemented by connecting short rigid segments into chains. The key is how to properly implement the force and torque induced by deformation at the rigid segment joints. There are two ways to implement this, which we shall detail in the following. The first method implements the deformation of each joint with two linear Hookean springs and requires no modification to the current codebase. The second method directly incorporates the bending rigidity as a new set of constraints in the geometric constraint minimization solver, but requires some extensions to the current codebase.

## Method 1: use two Hookean springs

**Appendix 7—table 1.** The parameters of the two springs controlling extension and bending, respectively.

The relation between $\ell_B^0$ and $\ell_E^0$ determine the equilibrium configuration of the two connected filaments. When $\ell_B^0 \geq \ell_E^0 + 2d_B$, the straight configuration is the preferred configuration.

| Role | spring stiffness constant | free length |
|---|---|---|
| Bending | $\kappa_B$ | $\ell_B^0$ |
| Extension | $\kappa_E$ | $\ell_E^0$ |

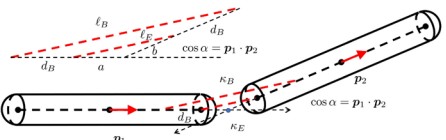

**Appendix 7—figure 1.** The geometry of two short rigid straight fibers connected at a bending joint. The separation is exaggerated to clearly show the geometry. $\boldsymbol{p}_1$ and $\boldsymbol{p}_2$ are the orientation norm vectors of the two segments. $\kappa_E$ and $\kappa_B$ are the stiffness of the spring for extension and the spring for bending. $d_B$ is the displacement distance from the joint rotation center. In the more detailed view of the deformed geometry of the two springs, $a, b$ are the lengths of the two edges of the triangle. $\ell_E = \sqrt{a^2 + b^2 + 2ab \cos \alpha}$. $\ell_B = \sqrt{(a + d_B)^2 + (b + d_B)^2 + 2(a + d_B)(b + d_B) \cos \alpha}$.

We can use two permanently bound springs for each joint, as shown in **Appendix 7—figure 1**, to implement the bending rigidity. The separations in the figure is exaggerated to show the geometry clearly. The energy of the two springs depend on their lengths $\ell_E, \ell_B$ geometrically:

$$U = \tfrac{1}{2} \kappa_E (\ell_E - \ell_E^0)^2 + \tfrac{1}{2} \kappa_B (\ell_B - \ell_B^0)^2 \tag{74}$$

With the deformed geometry, the lengths of the two springs are:

$$\ell_E = \sqrt{a^2 + b^2 + 2ab \cos \alpha} \tag{75}$$

$$\ell_B = \sqrt{(a + d_B)^2 + (b + d_B)^2 + 2(a + d_B)(b + d_B) \cos \alpha} \tag{76}$$

When $\alpha \to 0$, the energy $U$ of the two springs can be expanded as:

$$
\begin{aligned}
U \;=\; & \tfrac{1}{2}\left(\kappa_B(a+b+2d_B-\ell_B^0)^2 + \kappa_E(a+b-\ell_E^0)^2\right) \\
& + \left[\frac{\kappa_B(-a-d_B)(b+d_B)(a+b+2d_B-\ell_B^0)}{2(a+b+2d_B)} - \frac{ab\kappa_E(a+b-\ell_E^0)}{2(a+b)}\right]\alpha^2 \\
& + \tfrac{1}{24}\kappa_B\left(\frac{(a+d_B)(b+d_B)\left(a^2+d_B a(a+b)-ab+b^2+d_B^2\right)(a+b+2d_B-\ell_B^0)}{(a+b+2d_B)^3} + \frac{3(a+d_B)^2(b+d_B)^2}{(a+b+2d_B)^2}\right)\alpha^4 \\
& + \tfrac{1}{24}\kappa_E\left(\frac{ab(a^2-ab+b^2)(a+b-\ell_E^0)}{(a+b)^3} + \frac{3a^2b^2}{(a+b)^2}\right)\alpha^4 \\
& + O(\alpha^6).
\end{aligned}
\tag{77}
$$

Here in the first term is simply the linear extension of both springs when α is small. The two-spring system generate a equivalent extensional rigidity $\kappa_B + \kappa_E$. The second $\alpha^2$ term governs the bending energy. We can tune the five parameters $\ell_E^0, \ell_B^0, d_B, \kappa_E, \kappa_B$ such that the connected segments reproduce the desired mechanical behavior of a flexible filament. Although the expansion *Equation 77* is general and can be fitted to many different models by tuning the five parameters, it is too complicated to be conveniently used in an actual simulation. In the following we discuss simpler special cases which are more relevant to biological filaments.

Special case 1 When model some bio-filaments such as microtubules, we sometimes assume filaments are inextensible, i.e., $\kappa_E = \infty$ and $\ell_E = \ell_E^0$. In this special case, the energy of the two springs depends only on $U = \tfrac{1}{2}\kappa_B(\ell_B - \ell_B^0)^2$. By imposing $\ell_E = \ell_E^0$, we can solve for $b$:

$$
b = \tfrac{1}{2}\left(\sqrt{2}\sqrt{a^2\cos(2\alpha)-a^2+2\ell_E^{0^2}} - 2a\cos(\alpha)\right)
\tag{78}
$$

Then in this case $U$ depends on $\alpha^4$ in the limit of $\alpha \to 0$. To simplify the notations of the expansion, we define: $s = \ell_B^0 - \ell_E^0 - 2d_B$.

The value $s$ defines three cases of the equilibrium configuration:

- $s > 0$. The equilibrium configuration of the joint is a straight line, and the bending spring is compressed at equilibrium.
- $s = 0$. The equilibrium configuration of the joint is a straight line, and the bending spring is not compressed nor stretched at equilibrium.
- $s < 0$. The equilibrium configuration of the joint is bent.

For the first two cases, the equilibrium configuration is a straight line and we can expand $U$ in the limit of $\alpha \to 0$:

$$
\begin{aligned}
U \;=\; & \frac{\kappa_B s^2}{2} \\
& + \frac{d_B\kappa_B\left(2a^2-2a\ell_E^0+\ell_E^0(d_B+\ell_E^0)\right)}{2\ell_E^0(2d_B+\ell_E^0)}s\alpha^2 \\
& + \frac{d_B^2\kappa_B\left(2a^2-2a\ell_E^0+\ell_E^0(d_B+\ell_E^0)\right)^2}{8\ell_E^{0^2}(2d_B+\ell_E^0)^2}\alpha^4 \\
& + \left[\frac{3a^4 d_B^2\kappa_B}{2\ell_E^{0^2}(2d_B+\ell_E^0)^3} + \frac{a^4 d_B^3\kappa_B}{\ell_E^{0^3}(2d_B+\ell_E^0)^3} - \frac{a^3 d_B^2\kappa_B}{\ell_E^0(2d_B+\ell_E^0)^3} - \frac{4a^2 d_B^2\kappa_B}{3(2d_B+\ell_E^0)^3} - \frac{11a^2 d_B^3\kappa_B}{6\ell_E^0(2d_B+\ell_E^0)^3}\right. \\
& \left. + \frac{a^4 d_B\kappa_B}{4\ell_E^0(2d_B+\ell_E^0)^3} - \frac{7a^2 d_B\kappa_B\ell_E^0}{12(2d_B+\ell_E^0)^3} + \frac{5ad_B^3\kappa_B}{6(2d_B+\ell_E^0)^3} + \frac{5ad_B^2\kappa_B\ell_E^0}{6(2d_B+\ell_E^0)^3} + \frac{ad_B\kappa_B\ell_E^{0^2}}{3(2d_B+\ell_E^0)^3}\right. \\
& \left. - \frac{d_B^2\kappa_B\ell_E^{0^2}}{12(2d_B+\ell_E^0)^3} - \frac{d_B^3\kappa_B\ell_E^0}{12(2d_B+\ell_E^0)^3} - \frac{d_B^4\kappa_B}{24(2d_B+\ell_E^0)^3} - \frac{d_B\kappa_B\ell_E^{0^3}}{24(2d_B+\ell_E^0)^3}\right]s\alpha^4 \\
& + O(\alpha^6).
\end{aligned}
\tag{79}
$$

With this form, it is clear that the bending energy is tunable with the parameter $s$, that is, how much the bending spring is strained in the equilibrium configuration. Note that $s$ here is a constant determined by the lengths $\ell_E, \ell_B, d_B$ only. Therefore, the first term $frac12\kappa_B s^2$ only 'shifts' the zero-point of the energy. This term does not contribute to the stretching or bending energy of the joint. When $s = 0$, the leading order terms all vanish and $U(\alpha) \propto \alpha^4$. When $s > 0$, the leading order terms are non-zero and the energy is asymptotically a quadratic function of α: $U(\alpha) \propto \alpha^2$.

Special case 2 If we further assume that $\kappa_E = \infty$ and $\ell_E = \ell_E^0 = 0$, we have that $a = b = 0$. This means the extension spring degenerates into a point joint between the two segments. In this case the energy $U$ can be further simplified:

$$
U = \tfrac{1}{2}\kappa_B\left(-\sqrt{2d_B\cos\alpha(d_B+\ell_E^0)+2d_B^2+2d_B\ell_E^0+\ell_E^{0^2}}+2d_B+\ell_E^0+s\right)^2
\tag{80}
$$

The expansion of $U$ as $\alpha \to 0$ is also further simplifed:

$$
\begin{aligned}
U &= \frac{\kappa_B s^2}{2} + \frac{d_B \kappa_B s(d_B + \ell_E^0)}{2(2d_B + \ell_E^0)} \alpha^2 \\
&+ \frac{d_B \kappa_B (d_B + \ell_E^0)\left(3d_B(d_B + \ell_E^0)(2d_B + \ell_E^0) - s\left(d_B^2 + d_B \ell_E^0 + \ell_E^{0^2}\right)\right)}{24(2d_B + \ell_E^0)^3} \alpha^4 \\
&+ O(\alpha^6)
\end{aligned}
\tag{81}
$$

Here we have the same conclusion as the previous special case, that the dependence of $U$ on α can be tuned between $\alpha^4$ and $\alpha^2$ by choosing a proper value of $s$.

## Method 2: use bilateral constraints

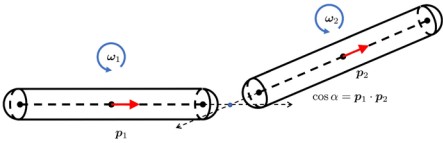

**Appendix 7—figure 2.** The geometry of two short rigid straight fibers connected at a bending joint. The separation is exaggerated to clearly show the geometry. $\boldsymbol{p}_1$ and $\boldsymbol{p}_2$ are the orientation norm vectors of the two segments. $\boldsymbol{\omega}_1$ and $\boldsymbol{\omega}_2$ are the rotational angular velocities. $U_B$ is the bending energy of this joint. α is the angle from $\boldsymbol{p}_1$ to $\boldsymbol{p}_2$. E is the bending rigidity modulus.

Here we briefly derive the constraint optimization formulation for handling the bending rigidity of flexible fibers with bilateral constraints. To fit in the geometric constraint formulation, we represent a long and flexible fiber as many short rigid straight fibers chained together by joints. The linear extension of each joint can be straightfordly handled by the bilateral spring constraints as for those doubly bound motors. For the bending rigidity, we first realize that for each joint the two norm orientation vectors $\boldsymbol{p}_1$ and $\boldsymbol{p}_2$ form a plane. This plane is orthogonal to a unit norm vector

$$
\hat{\boldsymbol{T}} = \frac{\boldsymbol{p}_1 \times \boldsymbol{p}_2}{|\boldsymbol{p}_1 \times \boldsymbol{p}_2|}
\tag{82}
$$

For most relevant biological filaments, the bending rigidity is isotropic along different directions on a cross-section of the filament. In other words, the recovering torque is always co-linear with the vector $\boldsymbol{p}_1 \times \boldsymbol{p}_2$ and the recovering deformation is always in plane spanned by $\boldsymbol{p}_1$ and $\boldsymbol{p}_2$. This is important because we can simplify the deformation to in-plane rotations in the following derivations. Note that this plane can be different for each joint since each joint is handled as an independent constraint in our method.

There are different models of how the bending energy depends on the deformation, $\boldsymbol{p}_1 \cdot \boldsymbol{p}_2$.
Case 1:

$$
U_B = E(1 - \boldsymbol{p}_1 \cdot \boldsymbol{p}_2)^2.
$$

When the angle α between $\boldsymbol{p}_1$ and $\boldsymbol{p}_2$ is small, we have:

$$
U_B \approx E(1 - (1 - \alpha^2/2))^2 \propto \alpha^4.
$$

Case 2:

$$
U_B = E(1 - \boldsymbol{p}_1 \cdot \boldsymbol{p}_2).
$$

In this form when $\alpha \to 0$ the energy depends on the second order instead of the fourth order of the angle:

$$
U_B \approx E(1 - (1 - \alpha^2/2)) \propto \alpha^2.
$$

The following derivation and method still applies.

The two cases can be handled in the same way. In the following we derive the equations for the first case, where the second case only requires a simpler small α expansion in the derivation.

There is one more relation we can utilize to simplify the derivation. Assume that $\omega_1$ and $\omega_2$ have arbitrary directions, and to the first order of $\Delta t$ the orientation vectors $\boldsymbol{p}_1$ and $\boldsymbol{p}_2$ rotates within $\Delta t$:

$$\boldsymbol{p}_1 \rightarrow \boldsymbol{p}_1 + \boldsymbol{\omega}_1 \times \boldsymbol{p}_1 \Delta t \tag{83}$$

$$\boldsymbol{p}_2 \rightarrow \boldsymbol{p}_2 + \boldsymbol{\omega}_2 \times \boldsymbol{p}_2 \Delta t \tag{84}$$

Then, the bending energy after this rotation is:

$$U_B = E \left[ 1 - \boldsymbol{p}_1 \cdot \boldsymbol{p}_2 - \Delta t \left( \boldsymbol{p}_2 \cdot (\boldsymbol{\omega}_1 \times \boldsymbol{p}_1) + \boldsymbol{p}_1 \cdot (\boldsymbol{\omega}_2 \times \boldsymbol{p}_2) \right) \right]^2 \tag{85}$$

$$= E \left[ 1 - \boldsymbol{p}_1 \cdot \boldsymbol{p}_2 - \Delta t \left( \boldsymbol{\omega}_2 - \boldsymbol{\omega}_1 \right) \cdot (\boldsymbol{p}_2 \times \boldsymbol{p}_1) \right]^2, \tag{86}$$

where we have utilized the vector triple product identity:

$$\mathbf{a} \cdot (\mathbf{b} \times \mathbf{c}) = \mathbf{b} \cdot (\mathbf{c} \times \mathbf{a}) = \mathbf{c} \cdot (\mathbf{a} \times \mathbf{b}). \tag{87}$$

This means, to the first order of $\Delta t$ only the component of rotation $\omega_1$ and $\omega_2$ that is inside this plane spanned by $\boldsymbol{p}_1, \boldsymbol{p}_2$ affect the bending energy. Therefore, to the first order of $\Delta t$ we can simplify the bending rigidity problem inside this spanned plane, although in reality the filament segments have true 3D rotations.

We denote the current and next timesteps by $n$ and $n + 1$. We have, to the first order:

$$\alpha^{n+1} = \alpha^n + (\omega_2^{n+1} - \omega_1^{n+1}) \Delta t. \tag{88}$$

The rotational mobility matrix for these two rods is:

$$\mathcal{M} = \begin{bmatrix} M_1^R & 0 \\ 0 & M_2^R \end{bmatrix}, \tag{89}$$

where $M_1^R$ and $M_2^R$ are inverse of rotational drag coefficients for those two segments. The torque generated by the joint on each segment can be calculated by the derivative of bending energy $U_B$. More specifically:

$$\omega_1^{n+1} = -M_1^R T^{n+1} \hat{\boldsymbol{T}} \tag{90}$$

$$\omega_2^{n+1} = M_2^R T^{n+1} \hat{\boldsymbol{T}}, \tag{91}$$

where the scalar torque $T^{n+1}$ is:

$$T^{n+1} = -E\alpha^{n+1,3} = -E \left[ \alpha^n + (\omega_2^{n+1} - \omega_1^{n+1})\Delta t \right]^3 \tag{92}$$

$$= -E \left[ \alpha^{n,3} + 3\alpha^{n,2}\omega_2^{n+1}\Delta t - 3\alpha^{n,2}\omega_1^{n+1}\Delta t \right] \tag{93}$$

where the higher order terms in $\Delta t$ have been neglected. If the bending energy Case 2 is used, instead of $T^{n+1} \propto -E\alpha^{n+1,3}$ we have $T^{n+1} \propto -E\alpha^{n+1}$. We can replace the expansion accordingly and the derivation remains valid.

Combining all of the above, we are effectively integrating the dynamics of all rods while ensuring *Equation 90*. Skipping the timestep index $n$, we can write the result in the same way as the bilateral Hookean spring constraints as:

$$0 = \left\{ \mathcal{D}^T \begin{bmatrix} M_1^R & 0 \\ 0 & M_2^R \end{bmatrix} \mathcal{D} + \frac{1}{K} \right\} [T] + \frac{1}{3}\alpha^n \frac{1}{\Delta t} \perp T \in R, \tag{94}$$

where $K = 3E\alpha^{n,2}$ and the geometric matrix $\mathcal{D}$ defines the direction of torque on each rod:

$$\mathcal{D} = \begin{bmatrix} -\hat{\boldsymbol{T}} \\ \boldsymbol{T} \end{bmatrix}. \tag{95}$$

The left side of *Equation 94* means the motion of filament segments must satisfy the torque-deformation relation, while the right side means the torque can take any values.

*Equation 94* is mathematically identical to the Hookean spring constraints and can be incorporated in the constraint minimization problem in the same way.

We can solve this two-segment problem analytically if the constraint optimization problem contains only *Equation 94*, in the absence of collisions and Hookean springs:

$$(\omega_2 - \omega_1)\Delta t = -\frac{1}{3}\frac{M_1^R + M_2^R}{2(M_1^R + M_2^R) + \frac{1}{K}}\alpha. \tag{96}$$

This simply means that if a straight fiber is bent to angle $\alpha$, its recovering motion within each timestep is proportional to the current angle $\alpha$. More importantly, $K \to \infty$ as the bending rigidity modulus $E$ increases to infinity. In this case, $1/K \to 0$ and the above solution is still stable, and is simplified to $(\omega_2 - \omega_1)\Delta t = -\frac{1}{6}\alpha$. This means the solution to *Equation 94* has very strong temporal integration stability even when the deformation force is infinitely stiff, the same as what we discussed for the infinitely stiff Hookean spring case.

