## [Editor Report]

This article presents a new method for simulating cytoskeletal dynamics inside cells. This is an important problem in the life sciences, and the numerical methods and derived results described in the paper seem very promising to facilitate computational modelling of cell dynamics. Although the user- friendliness of the software can still be improved, the method will be of interest to a broad community of biologists and biophysicists.

---

## [Decision Letter]

**Decision letter after peer review:**

Thank you for submitting your article "*aLENS*: towards the cellular-scale simulation of motor-driven cytoskeletal assemblies" for consideration by *eLife*. Your article has been reviewed by 3 peer reviewers, and the evaluation has been overseen by a Reviewing Editor and Anna Akhmanova as the Senior Editor. The reviewers have opted to remain anonymous.

Essential revisions:

As you will see from the reports below, the referees were favourably impressed by the performance on your numerical method, which allows to stimulate a large number of filaments and motors in a reasonable time. This paper is seen as the presentation of a new tool illustrated by examples, rather than the presentation of new scientific results. The essential revisions include:

1. Software availability (all referees) To be of value, the software should be freely available and useable. This does not seem to be the case at present.

2. Benchmarking (referees 1 and 3) It is important to provide some level of validation of the algorithm, which is currently missing. This can be done by comparing some aspects of the numerical results to existing analytical models. Regarding the binding and motor activity of cross linker, some simple toy model (e.g. gliding assay for motors, bound fraction for crosslinkers) would already be very informative, but more complex properties could also be considered as in Lera-Ramirez and Nedelec , Cytoskeleton 2019 (Theory of antiparallel microtubule overlap stabilization by motors and diffusible crosslinkers). Regarding collective effect, validation of the results could follow approaches such as the ones employed in Gao etal. PRE 2015 (Multiscale modeling and simulation of microtubule-motor-protein assemblies). Performance benchmark compared to other available algorithm should also be discussed.

3. Modularity (referees 2 and 3). At present, only rigid filaments interacting with one type of crosslinker are presented. The extension flexible filaments and different types of interacting proteins are discussed as possibilities but are not implemented at the moment. Although this can be seen as the natural next step, this modularity is essential for the algorithm to be useful to the community.

*Reviewer #1:*

The study by Yan et al., developed a novel computational framework for modelling cytoskeletal cellular processes that allows for further investigations into the material properties associated with such processes. The computational methodology involves modelling cytoskeletal filaments as rigid spherocylinders while the Hookean law is employed to model crosslinkers. The computational algorithm, aLENS performs three key tasks in a sequential manner that lends itself naturally to high performance computing. To demonstrate the applicability and usefulness of aLENS, the authors present (i) self-aligning and buckling networks for a significantly large number of filaments than previously studied, and (ii) the interplay between polarity of motor walking and polarity of filaments, which seems to suggest that the ability of motors to continuously walk without end-pausing is crucial to effective polarity sorting. The authors also investigated the formation of asters, which seem to form when crosslinking motors reorganise filaments so that their minus ends are clustered and held tight by paused motors.

Strengths

1. The development of an alternative novel computational framework that offers far more flexibility, applicability and is scalable across multi-scales.

2. The methodology overcomes the timescale limitations imposed by conventional explicit time-stepping methods that are key to modelling the dynamics of the cytoskeletal filaments and motors.

3. aLENS utilizes efficiently high-performance parallel computing resources to scale to cellular scale systems.

Weakness

1. The lack of bench-marking that is associated with algorithm comparisons for performance and robustness.

2. The lack of rigorous validation of the algorithm against suitably identified grounds truths.

3. No clear demonstration of the efficiency of the algorithm and how it compares to current conventional algorithms of this nature.

4. There are no clear comparisons between predictions of the computational algorithm and experimental data or observations.

My recommendations to authors are as follows:

(i) To consider rigorous validation of the computational methodology either by using synthetic or experimental data.

(ii) To demonstrate computational efficiency and robustness of the algorithm by comparing results to current conventional methodologies.

(iii) To demonstrate efficiency and accuracy when aLENS is compared to current conventional methodologies.

*Reviewer #2:*

In this article, the authors present a new method for modeling cytoskeletal networks inside cells. In particular they model the interactions between filaments (microtubules in this case but the method is not limited to this) and motors. The main contribution appears to be in the efficiency of this method. The main challenge in this area of numerical research is handling steric interactions between the large numbers of interacting cytoskeletal filaments. Essentially, hard repulsion methods require very small timescales that can make reasonable simulations infeasible while soft repulsion approximations of these can lead to numerical issues. Their method takes a mathematically different approach to addressing this issue that may substantially speed these simulations allowing users to simulate more realistically sized systems consisting of up to 10^6^-10^7^ filaments.

This is a technically strong and well written article addressing a problem of significant importance. I will note that it is difficult for me assess the value of its content to the field without seeing how this is implemented. The main value of this method is that it can do what other methods already do, but faster. My understanding is that much of the method is previously published and has been repackaged for use in this domain. In that sense, it is a tool. That is only of value if it is deployed in a manner that is well structured, documented, and at least somewhat usable and modifiable by technically capable users. Since the github link has not been included, I cannot assess this at the moment. Given that this is life sciences journal rather than a numerical methods journal, I think for publication at *eLife* it is critical that this be more than a methods article.

Without seeing this, I have a few questions. The computationally intensive elements of this method are in C++, but what about the more model specification oriented elements. Is this fully C++ or are you interfacing with a higher level language? Second, is this essentially an internal use code, or are you attempting to make this at least somewhat usable to other researchers. I strongly recommend the latter otherwise this will not really be an advancement over approaches such as Cytosim or MEDYAN. Faster but less usable is a losing combination.

I think this approach has strong potential, provided this issue is addressed. Below I'll note a few specific comments.

Specific comments

Line 74 – Here it is mentioned that all filaments will be considered to be strait and rigid but that this can be relaxed by jointing smaller segments together. While these filaments may have long persistence lengths relative to the cell size, their bending capacity is still vitally important to their dynamics and so if this is simple to implement, the authors should include this capability in the origination of this method.

Equation 3 – Please specify here what [s,p] are. As far as I can tell, these are not even fully defined in the SM. Also, is there really value in calling this a quaternion? Are you ever using either the geometric or algebraic properties of these? If not, I would suggest just defining it as a set of orientation variables. Also, for the general reader, it would be useful to describe why the orientation variable here is in R4 rather than R3.

SM Figures 4 / 5 – Can you clarify what you are quantifying for speed? Is this wall time / timestep or is it wall time / simulated second.

*Reviewer #3:*

In this article, the authors describe a new software for simulating cytoskeleton assemblies, and provide exiting examples of the software capabilities. The software offers the possibility to treat steric interactions as constraints rather than as stiff potentials, which should greatly improve performance. Thermodynamics of protein binding also seem carefully implemented.

While the software ships with great features, and fulfill a possible need in the field, there are several issues to take in consideration:

Software issues:

1- (No link to the software is provided in the article. aLens nonetheless is available on GitHub after a quick search, so this review is based on the current GitHub version)

My first concern is the difficulty to compile aLens. Installing aLens requires running an external script to alter one's environment (or significantly altering the compile file CMakeList.txt). Therefore, I was not able to compile this program, being unwilling to run a script to change my environment. Compiling of the software should be made much easier to be used by a wider, biology-oriented audience.

2- Modularity:

While the authors mention the cross linking model to have a modular design, the software itself does not appear to be modular. There is currently no possibility of having different types of proteins (or different type of filaments). This is a very strong limitation for its general use.

Scientific issues:

1 – Currently, aLens can only simulate straight filaments. The authors claim that aLens could easily as chain of short jointed segments. I would not think this easy, except possibly for the authors. However, many studies showed the paramount importance of bending rigidity in the mechanics of networks. Notably, Lenz and Gardel showed that because of this flexibility, networks tended to be globally contractile rather than extensile (filaments can buckle but not stretch). Therefore, aLens currently has access to only a manifold of the phase diagram, that may not be relevant e.g. for example 1, figure 2. Moreover, the filaments are not dynamic.

2 – While very interesting examples are provided, the authors do not provide a verification of their algorithm and implementation. While simple examples such as buckling cannot be addressed, maybe collective effects could be used as theoretical benchmarks? These could include collective effects of motors and nematic ordering, for which there exist theoretical results.

The authors therefore found success in simulating large systems with adequate thermodynamics. This seems like a notable technical advance. They were successful in providing impressive examples derived from the software. However, the lack of benchmarking (theoretical and performance-wise) mean that is not currently possible to assess exactly how successful the authors were.

The lack of filament flexibility and dynamics, as well as the impossibility to use more than one type of proteins will drastically limit its use by the biological community. The lack of code modularity will limit the possibility of external developers participating.

Overall, it seems that this is a very powerful approach, but the article needs to find a message. It could be emphasizing either:

– the implementation, in which case theoretical, and maybe performance benchmarks should be provided.

– the software package itself, in which case the software should be made more usable by the community, and possibly encompass a more general family of problems (or make them at least possible to implement via code modularity).

– the scientific results: in which case theoretical benchmark should be provided for simpler systems as a verification, and more complicated results should be put in their scientific context.

Currently, while very impressive technically, the software and associated article seem to fall short in each category.

---

## [Author Response]

Essential revisions:As you will see from the reports below, the referees were favourably impressed by the performance on your numerical method, which allows to stimulate a large number of filaments and motors in a reasonable time. This paper is seen as the presentation of a new tool illustrated by examples, rather than the presentation of new scientific results. The essential revisions include:1. Software availability (all referees) To be of value, the software should be freely available and useable. This does not seem to be the case at present.

We agree that it is important that *aLENS* be freely available and useable. The source code has been, and is, available on GitHub (https://github.com/flatironinstitute/aLENS) but this was not made sufficiently clear in the original submission. Now we have added a footnote to the last section about software availability. On GitHub detailed instructions have been provided for compiling the source and, as an aid to users, we have now added an automated script in the source code to install dependency libraries. We have also packaged the precompiled executable as a Docker image, freely available on Docker Hub (https://hub.docker.com/r/wenyan4work/alens), so that any interested users can run *aLENS* on any operating system, without compilation.

In sum, in this revision we have provided the full information of how to obtain and compile the source code, with additional resources to use the program in more user-friendly ways, e.g., precompiled executables.

2. Benchmarking (referees 1 and 3) It is important to provide some level of validation of the algorithm, which is currently missing. This can be done by comparing some aspects of the numerical results to existing analytical models. Regarding the binding and motor activity of cross linker, some simple toy model (e.g. gliding assay for motors, bound fraction for crosslinkers) would already be very informative, but more complex properties could also be considered as in Lera-Ramirez and Nedelec , Cytoskeleton 2019 (Theory of antiparallel microtubule overlap stabilization by motors and diffusible crosslinkers). Regarding collective effect, validation of the results could follow approaches such as the ones employed in Gao etal. PRE 2015 (Multiscale modeling and simulation of microtubule-motor-protein assemblies). Performance benchmark compared to other available algorithm should also be discussed.

We agree with the overall comment. To address this issue, we have added a new section *Verification and Benchmarking* to the manuscript, providing details about verification and performance benchmarks. On verification, one of the few available ground-truth cases, centrally relevant to this often-dense system, is the isotropic-nematic transition of rigid Brownian liquid crystals. We have shown in recent work [1] that we recover quantitatively the equation of state and isotropic-nematic phase transition for this system using the core steric interaction solver of *aLENS*. Of course, this example does not include motors. To address this, we chose two experimental systems that focus on different aspects of our method.

Reviewer 3 believed that *aLENS* did not have the capability to handle mixed motor types. This is not the case. Thus, as our first validation we compare with the experiments in [2], focusing on a simulated gliding assay in which directed transport of microtubules is controlled by mixtures of active and inactive motors. Our new Figure 2 in the revision shows that our simulation results are in excellent quantitative agreement with experiments over the entire range of active motor number fraction, while also demonstrating the capability of *aLENS* in handling multiple motor types. Further, our current *aLENS* implementation also supports simulation of arbitrary mixtures of filaments with different radii and lengths as well as spherical particles.

We next compare with the experiments in [3], focusing on the observed large scale collective behavior there. They show that in densely packed and highly crosslinked microtubule-XCTK2 assemblies (XCTK2 is a minus-end directed kinesin motor), microtubules show nearly constant motor-induced speeds with those speeds nearly independent of motor concentration and local mean polarity of the filament assemblage. This is a non-trivial phenomena that occurs in densely packed and highly crosslinked assemblies, and has also been observed in extract spindles [4]. Our new Figure 3 in the revision shows that our simulation results agree with these experiments. This confirms the theoretical interpretation given in [3], based on continuum modeling, that these highly aligned microtubule/motor systems can be seen as two interpenetrating active gels being pulled past each other by motors.

For verification, we chose to directly compare with experimental data, rather than with other simulation work, because we focus on the predictive power of our approach, and prefer to avoid debate about technical detail differences between our methods and other simulation tools. To this end, we show that our simulation methods and modeling can directly match experimental observations. We note that for all comparisons we chose motor parameters based on experimental estimates for that particular motor type, or for other members in the same motor family if the data for that particular motor is not available. In particular, we did not perform any parameter fitting process in these benchmark simulations, which nonetheless accurately reproduce the experimental measurements. These two direct comparisons with experiment cover multiple aspects of our method and implementation, including mixed motor species, long-time directed motion, densely packed assemblies, and large-scale collective interactions between motors and filaments. We believe that this provides convincing verification of the simulation methods.

For the referees’ interest, we note that we are preparing a new paper where we combined *aLENS* simulation results with analytic modeling to reveal a self-regulating mechanism for the “self-straining motion” of densely crosslinked nematic bundles of microtubules [3]. This lays the microscopic foundation of the continuum theory, and its generalizations, for such systems [3].

In the revision, we have also added a large scale parallelization efficiency test case to show the major advantage of *aLENS*. For a system with 4 million objects (1 million microtubules and 3 million motors) and 8 million constraints, we achieve almost ideal linear speed-up with increasing resources up to 1536 cores, at which point one timestep takes less than 1 second wall clock time. Such large-scale parallel performance has never been demonstrated by other methods in this field, to the best of our knowledge.

3. Modularity (referees 2 and 3). At present, only rigid filaments interacting with one type of crosslinker are presented. The extension flexible filaments and different types of interacting proteins are discussed as possibilities but are not implemented at the moment. Although this can be seen as the natural next step, this modularity is essential for the algorithm to be useful to the community.

Our current implementation supports an unlimited number of different species of motors, and we have provided one such example with two species of motors as the first comparative example in the new section *Verification and Benchmark* of the revision.

Regarding the incorporation of flexible filaments, we have added a section in the supplemental material covering two different methods for implementing bending rigidity. One method requires no modification of the current codebase as it involves only additional Hookean springs. The other incorporates the bending rigidity as a set of equality constraints, solved simultaneously with collision and doubly bound motors, within our constraint-optimization solver.

Our codebase is actually modular. It follows a standard object-oriented design, so that users can conveniently swap our motor model for their own models, or add more features including additional constraints (as above for bending rigidity), using standard C++ programming techniques for high performance. In addition to the methodology derivations in the supplemental material describing the motor model, we have also included a detailed document, along with the source code distributed via GitHub, describing the codebase implementation of the motor protein model Protein/ProteinModel.md. Our GitHub documentation provides a clear roadmap for developing additional modules.

Further development of *aLENS* does require some C++ programming knowledge but such expertise is widely found in academia and is certainly not limited to our group. Indeed, we have already witnessed the development of *aLENS* extensions in on-going development work with other collaborating groups.

Reviewer #1 (Recommendations for the authors):The study by Yan et al., developed a novel computational framework for modelling cytoskeletal cellular processes that allows for further investigations into the material properties associated with such processes. The computational methodology involves modelling cytoskeletal filaments as rigid spherocylinders while the Hookean law is employed to model crosslinkers. The computational algorithm, aLENS performs three key tasks in a sequential manner that lends itself naturally to high performance computing. To demonstrate the applicability and usefulness of aLENS, the authors present (i) self-aligning and buckling networks for a significantly large number of filaments than previously studied, and (ii) the interplay between polarity of motor walking and polarity of filaments, which seems to suggest that the ability of motors to continuously walk without end-pausing is crucial to effective polarity sorting. The authors also investigated the formation of asters, which seem to form when crosslinking motors reorganise filaments so that their minus ends are clustered and held tight by paused motors.Strengths1. The development of an alternative novel computational framework that offers far more flexibility, applicability and is scalable across multi-scales.2. The methodology overcomes the timescale limitations imposed by conventional explicit time-stepping methods that are key to modelling the dynamics of the cytoskeletal filaments and motors.3. aLENS utilizes efficiently high-performance parallel computing resources to scale to cellular scale systems.

We thank the reviewer for the positive feedback.

Weakness1. The lack of bench-marking that is associated with algorithm comparisons for performance and robustness.2. The lack of rigorous validation of the algorithm against suitably identified grounds truths.3. No clear demonstration of the efficiency of the algorithm and how it compares to current conventional algorithms of this nature.4. There are no clear comparisons between predictions of the computational algorithm and experimental data or observations.My recommendations to authors are as follows:(i) To consider rigorous validation of the computational methodology either by using synthetic or experimental data.(ii) To demonstrate computational efficiency and robustness of the algorithm by comparing results to current conventional methodologies.(iii) To demonstrate efficiency and accuracy when aLENS is compared to current conventional methodologies.

We appreciate the points the reviewer is making. As the reviewer can see, we have addressed many of these issues in our response to the editors above. Here we respond in more detail to the reviewer’s comment regarding comparison with conventional algorithms. One main focus of *aLENS* design is to guarantee the temporal stability for systems with steric interactions and the elastic forces arising from doubly bound crosslinking motors. On the former, as a good point of comparison we will compare *aLENS* with the use of steep and stiff WCA potentials to prevent geometric overlaps. As an aside, we note that steric interactions are often either entirely neglected (as in AFINES [5]) or approximated by soft potentials (as in Cytosim [6]).

First of all, the original unmodified WCA potential is unstable for the smallest timestep we tried ∆*t* = 4*.*43 × 10^−7^ s, and none of the simulations can generate meaningful data, unless in the limit of zero volume fraction where the microtubules do not touch each other.

The WCA potential must be softened for practical simulations. The strategy [8] is to linearize the WCA potential for *r < r_c_*, where *r_c_* is a cutoff distance, but maintain the continuity of the modified force at *r_c_*. This linearization strategy generates a constant repulsive force when the filaments are closer than *r_c_*. Then, the timestep size must be small enough such that the chance of a pair getting closer than *r_c_* is small. With this strategy, the softened WCA potential can reproduce the equation of state for the entire range of isotropic-nematic transition.

In *aLENS* we keep the order of accuracy at first order in time-step, comparable to the conventional explicit timestepping methods using WCA potentials, but decouple the stability requirement from the accuracy requirement. The constraint solver of *aLENS* allows stable treatment of both collision and crosslinker forces in an unconditionally stable method.

This means if the accuracy constraint is sufficiently high then conventional methods will be stable and the two methods will have comparable (first-order) accuracy. As shown in Author response image 1, *aLENS* is equally accurate as conventional methods in reproducing the equation of state at the same timestep size, but remains stable if the timestep is increased by 10 − 100×, with about 10% − 20% loss in accuracy, in accordance with its first-order convergence. For the such systems without crosslinkers, this stability may also be achieved using conventional softened WCA potentials, if the potential is further softened with increasing timesteps. This softening strategy, however, cannot be applied to the Hookean spring forces of doubly bound crosslinkers, without significantly affecting the desired mechanical properties of the crosslinkers. In our previous work [9], we used a similar explicit timestepping method to handle the crosslinkers which limited crosslinker stiffness to ∼ 1pNµm^−1^, which is significantly softer than realistic crosslinkers.

**Author response image 1. sa2fig1:** Comparison of the equation of state for 1µm microtubules at room temperature across the isotropicnematic transition range of volume fractions. *ϕ_cp_* = 0*.*9036 is the close packing volume fraction. The reference data is adapted from [7], generally accepted as the ground truth for rigid Brownian liquid crystal equation of state. Simulations using linearized WCA potential uses the code by [8], which has a constant repulsive force when the distance between rods is smaller than 0*.*812*D*_MT_, where *D*_MT_ is the microtubule diameter.

The linearized optimization problem is constructed such that even in the limit where the crosslinkers become infinitely stiff, i.e., non-compliant clamps or in the limit where an infinite number of crosslinkers connect a pair of filaments, the minimization problem is still convex and the solution remains stable.

This stability guarantee of *aLENS* is crucial in large-scale Brownian dynamics simulations because a large Brownian displacement or for a geometry where many crosslinkers bind two filaments, although the probability is very low, is almost surely generated for systems with tens of thousands of objects. In conventional potential type methods, such situations require very complicated ‘fail-safe’ mechanisms such as partially linearized potentials, soft-core potentials, force limiters, or displacement limiters. If these ‘fail-safe’ mechanisms are not properly fine-tuned, one unstable event between a single pair of filaments may crash the entire simulation. None of these are necessary in *aLENS* because *aLENS* is designed to be stable. When large Brownian jumps or other large stochastic crosslinker forces occur, locally the accuracy is lost for this object at this time, but this local error usually resumes to its ‘normal’ state and statistically the sampled system property is not affected. Therefore, the users of *aLENS* have the option to either quickly scan the parameter space with large timesteps if larger error is acceptable, or accurately scrutinize a few cases with small timesteps.

One major motivation of developing *aLENS* is to correctly simulate densely packed and crosslinked filament systems accurately. The editors might consider the possibility that *aLENS* is also providing the benchmarking computational problems that would be so very useful to this field.

Reviewer #2 (Recommendations for the authors):In this article, the authors present a new method for modeling cytoskeletal networks inside cells. In particular they model the interactions between filaments (microtubules in this case but the method is not limited to this) and motors. The main contribution appears to be in the efficiency of this method. The main challenge in this area of numerical research is handling steric interactions between the large numbers of interacting cytoskeletal filaments. Essentially, hard repulsion methods require very small timescales that can make reasonable simulations infeasible while soft repulsion approximations of these can lead to numerical issues. Their method takes a mathematically different approach to addressing this issue that may substantially speed these simulations allowing users to simulate more realistically sized systems consisting of up to 10^6^-10^7^ filaments.This is a technically strong and well written article addressing a problem of significant importance. I will note that it is difficult for me assess the value of its content to the field without seeing how this is implemented. The main value of this method is that it can do what other methods already do, but faster. My understanding is that much of the method is previously published and has been repackaged for use in this domain. In that sense, it is a tool. That is only of value if it is deployed in a manner that is well structured, documented, and at least somewhat usable and modifiable by technically capable users. Since the github link has not been included, I cannot assess this at the moment. Given that this is life sciences journal rather than a numerical methods journal, I think for publication at eLife it is critical that this be more than a methods article.Without seeing this, I have a few questions. The computationally intensive elements of this method are in C++, but what about the more model specification oriented elements. Is this fully C++ or are you interfacing with a higher level language? Second, is this essentially an internal use code, or are you attempting to make this at least somewhat usable to other researchers. I strongly recommend the latter otherwise this will not really be an advancement over approaches such as Cytosim or MEDYAN. Faster but less usable is a losing combination.I think this approach has strong potential, provided this issue is addressed. Below I'll note a few specific comments.

We thank the reviewer for this constructive feedback. We have addressed some of these issues above in our response to the editors. We do disagree with the statement that the main value in the new work is that it does what other methods already do, but faster. While it may appear at first glance that the new method is similar to others that are currently available, the choice of model and implementation are significantly different. That allows *aLENS* to be more accurate.

In particular, we accurately handle the steric collisions and crosslinking interactions in a unified constraint solver while maintaining very strong temporal integration stability. This allows *aLENS* to be more accurate for treating cytoskeletal systems. For example, in previous work studying rods in the absence of motors we demonstrated that the constraint method can accurately and fully reproduce the equation-ofstate and isotropic-nematic phase transition in dense systems of thermally fluctuating rods. By contrast, in other currently available simulation packages steric interactions are sometimes neglected (in AFINES [5]) or approximated by soft potentials (in Cytosim [6]). These approximations are suitable for relatively dilute systems, but are less accurate for dense packing.

Moreover, our treatment of motor/crosslinker binding and unbinding correctly recapitulates the distribution and chemical kinetics of crosslinking proteins in the equilibrium limit, which is important for quantitative modeling of motor mechanochemistry. Other available packages make use of ad hoc rules for crosslinker binding and unbinding, which for some problems can give physically unrealistic results.

Finally, the method here incorporates Brownian dynamics, which is neglected in some other packages (e.g., MEDYAN).

For these reasons, we believe that in addition to the improved computational efficiency, the range of capabilities that aLENS offers are not present in any other available method.

In the revision we provide further information on the implementation and availability of *aLENS*. In short, it is fully implemented in C++ to ensure efficiency, following a standard object-oriented design. It is available as open source code on GitHub, and we also provide a precompiled ready-to-run binary executable through Docker Hub. The design is modular such that, for example, the motor model can be swapped out for other models.

Specific commentsLine 74 – Here it is mentioned that all filaments will be considered to be strait and rigid but that this can be relaxed by jointing smaller segments together. While these filaments may have long persistence lengths relative to the cell size, their bending capacity is still vitally important to their dynamics and so if this is simple to implement, the authors should include this capability in the origination of this method.

We agree with the reviewer that filament bending can be quite important in cytoskeletal systems. Accordingly, we have now included in the revised supplemental material a detailed description of how flexible filaments with bending rigidity can be implemented within the constraint solver in *aLENS* with two different methods. One of them requires no new additions to the codebase. We can further prove that both methods are still stable even in the limit of infinite bending rigidity, similar to the stability for infinitely stiff doubly bound springs. One of our collaborators is working on the extension and will present the method and applications in a separate publication.

Equation 3 – Please specify here what [s,p] are. As far as I can tell, these are not even fully defined in the SM. Also, is there really value in calling this a quaternion? Are you ever using either the geometric or algebraic properties of these? If not, I would suggest just defining it as a set of orientation variables. Also, for the general reader, it would be useful to describe why the orientation variable here is in R4 rather than R3.

We appreciate the reviewer’s suggestion to clarify this point. In response to this comment, we have clarified the definition in the revised main text. In particular, the use of quaternions is a standard approach in rigid body rotational kinematics. [*s,p*] are the scalar and vector parts of the quaternion, respectively. Representing orientations using quaternions has several crucial advantages, including its compact memory footprint (4 for a quaternion but 9 for a rotation matrix) and the important singularity-free property (the use of Euler angles may lead to the problem of gimbal locks). We therefore make use of this standard practice in the computational geometry of rigid body kinematics.

SM Figures 4 / 5 – Can you clarify what you are quantifying for speed? Is this wall time / timestep or is it wall time / simulated second.

The units are wall clock time per simulation timestep. We have clarified this in the revised figure captions.

Reviewer #3 (Recommendations for the authors):In this article, the authors describe a new software for simulating cytoskeleton assemblies, and provide exiting examples of the software capabilities. The software offers the possibility to treat steric interactions as constraints rather than as stiff potentials, which should greatly improve performance. Thermodynamics of protein binding also seem carefully implemented.While the software ships with great features, and fulfill a possible need in the field, there are several issues to take in consideration:

We thank the reviewer for the constructive comments.

Software issues:1- (No link to the software is provided in the article. aLens nonetheless is available on GitHub after a quick search, so this review is based on the current GitHub version)My first concern is the difficulty to compile aLens. Installing aLens requires running an external script to alter one's environment (or significantly altering the compile file CMakeList.txt). Therefore, I was not able to compile this program, being unwilling to run a script to change my environment. Compiling of the software should be made much easier to be used by a wider, biology-oriented audience.

We thank the reviewer for this important comment. We hope the reviewer will be satisfied with our response, which can be found in our reply to the editor’s comments.

2- Modularity:While the authors mention the cross linking model to have a modular design, the software itself does not appear to be modular. There is currently no possibility of having different types of proteins (or different type of filaments). This is a very strong limitation for its general use.

Thank you again for this comment. And, again, we think the reviewer will be satisfied on this through our response to the editor’s comments. In short, *aLENS* does have a modular design, and is perfectly capable of simultaneously handling multiple motor types, as we now demonstrate in our new section on verification and validation. Further, our current *aLENS* implementation also supports simulation of arbitrary mixtures of filaments with different radii and lengths as well as spherical particles.

Scientific issues:1 – Currently, aLens can only simulate straight filaments. The authors claim that aLens could easily as chain of short jointed segments. I would not think this easy, except possibly for the authors. However, many studies showed the paramount importance of bending rigidity in the mechanics of networks. Notably, Lenz and Gardel showed that because of this flexibility, networks tended to be globally contractile rather than extensile (filaments can buckle but not stretch). Therefore, aLens currently has access to only a manifold of the phase diagram, that may not be relevant e.g. for example 1, figure 2. Moreover, the filaments are not dynamic.

We agree with the reviewer that filament flexibility and polymerization dynamics are important in cytoskeletal systems. Therefore, we have included a new section in the supplemental material to show how flexible filaments with adjustable extensional and bending rigidity can be implemented in *aLENS*, via two different methods. The first method requires no modifications to the current codebase, while the second method incorporates bending joints as new constraints in the constrained minimization solver but requires some code extensions. Both types of method/proposed implementation resolve collisions, doubly bound motor forces, and bending rigidity in a single unified solver. Further, both of these remain stable in the limit where the extensional and bending rigidity of filaments are infinite.

While we have not yet implemented filament polymerization dynamics, this could be done in a straightforward way since the change of filament length can be completely decoupled from the geometric constraint optimization solver. The only necessary improvement to the codebase will be to address how a crosslinker interacts with a dynamic filament, i.e., how the crosslinker tracks its own binding status and location while walking along a dynamic filament. Although currently not implemented, this feature is in our future roadmap of *aLENS*.

2 – While very interesting examples are provided, the authors do not provide a verification of their algorithm and implementation. While simple examples such as buckling cannot be addressed, maybe collective effects could be used as theoretical benchmarks? These could include collective effects of motors and nematic ordering, for which there exist theoretical results.The authors therefore found success in simulating large systems with adequate thermodynamics. This seems like a notable technical advance. They were successful in providing impressive examples derived from the software. However, the lack of benchmarking (theoretical and performance-wise) mean that is not currently possible to assess exactly how successful the authors were.The lack of filament flexibility and dynamics, as well as the impossibility to use more than one type of proteins will drastically limit its use by the biological community. The lack of code modularity will limit the possibility of external developers participating.

Thank you for the comments. We have addressed many of them directly and constructively in our response to the editors. In short, we have added a new *Verification and Benchmarks* section in the revised manuscript to verify that our simulation results are in good quantitative accord with experimental measurements for both small-scale, single microtubule systems with multiple motor types, and large-scale, many-filament systems that are driven collectively by motor-filament interactions. There you will also find a large scale parallel efficiency test, showing nearly ideal linear efficiency for 4 million objects on 1536 cores. Such scaling behavior and efficiency is not available with other methods.

Overall, it seems that this is a very powerful approach, but the article needs to find a message. It could be emphasizing either:– the implementation, in which case theoretical, and maybe performance benchmarks should be provided.– the software package itself, in which case the software should be made more usable by the community, and possibly encompass a more general family of problems (or make them at least possible to implement via code modularity).– the scientific results: in which case theoretical benchmark should be provided for simpler systems as a verification, and more complicated results should be put in their scientific context.Currently, while very impressive technically, the software and associated article seem to fall short in each category.

We thank the reviewer for their comments. We found them very helpful. We believe that in the revision we have strengthened the manuscript on all these fronts. We have provided both theoretical and performance benchmarks, improved code accessibility and usability for external users, emphasized its modularity, and used *aLENS* to solve problems of current scientific import.

One of our goals is to provide the cytoskeletal physics community with a new and transformative software tool which they can apply on their own problems and to which they can contribute new functionalities. That said, using *aLENS* assumes a certain level of programming expertise but that expertise is certainly not singular to our own group and is widely found in academia.

In summary, *aLENS* is designed to simulate cytoskeletal systems that are challenging to model with existing software, by enabling efficient and accurate treatment of steric interactions and crosslinking forces, chemical kinetics, and Brownian motion in a high-performance parallel implementation. This gives access to lengthscales and timescales of relevance to cell biology. The design of *aLENS* also allows continuous method development. New features such as bending, polymerization, and even hydrodynamics can and are being incorporated into *aLENS*.

References

[1] Wen Yan, Huan Zhang, and Michael J. Shelley. Computing collision stress in assemblies of active spherocylinders: Applications of a fast and generic geometric method. *The Journal of Chemical Physics*, 150(6):064109, 2019.

[2] Lara Scharrel, Rui Ma, René Schneider, Frank Jülicher, and Stefan Diez. Multimotor Transport in a System of Active and Inactive Kinesin-1 Motors. *Biophysical Journal*, 107(2):365–372, 2014.

[3] Sebastian Fürthauer, Bezia Lemma, Peter J. Foster, Stephanie C. Ems-McClung, Che-Hang Yu, Claire E. Walczak, Zvonimir Dogic, Daniel J. Needleman, and Michael J. Shelley. Self-straining of actively crosslinked microtubule networks. *Nature Physics*, 15(12):1295–1300, 2019.

[4] Daniel J. Needleman, Aaron Groen, Ryoma Ohi, Tom Maresca, Leonid Mirny, and Tim Mitchison. Fast microtubule dynamics in meiotic spindles measured by single molecule imaging: Evidence that the spindle environment does not stabilize microtubules. *Molecular Biology of the Cell*, 21(2):323–333, 2010. PMID: 19940016.

[5] Simon L. Freedman, Shiladitya Banerjee, Glen M. Hocky, and Aaron R. Dinner. A Versatile Framework for Simulating the Dynamic Mechanical Structure of Cytoskeletal Networks. *Biophysical Journal*, 113(2):448–460, 2017.

[6] Francois Nedelec and Dietrich Foethke. Collective Langevin dynamics of flexible cytoskeletal fibers. *New Journal of Physics*, 9(11):427–427, 2007.

[7] Peter Bolhuis and Daan Frenkel. Tracing the phase boundaries of hard spherocylinders. *The Journal of Chemical Physics*, 106(2):666–687, 1997.

[8] Robert Blackwell, Oliver Sweezy-Schindler, Christopher Baldwin, Loren E. Hough, Matthew A. Glaser, and M. D. Betterton. Microscopic origins of anisotropic active stress in motor-driven nematic liquid crystals. *Soft Matter*, 12(10):2676–2687, 2016.

[9] Peter J. Foster, Wen Yan, Sebastian Fürthauer, Michael J. Shelley, and Daniel J. Needleman. Connecting macroscopic dynamics with microscopic properties in active microtubule network contraction. *New Journal of Physics*, 19(12):125011, 2017.